# Recent Progress on Semiconductor Heterogeneous Photocatalysts in Clean Energy Production and Environmental Remediation

**Nahal Goodarzi** [1], **Zahra Ashrafi-Peyman** [2], **Elahe Khani** [2] and **Alireza Z. Moshfegh** [1,2,*]

1. Institute for Convergence Science & Technology (ICST), Sharif University of Technology, Tehran P.O. Box 11365-8639, Iran; nahal.gdrz@gmail.com
2. Department of Physics, Sharif University of Technology, Tehran P.O. Box 11555-9161, Iran; zahraashrafipeyman@yahoo.com (Z.A.-P.); elahe.khani.phy@gmail.com (E.K.)
* Correspondence: moshfegh@sharif.edu

**Abstract:** Semiconductor-based photocatalytic reactions are a practical class of advanced oxidation processes (AOPs) to address energy scarcity and environmental pollution. By utilizing solar energy as a clean, abundant, and renewable source, this process offers numerous advantages, including high efficiency, eco-friendliness, and low cost. In this review, we present several methods to construct various photocatalyst systems with excellent visible light absorption and efficient charge carrier separation ability through the optimization of materials design and reaction conditions. Then it introduces the fundamentals of photocatalysis in both clean energy generation and environmental remediation. In the other parts, we introduce various approaches to enhance photocatalytic activity by applying different strategies, including semiconductor structure modification (e.g., morphology regulation, co-catalysts decoration, doping, defect engineering, surface sensitization, heterojunction construction) and tuning and optimizing reaction conditions (such as photocatalyst concentration, initial contaminant concentration, pH, reaction temperature, light intensity, charge-carrier scavengers). Then, a comparative study on the photocatalytic performance of the various recently examined photocatalysts applied in both clean energy production and environmental remediation will be discussed. To realize these goals, different photocatalytic reactions including $H_2$ production via water splitting, $CO_2$ reduction to value-added products, dye, and drug photodegradation to lessen toxic chemicals, will be presented. Subsequently, we report dual-functional photocatalysis systems for simultaneous energy production and pollutant photodegradation for efficient reactions. Then, a brief discussion about the industrial and economical applications of photocatalysts is described. The report follows by introducing the application of artificial intelligence and machine learning in the design and selection of an innovative photocatalyst in energy and environmental issues. Finally, a summary and future research directions toward developing photocatalytic systems with significantly improved efficiency and stability will be provided.

**Keywords:** advanced oxidation process (AOPs); semiconductor-based photocatalysts; $H_2$ production; water splitting; $CO_2$ reduction; dye and drug photodegradation; dual-functional photocatalysis; industrial photocatalyst; artificial intelligence; machine learning

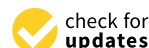



## 1. Introduction

With the continuous increase in global population and industrialization, the necessity for clean energy and a sustainable environment is increasing dramatically [1]. Based on U.S. Energy Information Administration (EIA) survey in 2019, energy consumption worldwide is expected to rise by approximately 50% between 2018 and 2050 [2]. About 83% of today's energy supply comes from fossil fuels (e.g., gas, petroleum, and coal). Fossil fuel combustion is responsible for releasing large amounts of $CO_2$, accounting for around 76% of annual greenhouse gas emissions, which creates severe problems for the environment and public

health, such as global warming and climate change [3,4]. Accordingly, eco-friendly and renewable energy resources, including solar, biomass, wind, geothermal, and hydropower, have been successfully exploited as practical alternatives for fossil fuels to address these problems [5,6]. Among several energy as mentioned above, solar energy, which is emitted from the sun at a rate of $3 \times 10^{23}$ kW and $1.8 \times 10^{14}$ kW of which is intercepted by the earth, is the most affordable and applicable energy source [7]. Hence, designing an efficient technique to transform solar energy into hydrogen and clean fuels such as carbon-based fuels through water splitting and carbon dioxide ($CO_2$) reduction, respectively, shows great potential to address both energy crises and environmental problems [8,9]. As a non-carbon-based energy carrier, hydrogen energy has attracted considerable attention due to its numerous advantages and may become a chemical fuel in the future. It has been proven that hydrogen has the potential to be a preferred alternative to reduce human society's reliance on non-renewable resources and decrease environmental concerns. Water, which covers 71% of the earth's surface, has excellent potential as a source of hydrogen energy. Using water power, we can extract $H_2$ by using an appropriate photocatalyst. This process not only provides a means of generating energy, but also helps to reduce greenhouse gas emissions. While the high flammability of hydrogen is considered a disadvantage, it is also beneficial because it eliminates the need for additional energy to combust the fuel. Additionally, once released, hydrogen diffuses quickly and is non-toxic. These properties make hydrogen a potentially safe and efficient energy source for the future [10].

There are various approaches for solar hydrogen production, including photocatalytic (PC) [11], photothermal catalytic (PTC) [12], photovoltaic electrochemical (PV-EC) [13], photoelectrochemical (PEC) [14], and photobiological (PB) [15]. In addition, some of these methods, including PV-EC [16], PEC [17], and PC [18], are also utilized for $CO_2$ reduction. The core of all the mentioned methods is the semiconductor photocatalysis, which is known as one of the promising approaches in carbon dioxide reduction and hydrogen production [19]. Inspiring natural photosynthesis, Figure 1 illustrates the application of solar energy by using semiconductor photocatalysts toward hydrogen generation from water splitting (Figure 1a) and $CO_2$ reduction to valuable chemicals (Figure 1b).

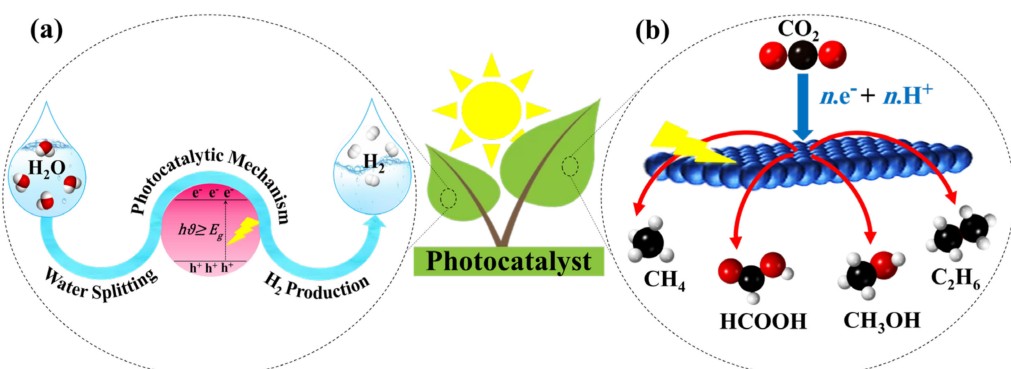

**Figure 1.** A schematic diagram of (**a**) hydrogen production from water splitting and (**b**) reduction of $CO_2$ to useful chemicals using solar light.

A timeline of hydrogen energy, highlighting its evolution from atom to fuel cell, and its potential application as a chemical fuel is schematically shown in Figure 2. Several research organizations have reported the practical feasibility of hydrogen-based fuel cells. Between 1950 and 2000, extensive research was carried out in developing of infrastructure and storage mechanisms from small to large-scale applications. Since 2000, the focus has shifted towards the commercializing of hydrogen energy and conducting system integration research. Based on this trend, it is predicted that by 2040, there will be significant growth in the hydrogen energy market and an increasing number of initiatives to replace conventional fuels with hydrogen energy [10,20].

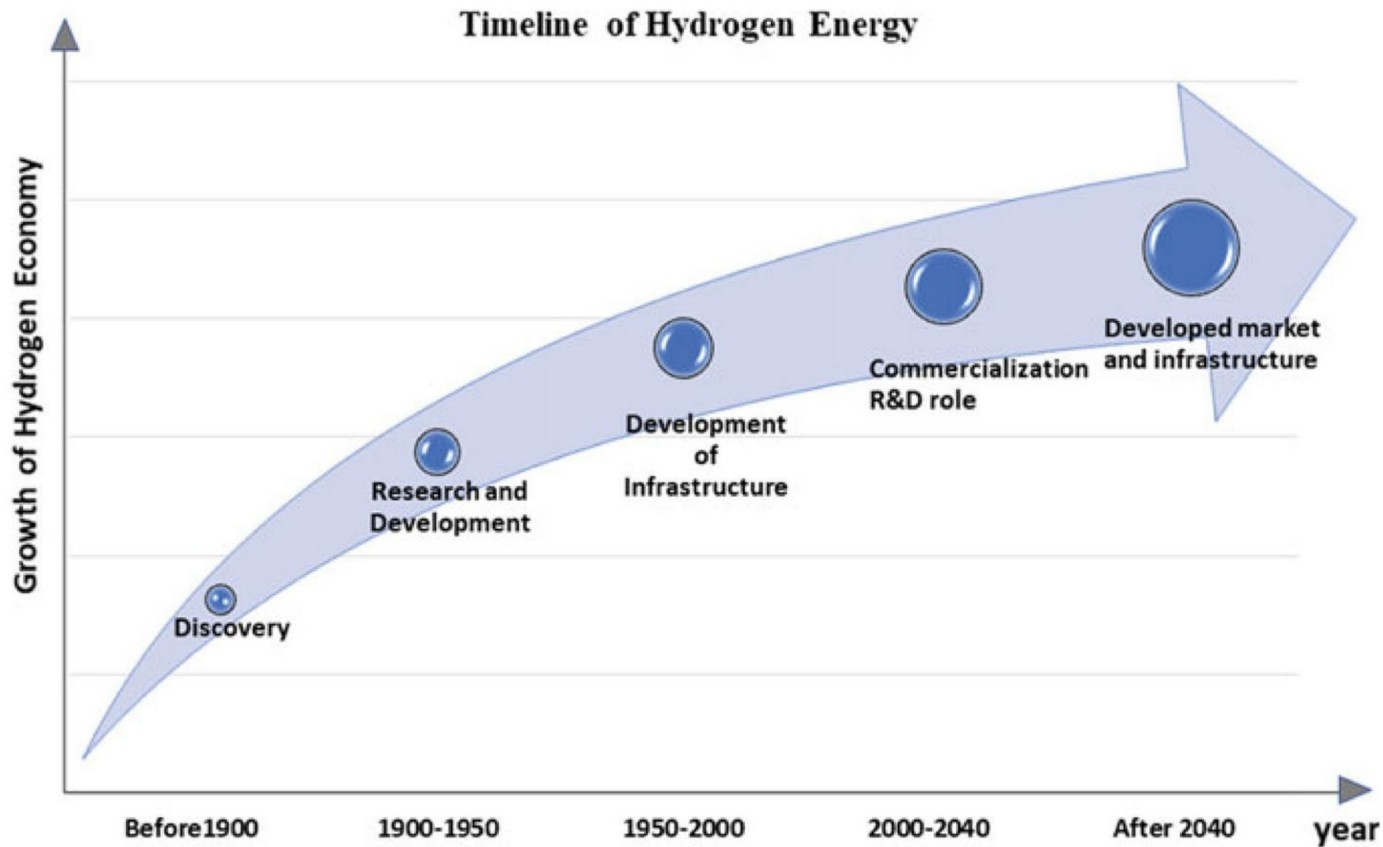

**Figure 2.** Timeline of hydrogen energy. Reproduced with permission from ref. [10].

The other most important aspect of environmental remediation in the modern world of today is water purification. Regarding the expansion of human activities, a significant number of organic contaminants, including industrial dyes, disinfectants, pesticides, agricultural fertilizers and pharmaceuticals compounds, as well as inorganic pollutants, including heavy metals, salts, solvents, and nutrients, are daily being released into the wastewater streams [21,22]. A schematic of various organic and inorganic water pollution sources is depicted in Figure 3.

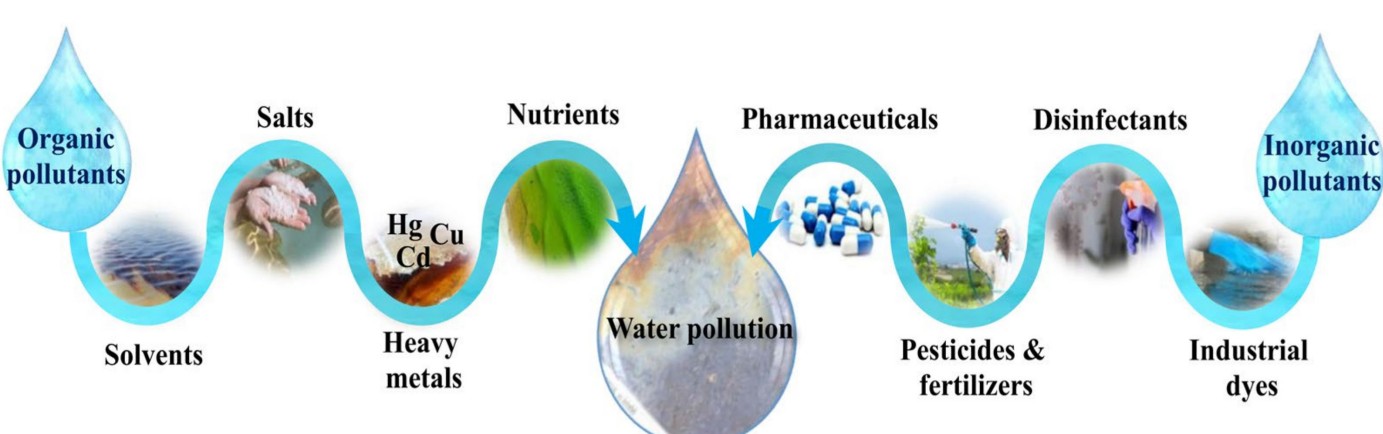

**Figure 3.** Various organic and inorganic sources of water pollution in the environment.

Long-term persistence in the environment and the water solubility of a wide range of compounds mentioned above have a significant potential to cause severe problems to the environment and health globally [21]. To overcome these issues, various conventional purification methods, including physical techniques (e.g., adsorption, membrane separation, flocculation, and coagulation), chemical methods (e.g., advanced oxidation process (AOPs)), and several microorganism-assistant biological treatments have been extensively used [23]. Physical methods by using adsorption techniques can remove organic and inorganic pollutants under different operating conditions [24,25]. However, the incomplete removal of organic pollutants and the generation of waste products during treatment can lead to several challenges for the desired process [24,26]. On the other hand, despite the excellent selectivity, membrane separation technology often leads to high operating costs and increased energy consumption because of membrane fouling [24,27]. Biological treatments using microbiological organisms will play an essential role in removing emergent contaminants from wastewater in the future. However, the need for physicochemical pretreatment and generation of biological sludge are the significant disadvantages of these methods [28]. Chemical methods, significantly advanced oxidation processes (AOPs), can be considered as a beneficial alternative in water purification treatments [29]. AOPs, introduced in 1981 by Glaze et al. [30], are defined as water purification processes by generating a sufficient amount of free hydroxyl radicals ($^\bullet$OH) as one of the most effective reactive oxygen species (ROS). In addition to hydroxyl radicals, other reactive species, including hydrogen peroxide ($H_2O_2$), anion radical ($^\bullet O_2^-$), and singlet oxygen ($^1O_2$), can be generated in AOPs [31]. Meanwhile, $^1O_2$ is a highly selective oxidant that can quickly oxidize electron-rich moieties in organic pollutants owning to its electrophilic nature. This form of oxygen species possesses a prolonged lifetime (nearly 2–4 μs) and high concentration (e.g., $10^{-14}$–$10^{-11}$ M) in water and demonstrates excellent resilience to inorganic anions and natural organic matter in wastewater [32]. These properties make this species suitable for selectively oxidizing high-priority contaminants commonly found in wastewater, such as endocrine-disrupting chemicals (EDCs), antibiotics, and pharmaceuticals [33]. Moreover, $^1O_2$ can be utilized for the inactivation of pathogenic microorganisms and the elimination of antibiotic-resistance genes in both natural and drinking water [34]. Several AOPs systems, such as catalytic ozonation, hydrogen peroxide ($H_2O_2$), persulfate (PS) activation processes, and photocatalysis, can be applied to generate $^1O_2$ [32]. In some ozone-based systems, $O_3$ can be activated and produce free peroxide species ($^\bullet O_2$) that can be directly converted to $^1O_2$ through an electron transfer mechanism. $H_2O_2$, as a common oxidant, can generate a variety of ROS after catalytic activation, followed by $^1O_2$ generation in the system. In persulfate (PS)-based AOPs, in the presence of carbon-based materials, the carbonyl (C=O) functional groups on the catalyst reacts with PS and generate $^1O_2$. Among the aforementioned approaches, photocatalysis is the most effective method to produce $^1O_2$ through either a directly energy transfer on the photocatalyst or indirectly from other ROS. Despite all the advantages of $^1O_2$, this species has low redox potential (0.81 V/SHE). As a result, its reaction rate with most organic compounds is lower than that of observed with radicals [35]. Owing to the high oxidation ability of the $^\bullet$OH radical (E° ($^\bullet$OH/$H_2O$) = 2.80 V/SHE), this species acts as a powerful agent to degrade a broad spectrum of toxic organic pollutants into harmless compounds and even mineralize them into $CO_2$ and $H_2O$ [36,37]. There are six types of AOPs based on the source nature of the $^\bullet$OH radical generation, including (a) photocatalysis [38], (b) UV–$H_2O_2$ processes [39], (c) photo-Fenton [40], (d) ozonation [41], (e) electro-Fenton [42], and (f) Sonlysis [43]. Different types of AOPs processes are depicted in Figure 4a–f.

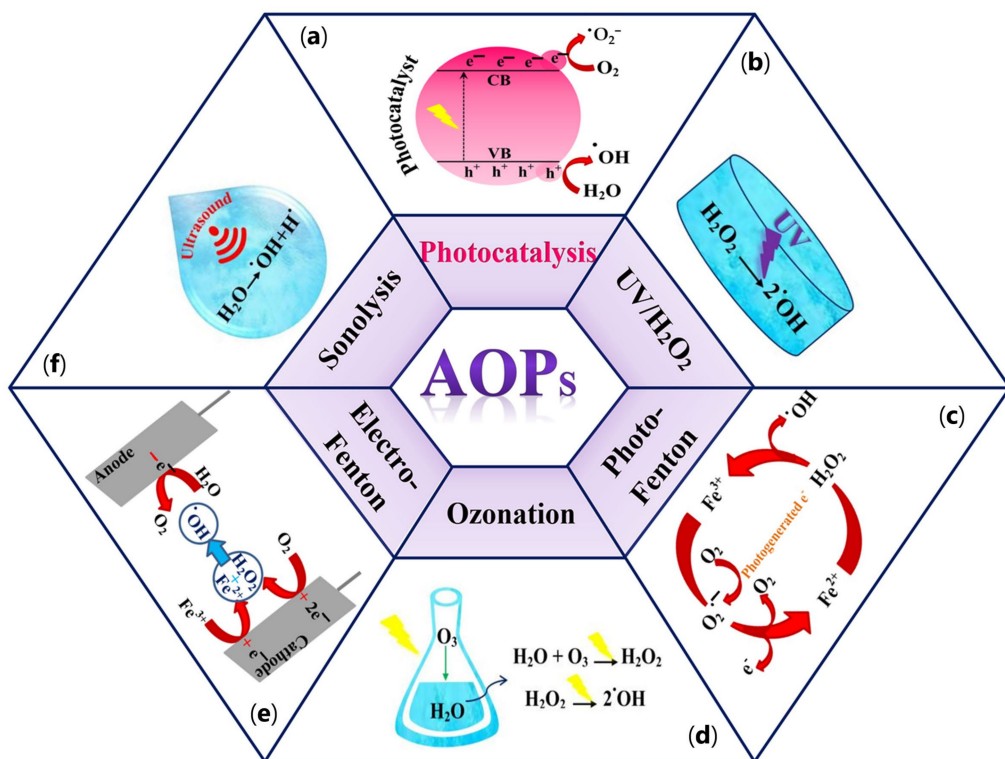

**Figure 4.** Different types of AOPs processes. (**a**) photocatalysis; (**b**) UV-$H_2O_2$ processes; (**c**) photo-Fenton; (**d**) ozonation; (**e**) electro-Fenton; (**f**) sonolysis.

Among various types of AOPs processes, the semiconductor-based photocatalytic process is highly expected to be the more desirable approach for wastewater treatment owing to its excellent properties, such as its environmentally friendly nature, simple manufacturing, complete mineralization capability, and reusability. Furthermore, the semiconductor photocatalyst can effectively decompose organic contaminants because of its high surface area using solar energy via a simple route and low operating cost without producing harmful byproducts [37,44,45].

Concerning the history of photocatalysis, it is established that the idea of artificial photosynthesis was first proposed by Verne [46] in 1874, but it took over 100 years for its realization in the lab. The light-driven oxygen evolution was first carried out by Boddy [47] in 1968 using n-type rutile ($TiO_2$). Müller's report in 1969 on the ability of ZnO to decompose isopropanol under UV light, demonstrated the potential of photocatalysts to degrade organic contaminants in water. Subsequently, a $TiO_2$-based semiconductor and a platinum-black electrode were historically employed by Fujishima and Honda in 1972 for photoelectrochemical (PEC) water splitting. In this period, extensive research was performed on a wide range of metal oxides (e.g., ZnO, NiO, and $WO_3$), metal sulfides (e.g., CdS, and ZnS), and mixed metal oxides (e.g., $SrTiO_3$) to degrade a wide range of organic contaminant. Between 1994 and 2000, significant progress was made in understanding the phenomenon and mechanism of photocatalysis due to the development of semiconductor band theory and the utilization of advanced characterization techniques. Since 2000, nanotechnology has attracted considerable attention, leading to the development of various nano-sized photocatalysts with several advantages, including increased charge transfer rates, larger specific surface area, and increased active sites. From 2011 to 2020, photocatalytic recycling attracted increasing attention due to the biosafety aspects of applying nanomaterials. The subsequent steps focused on developing many visible light photocatalysts to enhance photoconversion efficiency. Notably, research focus has shifted towards areas such as black $TiO_2$, non-$TiO_2$ junctions, plasmon-based photocatalysts, and catalyst design and tailoring its structure for a specific need [48,49]. Figure 5 shows a schematic

diagram of historical evolution of photocatalysts to remove water contaminants. Hence, the development of the photocatalytic process by using highly effective semiconductors with suitable bandgap positions has attracted a great deal of interest concerning clean energy generation, and environmental restoration in recent years. Based on the Scopus citation database during 2005–2023, through the keywords "photocatal*", "environment*", and "energy" in the title, abstract, and keywords, the number of publications on photocatalytic energy production and environmental remediation shows a continuous increasing trend which indicates the importance of this active field (Figure 6).

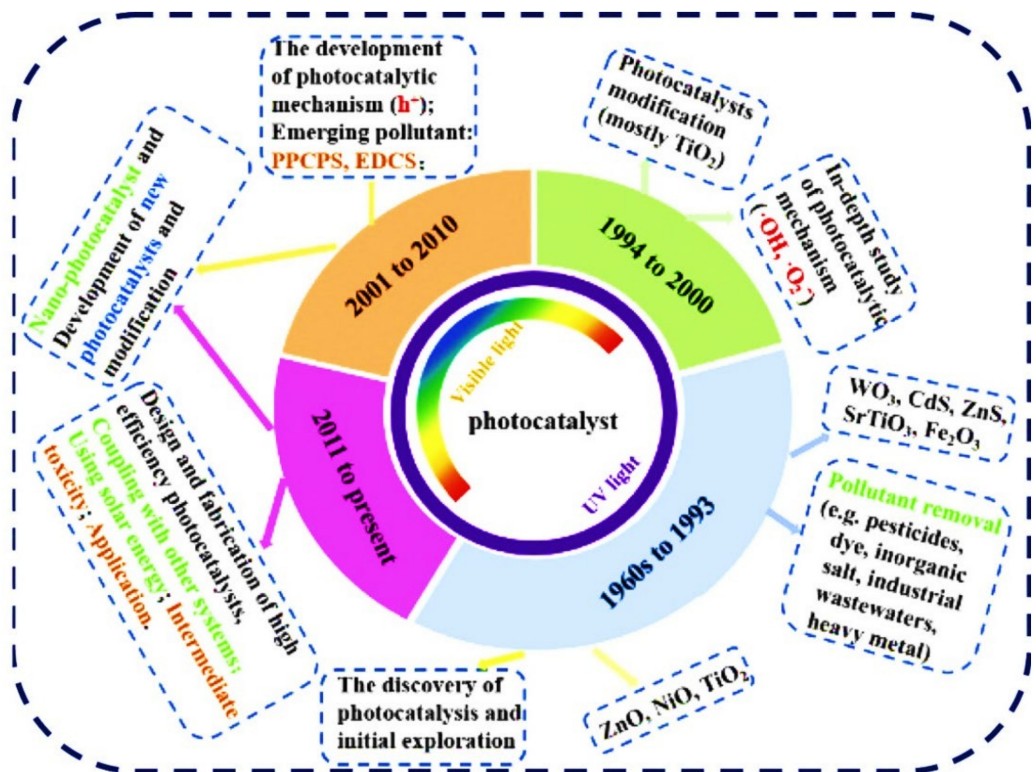

**Figure 5.** A schematic diagram of the historical evolution of photocatalysts for water contaminant removal. Reproduced with permission from ref. [48].

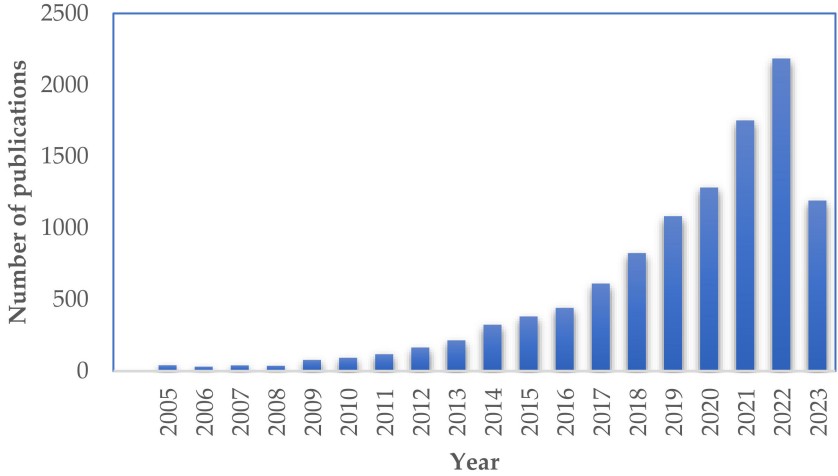

**Figure 6.** The number of publications during 2000 to 29 May 2023 in the Scopus citation database using keywords in (TITLE-ABS KEY("photocatal*")) AND (TITLE-ABS-KEY ("environment*")) AND (TITLE-ABS-KEY ("energy")).

This review aims to give a general overview of the fundamentals of photocatalysis and to provide the latest developments on the synthesis and applications of semiconductor heterogeneous photocatalysts for generating clean energy and photocatalytic wastewater treatment with the goal of environmental remediation.

## 2. Fundamentals of the Photocatalysis

### 2.1. Thermodynamics and Kinetics of the Photocatalytic Process

The thermodynamics of a photocatalytic reaction, determining whether the reaction will proceed, depends on the utilization rate of sunlight by the photocatalyst. In general, semiconductor photocatalysts can effectively catalyze both energetically downhill reactions (e.g., degradation of pollutants) and energetically uphill reactions (e.g., water splitting) [50,51]. On the other hand, the kinetics of the process, which describes the reaction rate, is related to the efficiency of charge-carrier separation and their transfer capability that is affected by the photocatalyst surface [51].

#### 2.1.1. Thermodynamics Aspects

In order to carry out a photocatalytic process thermodynamically, the semiconductor's conduction band (CB) and valence band (VB) edges should span the redox potentials of the reaction. Figure 7 shows the CB and VB positions of several typical inorganic and organic semiconductor photocatalysts at pH = 7. In terms of degradation of pollutants, the VB and CB edge position of the selected semiconductor must be more positive and negative than the potential of $^{-}OH/^{\bullet}OH$ (2.3 V vs. NHE) and $O_2/^{\bullet}O_2^{-}$ (−0.33 V vs. NHE), respectively. Similarly, for an ideal water-splitting reaction, the VB and CB edge position of the semiconductor should be more positive and negative than the redox potential of $H_2O/O_2$ (0.82 V vs. NHE) and $H^{+}/H_2$ (−0.41 V vs. NHE), respectively. For $CO_2$ reduction, different reduction potentials associated with different multi-electron reaction pathways exist [52]. In general, the more electrons involved in the reduction reaction, the more thermodynamically favorable the process due to the lower negative reduction potential.

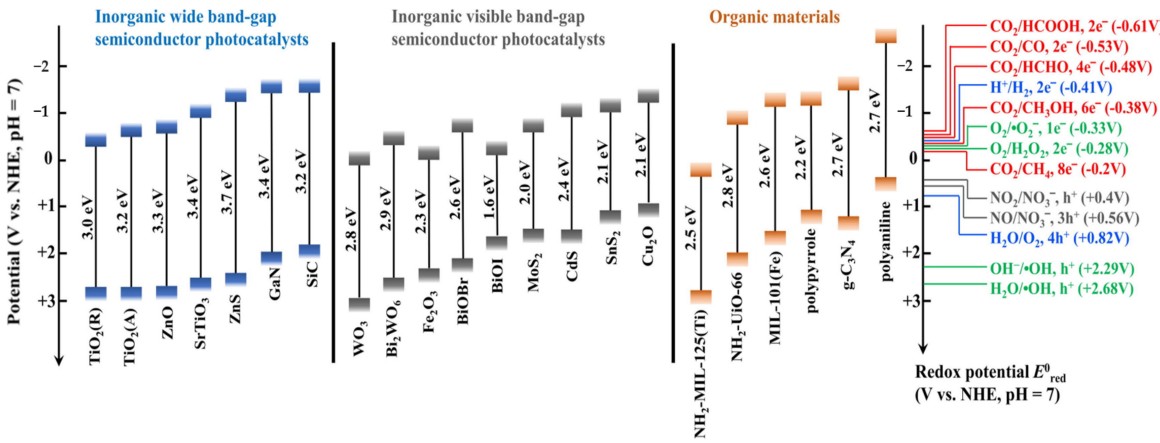

**Figure 7.** CB and VB positions of several typical inorganic and organic semiconductor photocatalysts. Reproduced with permission from ref. [52].

#### 2.1.2. Kinetics Aspects

Based on mass-action law, photocatalytic elimination of organic contaminants in aqueous media is a quasi-first-order reaction, and the kinetics of this reaction are described by Langmuir–Hinshelwood mechanism. The rate of the photocatalytic reaction (r) is calculated using the following equation:

$$r = -\frac{dC(t)}{dt} = k_{obs}C(t) \tag{1}$$

where r is the degradation rate, C(t) is the organic pollutants concentration at a given time t (mol·L$^{-1}$), and k$_{obs}$ (min$^{-1}$) is the observed rate constant. At the deficient concentration of organic pollutants, Langmuir kinetic model can be reduced to this equation:

$$\ln\frac{C_0}{C_t} = k_{app}t \tag{2}$$

where C$_t$ (mol·L$^{-1}$) and C$_0$ (mol·L$^{-1}$) are the organic contaminant concentration at time t and 0, and k$_{app}$ (min$^{-1}$) is the apparent rate constant. The higher the k$_{app}$ value, the better the photocatalytic activity. The slope of the linear correlation between time and ln$\frac{C_0}{C_t}$ represented the photodegradation rate [53,54].

### 2.2. Mechanism of the Photocatalytic Process

The general schematic of a typical photocatalytic reaction is shown in Figure 8. Upon light irradiation (ultraviolet-visible region) with equal or higher energy than the bandgap (E$_g$) energy of an appropriate semiconductor (E$_{h\nu} \geq$ E$_g$), excitation occurs and electrons (e$^-$) from the VB of a semiconductor photocatalyst transfer to its CB followed by generating holes (h$^+$) at its valence band (Equation (3)). These charge carriers can participate in oxidation–reduction reactions on the surfaces of a photocatalyst.

$$\text{photocatalyst} + h\nu \text{ (UV or visible light)} \rightarrow h^+(\text{VB}) + e^-(\text{CB}) \tag{3}$$

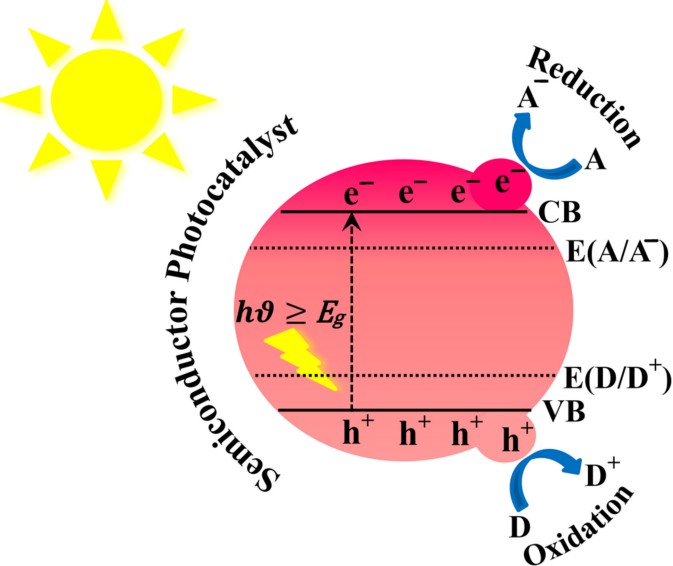

**Figure 8.** Schematic of the photocatalytic process mechanism.

In order to increase the rate of a photocatalytic reaction under UV-visible photoirradiation, it is necessary to select a semiconductor with a suitable band energy position, high light absorption ability, and retardation in the electron–hole recombination process.

For the case of hydrogen production from photoelectrochemical (PEC) water splitting, photogenerated e$^-$ and h$^+$ are separated as a stemming from the integrated electric field in a semiconductor. At the metal electrode surface and semiconductor photoanode surface, H$_2$O is reduced by photogenerated e$^-$ to produce H$_2$ and oxidized by h$^+$ to generate O$_2$, respectively [14].

In the context of photocatalytic water splitting, the VB and CB of the photocatalyst must be more positive than the O$_2$ production potential (0.82 V vs. NHE at pH = 7, Equation (4)) and more negative than the reduction potential of protons (−0.41 V vs. NHE,

at pH = 7, Equation (5)), respectively [55]. The oxidation and reduction reactions can be expressed below:

$$H_2O + 2\,h^+ \rightarrow \frac{1}{2}\,O_2 + 2\,H^+ + 2\,e^- \quad E = 0.82\ V\ vs.\ NHE \tag{4}$$

$$2H^+ + 2\,e^- \rightarrow H_2 \quad E = -0.41\ V\ vs.\ NHE \tag{5}$$

Further discussion with several examples on photocatalytic and photoelectrochemical water splitting on different semiconductors toward hydrogen generation will be presented in Section 4.

For the case of photocatalytic $CO_2$ reduction, the photoinduced $e^-$ and $h^+$ are transferred to an appropriate semiconductor surface or the cathode and the anode in the photocatalysis (PC) and PEC process, respectively [56]. The $e^-$ reduces $CO_2$ into clean fuels, and the $h^+$ oxidizes $H_2O$ into $O_2$ [57]. The proton reduction, especially in the presence of $H_2O$, always competes with the $CO_2$ reduction [56]. The first step of $CO_2^{\bullet-}$ intermediate generation is a well-known mechanism for $CO_2$ photoreduction that possesses a highly negative equilibrium potential ($-1.9$ V vs. NHE (Equation (6)) [56]. Moreover, there is another mechanism that includes successive proton-coupled electron transfer (PCET) reactions to overcome the high-energy $CO_2^{\bullet-}$ radical species generation (Equations (7)–(13)) [56]. The photocatalytic reduction of $CO_2$ to CO is a two-electron reaction that exhibits a comparatively low kinetic barrier in comparison to the other reactions in the complex organic compounds generation, such as $CO_2$ reduction to produce $CH_3OH$, $CH_4$, which are achieved through the transfer of six and eight electrons representing the high kinetic energy barrier, respectively [58,59].

Thermodynamically, the CB of a semiconductor must be more negative than the standard evolution potential of carbon-based fuel (e.g., CO, $CH_3OH$, $CH_4$, HCOOH, $C_2H_6$, $C_2H_5OH$, and even $H_2C_2O_4$ generated from $CO_2$ reduction reaction related on different reaction pathways) [57]. The evolution potential is $-0.24$ V vs. NHE at pH = 7 for $CH_4$ production (Equation (7)), and $-0.52$ V for CO generation as given in Equation (8), [55].

$$CO_2 + e^- \rightarrow CO_2^{\bullet-} \quad E = -1.90\ V\ vs.\ NHE,\ pH = 0 \tag{6}$$

$$CO_2 + 8\,H^+ + 8\,e^- \rightarrow CH_4 + 2H_2O \quad E = -0.24\ V\ vs.\ NHE,\ pH = 7 \tag{7}$$

$$CO_2 + 2\,H^+ + 2\,e^- \rightarrow CO + H_2O \quad E = -0.52\ V\ vs.\ NHE,\ pH = 7 \tag{8}$$

$$CO_2 + 6\,H^+ + 6\,e^- \rightarrow CH_3OH + H_2O \quad E = -0.39\ V\ vs.\ NHE,\ pH = 7 \tag{9}$$

$$CO_2 + 2\,H^+ + 2\,e^- \rightarrow HCOOH \quad E = -0.58\ V\ vs.\ NHE,\ pH = 7 \tag{10}$$

$$2\,CO_2 + 14\,H^+ + 14\,e^- \rightarrow C_2H_6 + 4H_2O \quad E = -0.27\ V\ vs.\ NHE,\ pH = 7 \tag{11}$$

$$2\,CO_2 + 12\,H^+ + 12\,e^- \rightarrow C_2H_5OH + 3H_2O \quad E = -0.33\ V\ vs.\ NHE,\ pH = 7 \tag{12}$$

$$4\,CO_2 + 4\,H^+ + 4\,e^- \rightarrow 2H_2C_2O_4 \quad E = -0.87\ V\ vs.\ NHE,\ pH = 7 \tag{13}$$

Further discussion with several examples of photocatalytic $CO_2$ reduction on different semiconductors toward hydrogen generation will be presented in Section 5.

For the case of pollutant degradation, the photoinduced $e^-$ and $h^+$ with enough energy can migrate to the photocatalyst surface, and produce ROS (such as $^{\bullet}OOH$, $^{\bullet}OH$, $h^+$, $H_2O_2$, and $^{\bullet}O_2^-$) via participating in the redox reaction as determined in Equation (14) to Equation (18). In detail, photoinduced $h^+$ can directly react with pollutant molecules

and produce $CO_2$, and $H_2O$ and other harmless products (Equation (14)) or react with $H_2O$ to give hydroxyl radicals ($^\bullet OH$) (Equation (15)). On the other side, photoinduced $e^-$ can react with $O_2$ molecules or $H^+$ ions to create superoxide radical anion ($^\bullet O_2{}^-$) and hydrogen peroxide ($H_2O_2$) (Equations (16) and (17)). Moreover, $^\bullet OH$ radicals can produce via a reaction between $e^-$, and $H_2O_2$ (Equation (18)). Furthermore, $^\bullet OOH$ can generate via a reaction between $^\bullet O_2{}^-$ and $H^+$ (Equation (19)). Finally, pollutant molecules can decompose into $CO_2$, $H_2O$, and harmless products via the reaction between reactive oxygen species (h$^+$, $^\bullet OH$, $^\bullet O_2{}^-$, $H_2O_2$, $^\bullet OOH$) after contact with the surface of the photocatalyst (Equation (20)) [60–63].

$$\text{h}^+ + \text{pollutant molecules} \rightarrow \text{harmless products} + CO_2 + H_2O \tag{14}$$

$$\text{h}^+ + H_2O \rightarrow {}^\bullet OH + H^+ \tag{15}$$

$$\text{e}^- + O_2 \rightarrow {}^\bullet O_2{}^- \tag{16}$$

$$\text{e}^- + (O_2, 2H^+) \rightarrow H_2O_2 \tag{17}$$

$$\text{e}^- + H_2O_2 \rightarrow 2{}^\bullet OH \tag{18}$$

$${}^\bullet O_2{}^- + H^+ \rightarrow {}^\bullet OOH \tag{19}$$

$$\text{pollutant molecules} + (\text{h}^+, {}^\bullet OH, {}^\bullet O_2{}^-, H_2O_2, {}^\bullet OOH) \rightarrow CO_2 + H_2O + \text{harmless products} \tag{20}$$

However, recombination of photoinduced $e^-$ and $h^+$ may occur by releasing heat and photons, which cause a reduction in photocatalytic quantum yield. In this regard, the probability of the recombination can be significantly reduced by modifying the structure of the semiconductor via different strategies such as the creation of the surface defects, metal or nonmetal doping, regulation of morphology, and construction of heterojunctions.

Further discussion with several examples of photocatalytic dye and drug degradation on different semiconductors toward hydrogen generation will be presented in Section 6.

### 2.3. Band Gap Determination

One of the most critical parameters in predicting the photocatalytic performance of a semiconductor is the determination of the band gap energy ($E_g$). For this purpose, UV-Vis spectrophotometry and diffuse reflectance spectroscopy (DRS), are commonly employed by analyzing their absorption spectrum. The optical property of a semiconductor can be given as follow:

$$R(\lambda) + T(\lambda) + E(\lambda) = 1, \; E(\lambda) = A(\lambda) + S(\lambda) \tag{21}$$

where $R(\lambda), T(\lambda), E(\lambda), A(\lambda),$ and $S(\lambda)$ represent the reflection, transmission, extinction, absorption, and scattering coefficient at a specific wavelength $\lambda$ [50]. The band gap energy of a semiconductor can be calculated by using the Tauc plot method and Kubelka–Munk theory, as presented in the following sections.

In 1966, Tauc presented a technique for calculating the band gap energy of a semiconductor by applying UV-visible spectroscopy [64]. According to this method, the correlation between band gap energy ($E_g$), and the optical absorption coefficient ($\alpha$) can be expressed as follow:

$$(\alpha h\nu) = A\left(h\nu - E_g\right)^n \tag{22}$$

where $\alpha$ is optical absorption coefficient, hν is incident photon energy, A is constant, $E_g$ is band gap energy, and n is a parameter that shows the type of electron transition and can have different values of 1.2, 2, 3, and 3.2, which are related to direct allowed, indirect allowed, indirect forbidden, and direct forbidden transitions, respectively. The quantity of optical absorption coefficient ($\alpha$) is estimated using the following equation:

$$\alpha = \frac{1}{d}\ln\left(\frac{1}{T}\right) \tag{23}$$

where $d$ and $T$ are the thickness and transmittance of the photocatalyst, respectively [50].

For opaque materials with less photoabsorption property, such as powder materials and rough surfaces, the Kubelka–Munk theory can be applied. It was initially introduced in 1931 [65] for determining absorption coefficient by using the DRS technique. This function relies on the reflection of light from the surface of an opaque sample (R). The Kubelka–Munk (K–M) function is calculated using the following equation:

$$\frac{\left(1 - R_d^2\right)}{2R_d} = \frac{\alpha}{S} \tag{24}$$

where $R_d$ is diffuse reflectance, $\alpha$ is the absorption coefficient, and S is the scattering coefficient [66]. After obtaining $\alpha$ for the above equation, the band gap of a semiconductor photocatalyst can be determined by using Equation (22).

*2.4. Efficiency Evaluation in a Photocatalytic Process*

Considering photocatalytic water splitting, a standard parameter for investigating the material efficiency of this reaction is a quantity called solar-to-hydrogen (STH) conversion efficiency as described below [6]:

$$\text{STH}(\%) = \frac{\text{output energy of H}_2\text{ production}}{\text{energy of incident solar light}} \times 100 \tag{25}$$

For the case of photoelectrochemical water splitting, an essential parameter to evaluate photoelectrode performance is a quantity called incident photon-to-current conversion efficiency (IPCE). The IPCE is assigned as measuring the amount of the collected photogenerated electrons at the back contact of the photoanode per incident photon as a function of the illuminating light wavelength. By using a standard three-electrode configuration, the value of IPCE can be obtained at a particular light wavelength ($\lambda$) by using the following equation:

$$\text{IPCE}(\%) = \left(\frac{1240 \times I}{\lambda \times P(\lambda)}\right) \times 100 \tag{26}$$

where $P(\lambda)$ is the light intensity ($\text{mW/cm}^2$) at a specific wavelength $\lambda$, and $I$ is the photocurrent density ($\text{mA/cm}^2$) [14,67,68].

Absorbed photon-to-current conversion efficiency (APCE) is another critical measurement that is an internal quantum efficiency representing optical quantity related to the reflection losses. The value of APCE can be calculated according to the equation described below [14,67]:

$$\text{APCE}(\%) = \frac{\text{IPCE}(\%)}{A(\lambda)} = \frac{\text{IPCE}(\%)}{1 - R(\lambda) - T(\lambda)} \tag{27}$$

where $R(\lambda)$, $A(\lambda)$, and $T(\lambda)$, are reflectance, absorption, and transmittance at the defined wavelength ($\lambda$), respectively. The higher amount of APCE shows the better catalytic activity of a photoelectrode.

For the case of photocatalytic degradation of pollutants, the commonly employed techniques to estimate the activity of a photocatalytic process are apparent quantum yield (AQE), conversion, the total organic compound (TOC), and selectivity. The calculation of AQE is followed by evaluating the ratio of the total number of reacted electrons to the total number of incident photons at a specific wavelength as described below [69]:

$$\text{AQE}(\%) = \frac{\text{Number of total reacted electrons}\left(\text{mol} \cdot \text{s}^{-1}\right)}{\text{Number of incident photons}\left(\text{einstein} \cdot \text{s}^{-1}\right)} \times 100 \tag{28}$$

For the case of multi-electron processes such as hydrogen generation, the number of electrons (n) must be considered in the equation as given below:

$$\text{AQE}(\%) = \frac{2 \times \text{Number of total reacted H}_2 \text{ molecules}\left(\text{mol} \cdot \text{s}^{-1}\right)}{\text{Number of incident photons}\left(\text{einstein} \cdot \text{s}^{-1}\right)} \times 100 \tag{29}$$

The quantification of the organic pollutant's degradation yield is commonly conducted by examining the pollutant concentration changes, which is known as conversion efficiency, as shown below:

$$\text{conversion}(\%) = \left(\frac{(C_0 - C)}{C_0}\right) \times 100 = \left(1 - \frac{C}{C_0}\right) \times 100 \tag{30}$$

where C (mol·L$^{-1}$) and C$_0$ (mol·L$^{-1}$) are the concentration level of an organic contaminant at time t and 0, respectively. Moreover, the total organic compound (TOC) is usually examined to measure photocatalytic activity as follows:

$$\text{TOC}(\%) = \left(\frac{[\text{TOC}]}{[\text{TOC}]_0}\right) \times 100 \tag{31}$$

where [TOC] and [TOC]$_0$ are the total organic compound at t and t = 0 during photocatalytic reaction [50]. The other two standard parameters to determine the efficiency in the photocatalytic processes are selectivity and yield, as defined below:

$$\text{selectivity}(\%) = \left[C_p / (C_0 - C_r)\right] \times 100 \tag{32}$$

$$\text{Yield}(\%) = \left(\frac{C_p}{C_0}\right) \times 100 \tag{33}$$

where C$_0$ represents the initial concentration of the reactant, C$_r$ denotes the concentration of the reactant at any time t, and C$_p$ is the concentration of the product after the reaction occurs in time t [69].

## 3. Factors Affecting the Photocatalytic Activity

In order to optimize the activity of a photocatalyst, several modification strategies can be applied to a photocatalyst. These are including morphology regulation, co-catalysts decoration, doping, defect engineering, surface sensitization, and heterojunction construction. In addition to the modification approach of a photocatalyst, parameters related to reaction conditions must be considered. These parameters are including photocatalyst concentration, initial contaminant concentration, pH, reaction temperature, light intensity, and charge-carrier scavengers (Figure 9) [53,62,70].

Therefore, it is required to optimize and control all of these factors to achieve high catalytic activity of a semiconductor photocatalyst by examining different conditions. Several modification strategies will be discussed in the following sections.

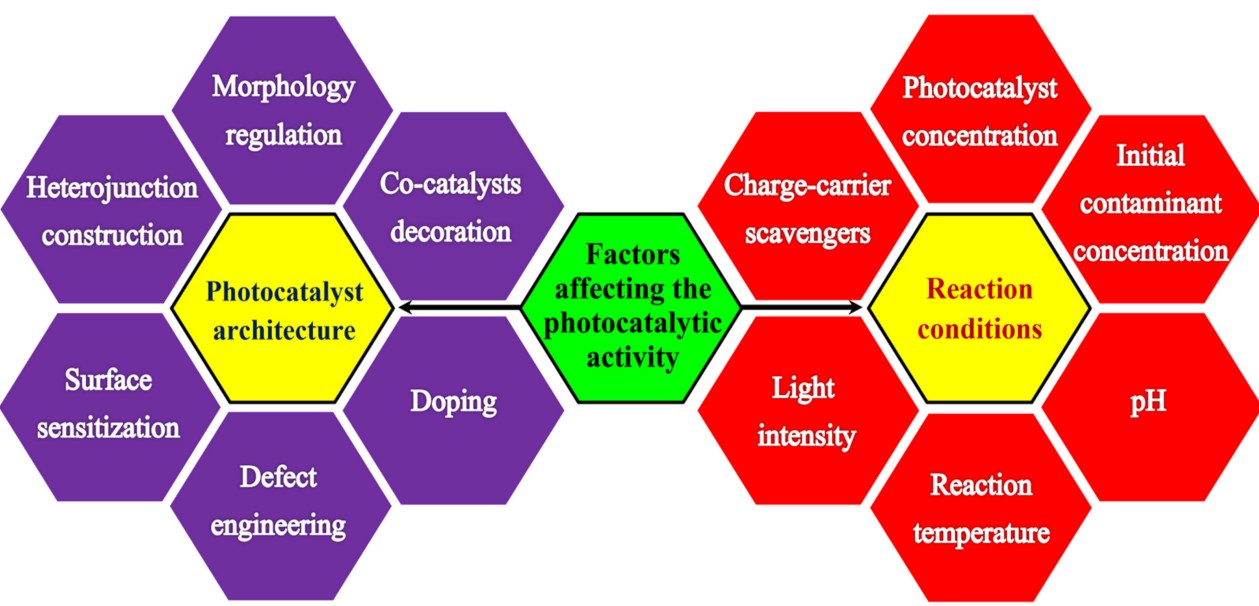

**Figure 9.** The most critical factors affecting activity of a photocatalytic reaction.

### *3.1. Photocatalyst Structural Modification Strategies*

#### 3.1.1. Morphology Regulation

Semiconductor morphology has a vital role in its photocatalytic performance. Based on their dimensions, semiconductor materials can be categorized as 0-dimensional (0D), 1-dimensional (1D), 2-dimensional (2D), and 3-dimensional (3D), all of which exhibit superior photocatalytic activity due to their intrinsic structural merits [71,72]. 0D semiconductors (e.g., nanoparticles and quantum dots) have the advantages of excellent up-conversion properties, strong light–matter interactions, good reduction ability, and tunable optical properties [73,74]. Benefiting from their unique advantages, several 0D materials such as carbon [75], black phosphorous [76], $Cu_2O$ [77], $Bi_2S_3$ [78], CdS [79], ZnS [80], and $Cu_5FeS_4$ quantum dots [81], Au nanoparticles [82], have been applied in designing highly efficient photocatalysts toward clean energy production and environmental remediation.

One-dimensional semiconductors (e.g., nanotubes, nanorods, nanowires, nanofibers) with noticeable advantages, such as rapid charge transfer along the 1D axis, excellent mechanical properties, high length-to-diameter ratio, and extensible surface functionality, have been widely used in the photocatalytic applications [72,83]. Moshfegh's research group fabricated different 1D ZnO-based photocatalysts to design appropriate morphology for efficient photocatalytic reactions toward application in environmental remediation [84–90].

Two-dimensional semiconductors have remarkable merits, such as large specific surface area, low recombination rate, and tunable electronic properties. They can be applied as good support for developing novel photocatalysts by structural engineering [91,92]. Several 2D semiconductors such as $g-C_3N_4$ [93], LDH [94], BiOX (X = Cl, Br, I) [95], hexagonal boron nitride [96], MXene [97], $WS_2$ [98], $WO_3$ [99], $ReS_2$ [100], $MOS_2$ [101] nanosheets, and reduced graphene oxide (rGO) [102] have been used to develop beneficial photocatalysts for photocatalytic degradation of pollutants and clean energy production.

Finally, 3D semiconductors with a network structure and outstanding properties such as excellent charge carrier mobility owing to interconnecting structures, high stability, large specific surface area, and superior mass transfer performance have gained tremendous attention [103–105]. During recent years, several 3D nanomaterials such as fullerene $C_{60}$ [106], metal–organic frameworks (MOFs) [107], covalent organic frameworks (COFs) [108], and three-dimensional graphene aerogel (3D GA) [38] have been employed as the fascinating materials for photocatalytic applications.

### 3.1.2. Co-Catalysts Decoration

Decoration of the semiconductors with metals such as Ni [109], Rh [110], Au [111], Ag [112], pd [113], and pt [114], has been known as a powerful strategy to improve photocatalytic performance. Promoting the charge carriers separation via trapping the photogenerated electrons by loaded metals, enhancing electron transportation, increasing the substrate adsorption, creating more surface active sites, and preventing the occurrence of any side reaction are the main reasons for improving the photocatalytic performance by developing the co-catalyst structures [60,115]. For the case of water splitting, Ortega et al. [116] elucidated the impact of loading Au nanoparticles as a co-catalyst with surface plasmon resonance (SPR) ability on $ZrO_2$–$TiO_2$ in the photocatalytic $H_2$ production. The presence of Au nanoparticles with optimum content of 5 wt.%, increased $H_2$ production four times greater than that of $ZrO_2$–$TiO_2$ as shown in Figure 10a. For the environmental issue, Gu et al. [117] reported that the introduction of AgNPs in 4,4,4,4-(porphyrin-5,10,15,20-tetrayl) tetrakis (benzoic acid) (TCPP) ligand into UiO-66-$NH_2$Zr-TCPP (AZT5) (Zr-TCPP) results in a remarkable enhancement in the photodegradation efficiency of Cr(VI). They have found rapid charge separation, and strong visible light absorption increased the reaction rate constants of the AgNPs-modified semiconductors to about 3.6–5.4 times compared to that of Zr-TCPP (0.075~0.114 $min^{-1}$) as shown in Figure 10b. Therefore, the results obtained from these types of studies confirm the participation in co-catalyst decoration can boost the activity of a photocatalytic reaction.

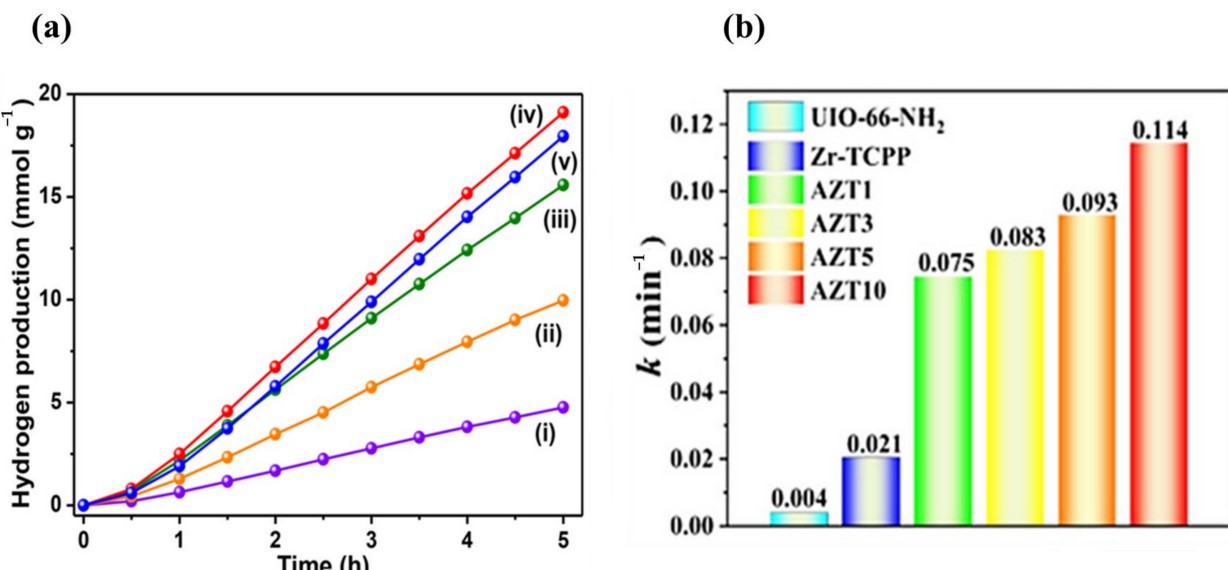

**Figure 10.** (**a**) $H_2$ production in the presence of various photocatalysts ((i) $ZrO_2$–$TiO_2$, (ii) 1 wt.% Au/$ZrO_2$–$TiO_2$, (iii) 3 wt.% Au/$ZrO_2$–$TiO_2$, (iv) 5 wt.% Au/$ZrO_2$–$TiO_2$, and (v) 7 wt.% Au/$ZrO_2$–$TiO_2$). Reproduced with permission from ref. [116]; (**b**) the first-order kinetic constants of Cr (VI) reduction by various photocatalysts. Reproduced with permission from ref. [117].

### 3.1.3. Doping

Tuning the electronic structure of photocatalysts and thus decreasing their kinetic energy barrier can be achieved by the elemental doping of semiconductors [118]. Several metallic cations such as $Cu^{2+}$ [119], $Bi^{3+}$ [120], $Co^{2+}$ [121], $Fe^{3+}$ [122], $La^{3+}$ [123], and non-metallic anions such as N [124], S [125], P [126], C [127], B [128], and Se [129] have been used as dopants for semiconductors and have led to a significant improvement in their photocatalytic activities. Moshfegh's research group has successfully developed efficient photocatalysts for environmental remediation and clean energy production by introducing different types of dopants such as S in $TiO_2$ nano-porous films [130], multi-walled carbon nanotubes (MWCNT) in ZnO electrospun nanofibers [131], Ce in ZnO nanocomposite thin

films [132], Ru in TiO$_2$ nanotubes [133], and P in g-C$_3$N$_4$ nanosheet [134]. In findings from their latest study, it is indicated that incorporation of P into the g-C$_3$N$_4$ nanosheets structure (PCNS) enhances the light absorption ability of the semiconductor due to the shifting band gap from 2.7 eV to ∼2.3 eV in comparison with g-C$_3$N$_4$ nanosheets, leading to a higher H$_2$ production via photoelectrochemical water splitting. Their results show that the mid-gap state formation in the PCNS played an essential role in maximizing the photocurrent density. They obtained the highest photocurrent density (0.5 mAs/cm$^2$ at 1.23 V vs. reversible hydrogen electrode (RHE)) for the BiVO$_4$-PCNS (BV-PCNS) photoanode as shown in Figure 11a [134]. In the case of environmental remediation, Chen et al. [135] reported that doping Bi-based MOF (CAU-17) with S element enhanced photocatalytic removal of tetracycline compared to pure CAU-17. Based on data analysis obtained from their energy diagram, band gap energy narrows from 3.57 to 3.37 eV after optimizing the S content (Figure 11b), resulting in better usage of light energy and thus improving the photocatalytic activity. The results demonstrate a remarkable enhancement in the photocatalytic performance via the doping strategy.

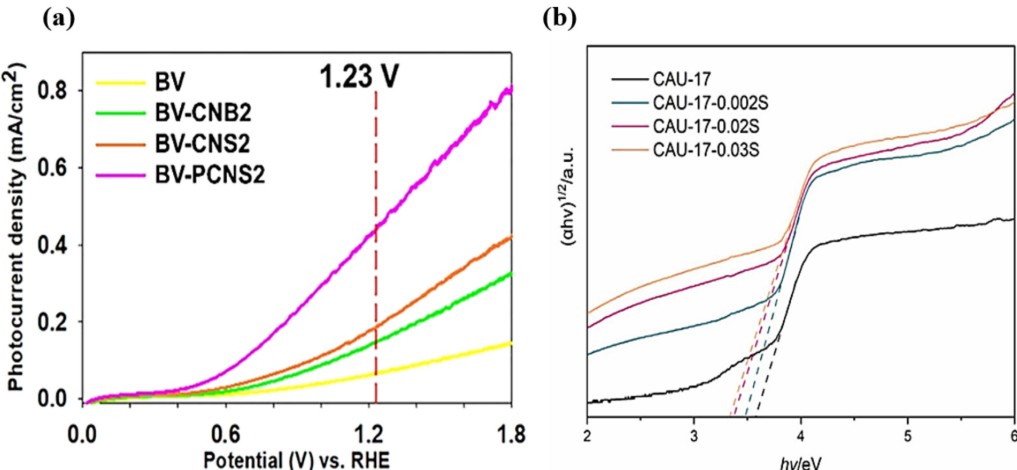

**Figure 11.** (**a**) LSV plots light conditions photoelectrochemical current density at 1 V vs. RHE. Reproduced with permission from ref. [134]; (**b**) band gap energy of CAU-17 before and after vulcanization. Reproduced with permission from ref. [135].

### 3.1.4. Defect Engineering

Defect engineering has been proven to be another promising strategy for achieving highly efficient photocatalysts. Among various types of defects such as 0D point, 1D line, 2D planer, and 3D volume defects, 0D point defects, including vacancies, can be quickly introduced on the surface of the semiconductors to optimize their optical and electronic properties, thereby improving their photocatalytic activity [136,137]. Effective carrier separation and transportation by reducing the band gap, improved light absorption, and enhanced reactive radicals formation can achieve by surface defects that act as active sites [138,139]. In comparison with several types of vacancies, including anion vacancies (e.g., oxygen [140], sulfur [141], carbon [142], and nitrogen [143] vacancies) and cation vacancies (e.g., titanium [144], zinc [145], and molybdenum [146] vacancies), anion vacancies, especially oxygen vacancies, have been widely studied because of their low formation energy [147]. In the case of environmental remediation, Moshfegh's research group [84] examined the impact of carbon doping and oxygen vacancy on the Methylene blue (MB) photodegradation in aqueous media. The photocatalytic performance pathway of ZnO-carbon is shown in Figure 12. The oxygen vacancy creates a new energy state under CB of the ZnO, which decreases the band gap in the ZnO-carbon and retards photogenerated carrier recombination by capturing photo-excited electrons.

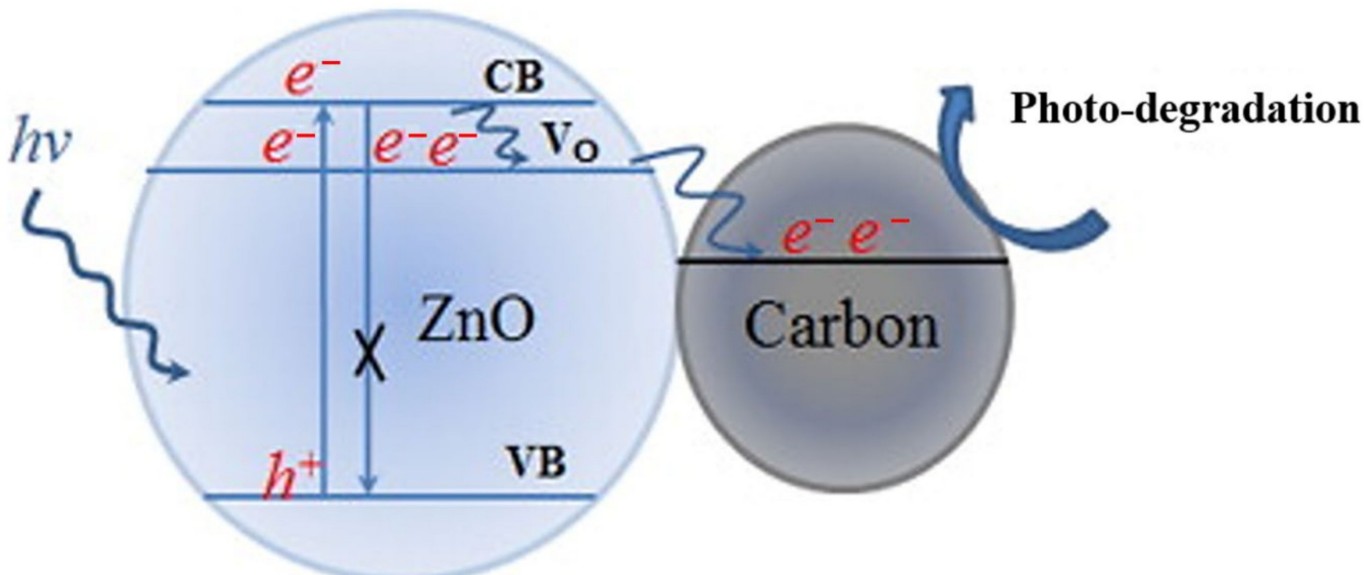

**Figure 12.** Band energy diagram and photocatalytic activity pathway of the ZnO-carbon under the UV light. Reproduced with permission from ref. [84].

### 3.1.5. Surface Sensitization

The capacity of harvesting visible light by photocatalysts can be increased by employing photosensitizers such as organic dyes and metal sulfide nanoparticles on the surface of the semiconductors [148]. In the case of clean energy production, Moshfegh's research group [149] used $Ag_2S$ nanoparticles for photoelectrochemical activation of $TiO_2$ nanotube arrays (TNA) through the dual-step anodization process of titanium sheets. $Ag_2S$ nanoparticles were deposited on the TNA by applying a sequential-chemical bath deposition (S-CBD) technique, as shown in Figure 13a. The highest photocurrent of 840 $\mu$A cm$^{-2}$ (with a power density of 25 W m$^{-2}$), exhibiting an approximately 15-fold increase compared to that of pure TNA under visible light, and the IPCE efficiency of nearly 20% at a wavelength of 600 nm was obtained in the case of the TNA/$Ag_2S$ photoanode that proved the positive effect of the surface sensitization strategy. In another study reported by this group [150], CdS nanoparticle sensitized titanium dioxide decorated graphene CdS/$TiO_2$/graphene (CTG(n)) fabricated as a practical photoanode for $H_2$ generation via water splitting. The CdS nanoparticle serving as the visible light harvesting played an essential role in achieving the highest photocurrent density of 4.5 A/m$^2$ and longest electron lifetime. Dinda et al. [151] fabricated 2,3-diaminophenazine (Phz) incorporated into NS-co-doped GQD (Phz-NS_GQD) photocatalyst to produce $H_2$ from water. Because of a planar rigid structure with $\pi$-conjugation in phenazine, the photogenerated electrons rapidly transferred from LUMO of phenazine to CB of GQD, resulting in a sustained system (for over two days) with a maximum apparent quantum yield of 49% for $H_2$ production. The energy profile for photocatalytic HER of Phz-NS_GQD has been shown in Figure 13b. In the case of environmental issues, Wei et al. [152] reported that dye-sensitized $Bi_2MoO_6$ by using rhodamine B as the sensitizer can remove ~98.3% of levofloxacin after 2h LED irradiation which is much greater than the ~43.5% degradation rate of the $Bi_2MoO_6$, as shown in Figure 13c. In a cyclic pathway, the oxidized dyes are reduced to ground state by electrons located on the oxygen vacancies of $Bi_2MoO_6$ followed by re-exited and injected into the CB of the $Bi_2MoO_6$ to remove levofloxacin with enhanced efficiency.

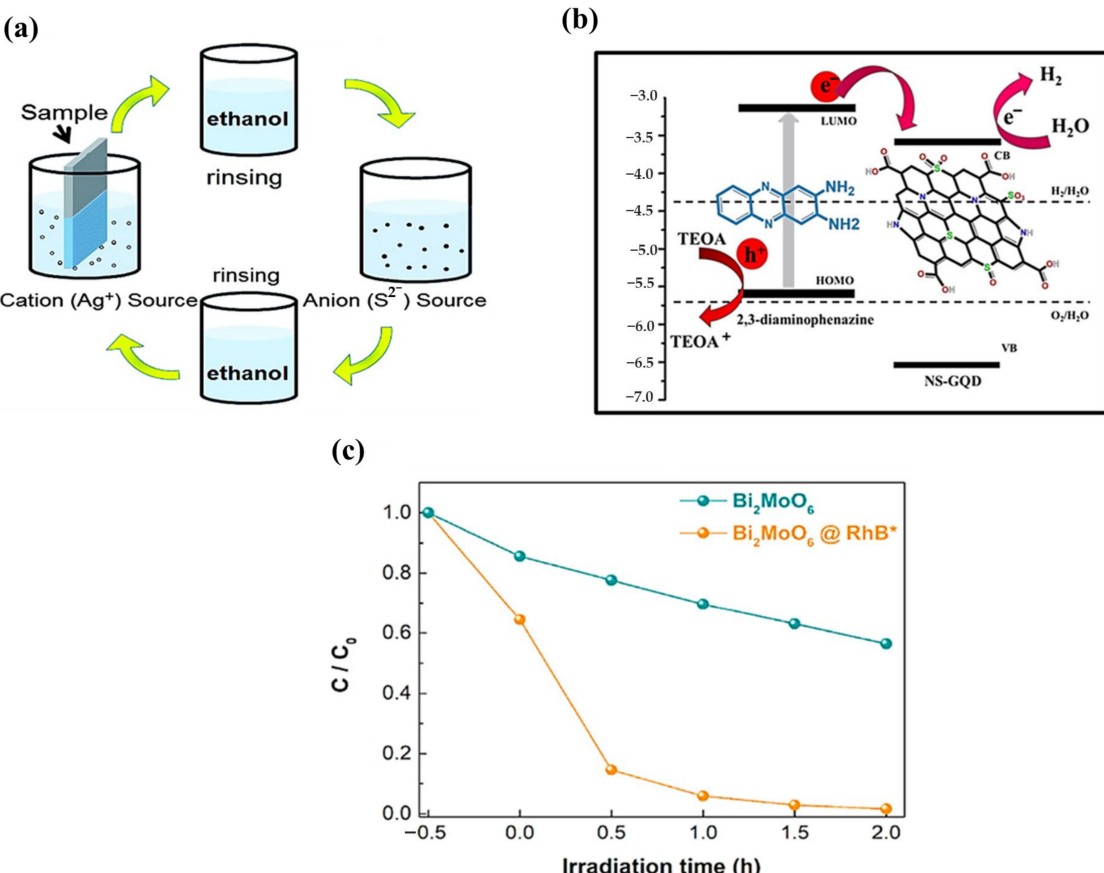

**Figure 13.** (**a**) A schematic of S-CBD process for depositing $Ag_2S$ nanoparticles. Reproduced with permission from ref. [149]; (**b**) energy diagram for photocatalytic HER of Phz-NS_GQD. Reproduced with permission from ref. [151]; (**c**) photodegradation rate of levofloxacin by $Bi_2MoO_6$ and $Bi_2MoO_6$@RhB under LED light irradiation. Reproduced with permission from ref. [152].

3.1.6. Heterojunction Construction

The combination of two or more semiconductors with different bandgaps is termed heterojunctions. Due to its notable advantages, including efficient charge separation, excellent light absorption, and low charge recombination, heterojunction construction has become a promising strategy for improving photocatalytic performance [153,154]. Different hetero-type junctions known as conventional, Z-scheme, and dual Z-scheme heterojunctions have been widely studied for designing high-performance photocatalysts in recent years [155]. The details of the charge carrier transformation in each heterojunction will be described in the following sections.

Conventional Heterojunctions

Based on the band edge alignment, conventional heterojunctions are classified into three types: (a) Type-I (straddling gap), (b) type-II (staggered gap), and (C) type-III (Broken gap) [156], as illustrated in Figure 14. In a type I heterojunction, the VB and CB of semiconductor I possess higher positive and negative potentials than the VB and CB of semiconductor II, respectively (Figure 14a). Consequently, the holes and electrons transfer from VB and CB of semiconductor I to VB and CB of semiconductor II constitute a smaller band gap than that of semiconductor I. Gathering electrons and holes in the semiconductor with smaller band gaps results in a higher recombination rate and lower redox potential for this system [157,158]. In a type II heterojunction, the VB of semiconductor II is higher positive than that of semiconductor I, and the CB of semiconductor I is higher negative than that of semiconductor II (Figure 14b). Consequently, the electrons and holes migrate to CB of semiconductor II and VB of semiconductor I, respectively; hence an efficient separation of

electron–hole pairs occur. However, the redox efficiency of both semiconductors decreases because both oxidation and reduction reactions take place on the VB and CB with weak redox potential [156]. For a type III heterojunction, as shown in Figure 14c, there is no overlap between the band gaps of two semiconductors, which restricts the charge carriers' separation and transmission [159].

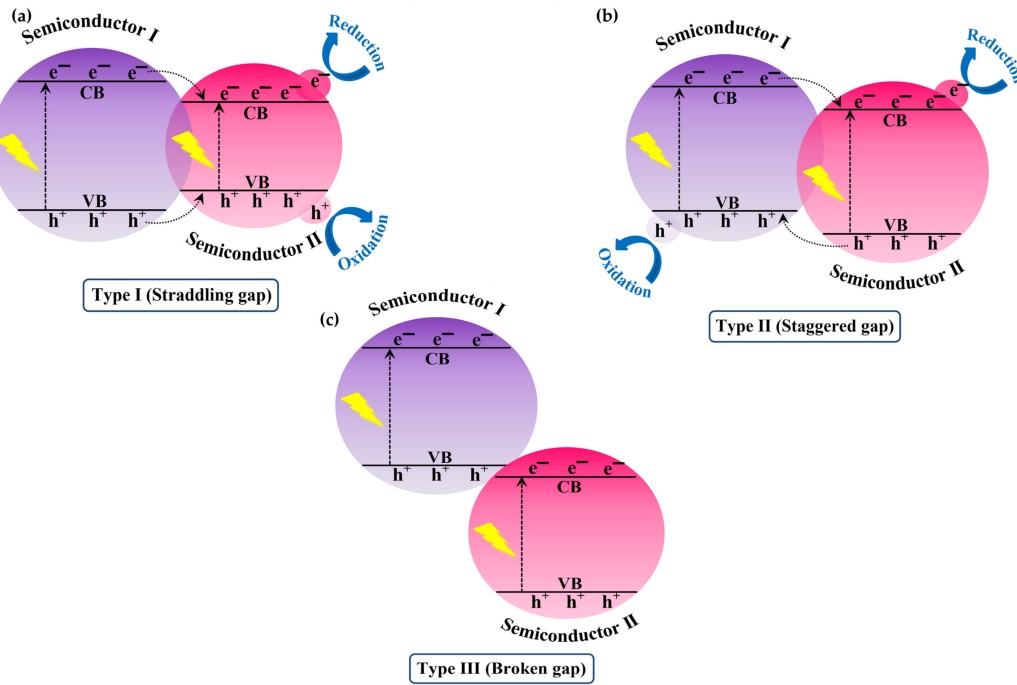

**Figure 14.** Schematic illustration of charge carrier transfer mechanism of the three conventional heterojunctions: (**a**) type-I (straddling gap); (**b**) type-II (staggered gap); (**c**) type-III (Broken gap).

Among conventional heterojunctions, type-II heterojunction is the most suitable structure for designing high-performance photocatalysts owing to its improved charge separation and transmission. Ahmad et al. fabricated [160] AgI/CdS binary composite via an in situ precipitation method and examined its performance for photodegradation of methyl orange (MO) and tetracycline hydrochloride (TCH) under visible light illumination. The as-prepared photocatalyst showed high efficiency of 94.5 and 91% for MO and TCH degradation, respectively. Based on their results obtained from the active species trapping experiment, a mechanism of type-II heterojunction formation is suggested. As shown in Figure 15a, the addition of isopropyl alcohol (IPA), as $^{\bullet}$OH scavenger, did not show a significant effect on the photocatalytic activity of MO degradation. Whereas, adding benzoquinone (BQ) to the reaction mixture dramatically reduced the MO photodegradation from 94.5 to 23%, indicating that $^{\bullet}O_2^{-}$ as an active reactive species played an essential role in the reaction mixture. Moreover, the presence of EDTA-2Na resulted in a moderate reduction of 60% in MO degradation, indicating that $h^{+}$ synergistically participated in the photodegradation process but to a lesser degree compared to $^{\bullet}O_2^{-}$. The schematic of the charge carrier transfer mechanism of the photocatalyst shows in Figure 15b. As shown in the Figure, the VB of AgI is higher positive than that of CdS (2.33 vs. 1.65), and the CB of CdS is higher negative than that of AgI (−0.73 vs. −0.47). Consequently, the electrons migrate to CB of AgI and react with $O_2$ to generate $^{\bullet}O_2^{-}$. Subsequently, the generated $^{\bullet}O_2^{-}$ species decomposed the organic pollutants. Simultaneously, the holes migrate to the VB of CdS and directly react with pollutants. Due to the relatively lower redox potential of CdS (+1.65) compared to the $^{\bullet}OH/OH^{-}$ (+2.38 eV vs. NHE) and the $H_2O/^{\bullet}OH$ (+2.72 eV vs. NHE), the generated holes are incapable of oxidizing $OH^{-}$ and $H_2O$ to form $^{\bullet}OH$.

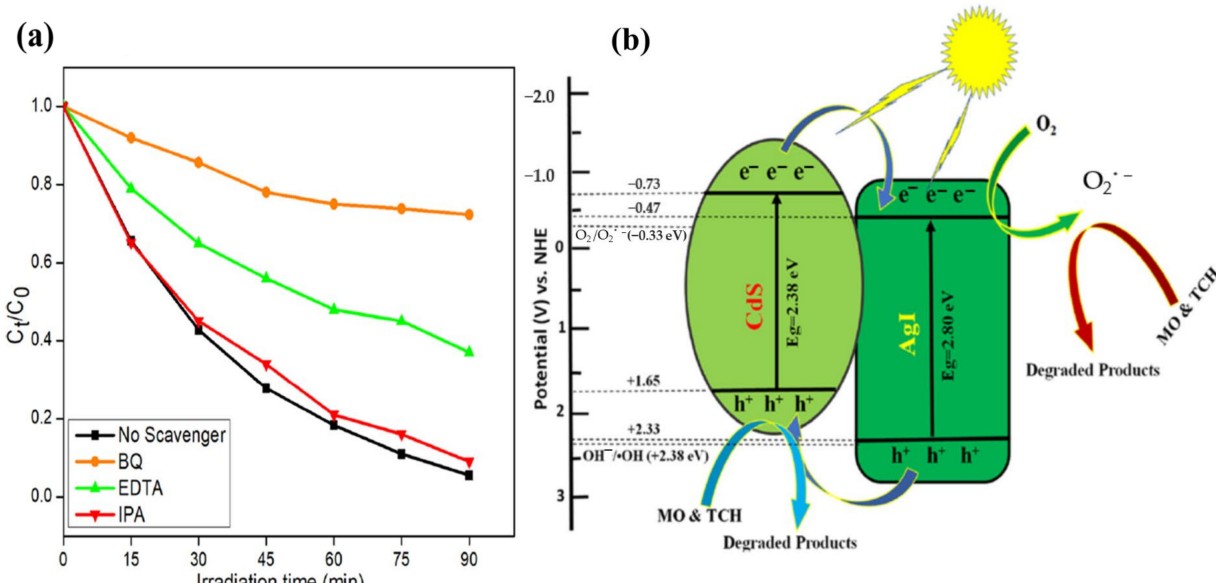

**Figure 15.** (**a**) Change in the MO concentrations in the absence and presence of various scavengers over AgI/CdS photocatalyst; (**b**) Schematic illustration of charge carrier transfer mechanism for MO and TCH photodegradation over the AgI/CdS photocatalyst. Reproduced with permission from ref. [160].

Z-Scheme Heterojunctions

In order to overcome the drawbacks of conventional heterojunctions, Z-scheme heterojunctions have been developed. During light illumination, a Z-shape carrier transport pathway is formed in these structures, which results in excellent redox abilities and increased light harvesting for the photocatalytic system [161]. Based on the different transport media, three categories of Z-scheme heterojunctions exist: traditional Z-scheme, all-solid-state Z-scheme, and direct Z-scheme [162], as depicted in Figure 16. In a traditional Z-scheme, introduced by Bard in 1979 [163], a pair of the acceptor (A)/donor(D), which is called shuttle redox mediator (including $Fe^{3+}/Fe^{2+}$, $[Co(phen)_3]^{3+/2+}$, $IO^{3-}/I^-$, $NO^{3-}/NO^{2-}$ and $[Co(bpy)_3]^{3+/2+})$ is incorporated between two semiconductors for improving charge carrier transportation [164]. As shown in Figure 16a, after illumination, both semiconductors undergo excitation to generate electrons and holes. Afterward, electrons with weak reduction potential generated in the CB of semiconductor I react with A and then reduce to D. Similarly, the holes with weak oxidation potential induced in the VB of semiconductor II react with D and then oxidize to A. Hence, the electrons and holes with excellent reduction and oxidation potentials remain separately in the CB and VB of semiconductor II and I, respectively. Despite its high redox ability, the practical application of the traditional Z-scheme is restricted because of the necessity of an aqueous solution and the light shielding effect produced by redox mediators. In addition, the possibility of backward reactions resulting from the lack of spatial restriction between shuttle redox mediators and reaction sites is high in this system. To address these drawbacks, an all-solid-state Z-scheme, fabricated by Tada et al. in 2006 [165], has been developed by replacing conductors (such as metals, carbon-based materials, and organic polymers) with redox pairs between two semiconductors (Figure 16b). Because of the high transfer ability of conductive materials, electrons with weak reduction potential generated in the CB of semiconductor I transfer to the semiconductor II through the conductor and recombine with holes with weak oxidation potential induced in the VB of semiconductor II, thereby the electrons and holes with higher reduction and oxidation ability are separately preserved. However, the transmission of the electrons to the photocatalyst surface through a conductor in a case of incomplete conductor placement between two semiconductors and the use of high-cost and photo-corrosive noble metals as a mediator are the main drawbacks of this system [161,162,166].

To overcome all the problems associated with using a mediator, a direct Z-scheme was introduced by Yu et al. in 2013 [167]. In a direct Z-scheme, as illustrated in Figure 16c, two semiconductors with different work functions are in contact with each other without a charge transfer mediator. Resulting of the difference in the work functions between the two semiconductors, electrons transfer from semiconductor II with a higher Fermi level to semiconductor I with a lower Fermi level energy until the Fermi levels are aligned. As a result, the positive and negative charges are formed at the interface of semiconductor I and II, respectively. Followed by, an internal electric field is formed between semiconductors, whereby electrons and holes with weak reduction and oxidation potential induced in the CB of semiconductor I and VB of semiconductor II are recombined, and the carriers with higher redox potential are separately preserved to participate in the reaction [159,168].

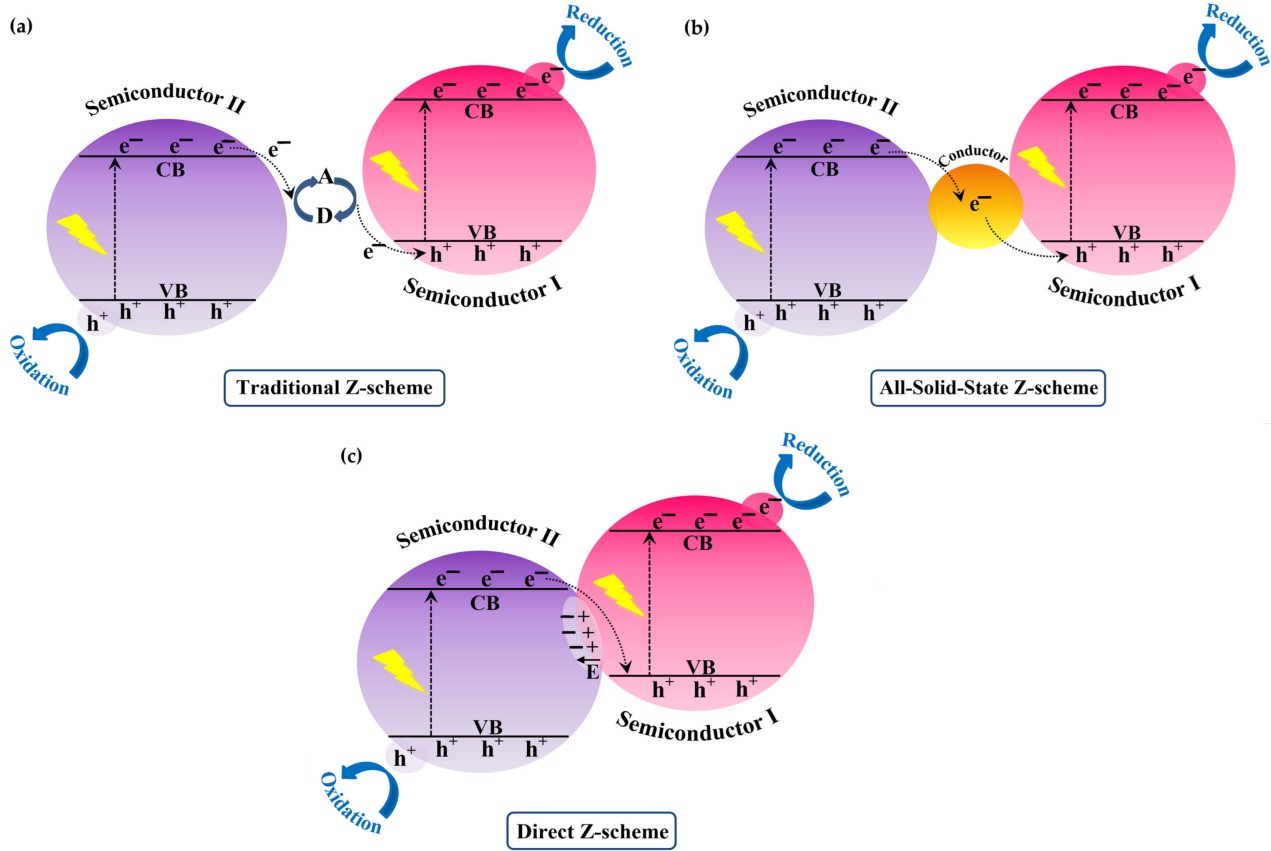

**Figure 16.** Schematic illustration of charge carrier transfer mechanism of the Z-scheme heterojunctions: (**a**) traditional Z-scheme; (**b**) all-solid-state Z-scheme; (**c**) direct Z-scheme.

Among the three types of Z-scheme heterojunctions, direct Z-scheme promises the most significant advantages. Eliminating the use of expensive noble metal as a mediator in the direct Z-scheme dramatically reduce the cost of the photocatalytic system compared to the all-solid-state Z-scheme. In addition, the absence of the redox mediator, differing from the traditional Z-scheme, avoids the backward reactions and reduces the shielding effect [168,169]. In the case of clean energy production, Wang et al. [170], synthesized $Bi_2WO_6/TiO_2$ photocatalyst through the solvothermal method and exhibited a remarkable efficiency of 12.9 mmol·$g^{-1}h^{-1}$ in $H_2$ production. Based on the radical trapping experiment in the presence of 5, 5-dimethyl-1-pyrroline-N-oxide (DMPO) as an •OH scavenger, four distinct peaks corresponding to •OH have been observed (Figure 17a). As depicted in Figure 17b, they showed that if the charge transfer mechanism is type II, the photogenerated electrons on the CB of $TiO_2$ transfer to the CB of $Bi_2WO_6$, and the photogenerated holes on the VB of $Bi_2WO_6$ transfer to VB of $TiO_2$ due to the more positive potential of the VB

and CB of $Bi_2WO_6$ compared to those of $TiO_2$. However, The $E_{CB}$ of $Bi_2WO_6$ (0.36 eV) exhibited a higher positive potential than the standard redox potential of $H^+/H_2$ (0 eV vs. NHE), making it impossible for electrons to reduce $H^+$ to $H_2$. Similarly, the $E_{VB}$ of $TiO_2$ (2.75 eV) is nearly equivalent to the redox potential of $H_2O/^\bullet OH$ (2.72 eV vs. NHE), resulting in inefficient oxidation of $H_2O$ by holes to generate $^\bullet OH$. These results confirmed that the fabricated photocatalyst is not formed a type-II heterojunction. As illustrated in Figure 17c, after light illumination and excitation of $TiO_2$ and $Bi_2WO_6$, due to the more negative Fermi level of $TiO_2$ compared to that of $Bi_2WO_6$, the electrons in $TiO_2$ transferred to $Bi_2WO_6$. Next, the internal electric field formed at the interface of the $Bi_2WO_6/TiO_2$ in the direction of $TiO_2$ to $Bi_2WO_6$, and the migration of electrons and holes from CB of $TiO_2$ to $Bi_2WO_6$ and VB of $Bi_2WO_6$ to $TiO_2$ occured. Then, the transformation of electrons on the CB of $Bi_2WO_6$ to VB of $TiO_2$ along the Z direction resulting in the accumulation of electrons and holes with high redox potential in CB of $TiO_2$ and VB of $Bi_2WO_6$ to participate in the photocatalytic reaction.

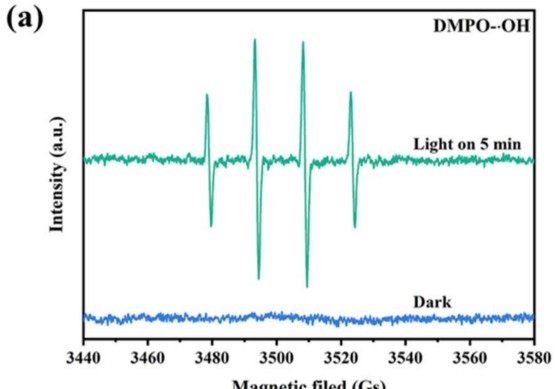

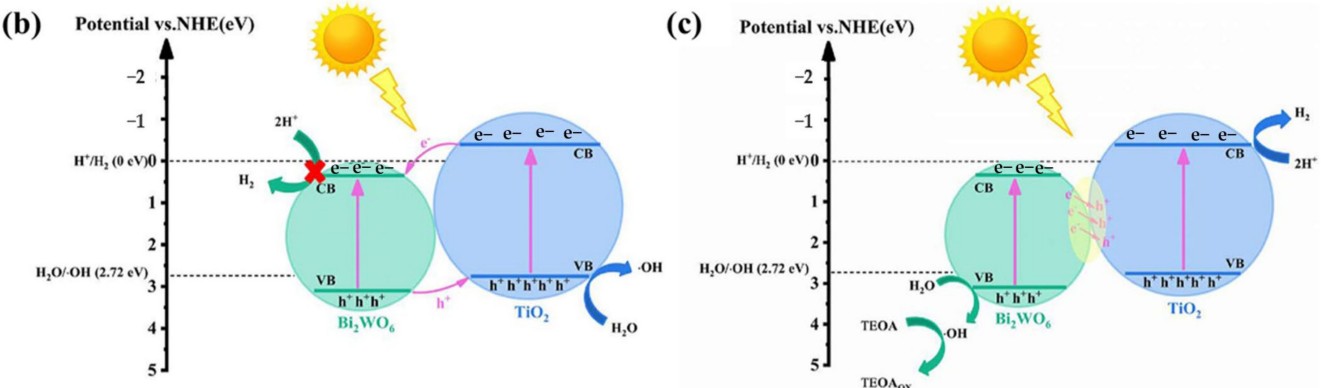

**Figure 17.** (**a**) ESR spectra of $^\bullet OH$; schematic illustration of photocatalytic mechanism: (**b**) traditional type-II, and (**c**) Z-scheme heterostructure. Reproduced with permission from ref. [170].

In the case of environmental issues, Moshfegh's research group [171] synthesized hybrid graphene quantum dots/ZnO nanowires (GQD/ZnO NWs) using an electrochemical method. The as-prepared photocatalyst with direct z-scheme heterostructure showed superior photocatalytic activity for the photodegradation of methylene blue (MB). As illustrated in Figure 18a, by loading the optimum value of GQD (0.4 wt.%), the highest photocatalytic activity was observed after 180 min, three times higher than the pure ZnO NWs. The schematic of the photocatalytic process demonstrates in Figure 18b, under sunlight illumination, the transformation of electrons from CB of ZnO to the HOMO level of GQDs caused recombination of charge carriers with lower redox potential. Thus, the charge carriers with superior redox potential remained in the LUMO level of GQD and the VB of ZnO to participate in the reaction.

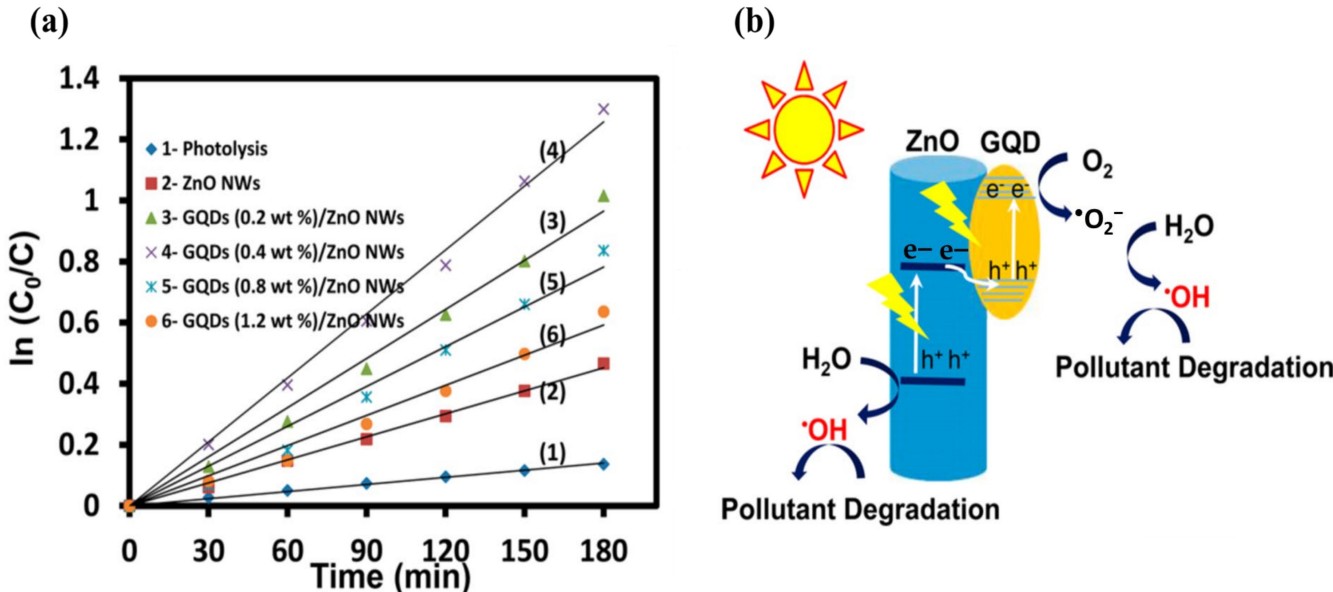

**Figure 18.** (**a**) The curve of ln ($C_0/C$) vs. reaction time for the ZnO NWs and GQD/ZnO NWs with different amount of GQD loadings (**b**); the schematics of the charge carrier pathway for the degradation of MB over GQDs/ZnO NWs photocatalyst under sun light illumination. Reproduced with permission from ref. [171].

Despite the high absorption ability and excellent redox activity, the slow charge transfer rate in Z-scheme heterojunctions as well as the complexity of the contact and high resistance at the interface of the two semiconductors, resulting in decreasing the real photocatalytic applications of these systems [172]. In this regard, the development and construction of dual Z-scheme heterojunctions have been verified as an efficient strategy to address these challenges.

Dual Z-Scheme Heterojunctions

The dual Z-scheme heterojunction construction, which takes advantage of the synergistic effect of three semiconductors, can significantly improve photocatalytic activity by increasing the oxidation or reduction center, enhancing charge carrier separation, increasing the light absorption range, improving sunlight utilization up to 40%, and increasing the carriers lifetime [153,172–174]. In 2015, the concept of a dual Z-scheme heterojunction was first introduced by Fu et al. [175] to explain the excellent photocatalytic performance of a multi-component $TiO_2$-Ag-$Cu_2O$ system in $H_2$ evolution. Following this work, numerous efforts have been made to develop a dual Z-scheme photocatalytic system for clean energy production and pollutant photodegradation. According to the Scopus citation database, by using keywords within the title section (TITLE-ABS-KEY("dual z-scheme")) OR TITLE-ABS-KEY ("double z-scheme"), the number of publications related to this new research area have grown immensely during recent years (Figure 19). From a total of 248 publications, about four were allocated for review articles.

The electron transfer pathway in dual Z-scheme heterojunctions is shown in Figure 20. As shown in Figure 20a, in an A-type heterojunction, both semiconductor I and semiconductor III have higher CB and VB energy levels than that of semiconductor II. After illumination, the excited electrons with lower reduction potential in the CB of semiconductor II recombine with the induced holes with lower oxidation potential in VB of semiconductor I and III. Hence, two strong reduction sites and one strong oxidation site are provided for photocatalytic reaction. Benefiting from the availability of two strong reduction sites in an A-type heterojunction, this system is more suitable for photocatalytic reduction reactions such as converting $CO_2$ to fuel. A B-type heterojunction fabricates by combining one semiconductor with elevated CB and VB energy levels (semiconductor II) than those of

the other two semiconductors (semiconductor I and semiconductor III) (Figure 20b). After illumination, the excited electrons with weak reduction potential in the CB of semiconductor I and III transfer to semiconductor II and recombine with the induced holes with weak oxidation ability. Consequently, two strong oxidation sites and one strong reduction site will be available for reaction. This dual Z-schematic system significantly enhances photocatalytic oxidation reactions, such as $H_2$ production from water splitting. It is noteworthy that both semiconductors are suitable for pollutant photodegradation due to the necessity of simultaneous existence of hydroxyl ($^\bullet$OH) and superoxide radical anion ($^\bullet$O$_2^-$) for the photocatalytic degradation of pollutants. In a C-type heterojunction, as shown in Figure 20c, there is no possibility to combine semiconductor I and III due to the significant potential energy difference between them. As a result, this system is only suitable for $H_2$ generation from water splitting and pollutant degradation. Finally, a D-type heterojunction is built by incorporating an electron mediator between two semiconductors (semiconductor I and II) to form a local electric field and thus increase the charge carrier transportation rate. As shown in Figure 19d, two strong reduction sites and one strong oxidation site are provided by this system, which is more advantageous for photocatalytic pollutant photodegradation [153,172].

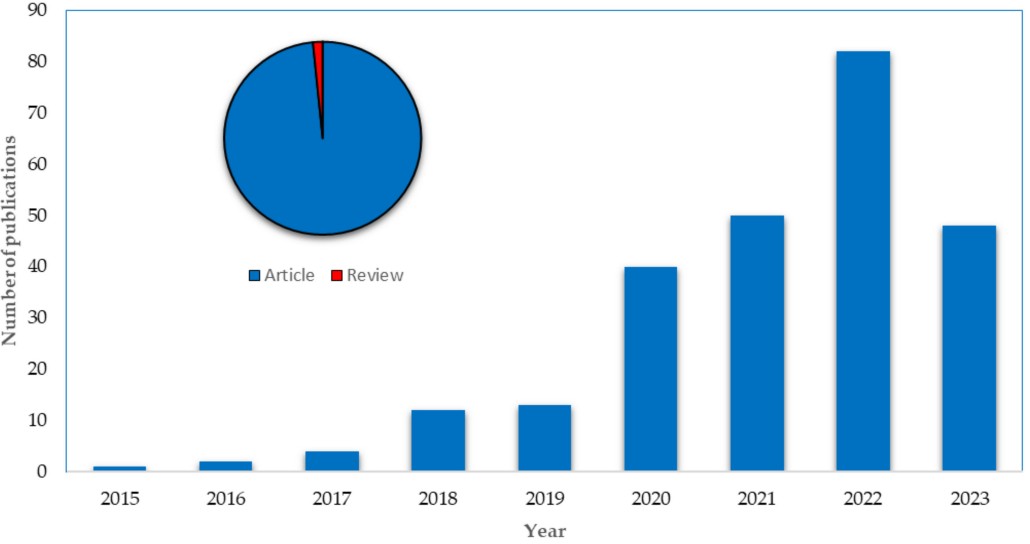

**Figure 19.** The number of publications during 2015 to 29 May 2023 in the Scopus citation database using keywords in (TITLE-ABS-KEY("dual z-scheme")) OR TITLE-ABS-KEY ("double z-scheme")) AND (LIMIT-TO (DOCTYPE, "ar")) OR LIMIT-TO (DOCTYPE, "re")).

Yin et al. [176] employed a precipitation approach with an ion-exchange method to synthesize a dual Z-scheme AgI/BiOI/C$_3$N$_5$ photocatalyst, which demonstrated outstanding photocatalytic activity for the degradation of Tetracycline hydrochloride (TCH). The prepared photocatalyst removed 91.6% of TCH within 50 min, as shown in Figure 21a. The photocatalytic process is schematically illustrated in Figure 21b. As can be seen, the charge transfer mechanism in this system was based on A-type dual Z-scheme process where both g-C$_3$N$_5$ and AgI have higher CB and VB energy levels than BiOI. After illumination, the excited electrons with lower reduction potential in the CB of BiOI recombined with the induced holes with lower oxidation potential in VB of g-C$_3$N$_5$ and AgI. Hence, two strong reduction sites are remained in the CB of C$_3$N$_5$ and AgI. Then the photogenerated electrons reacted with O$_2$ to produce $^\bullet$O$_2^-$. At the same time, one VB level with strong redox potential remains in BiOI, and then the photogenerated holes reacted with H$_2$O to produce $^\bullet$OH radicals. Therefore, this dual Z-scheme system has provided a suitable pathway for electron transfer resulting in superior photocatalytic performance.

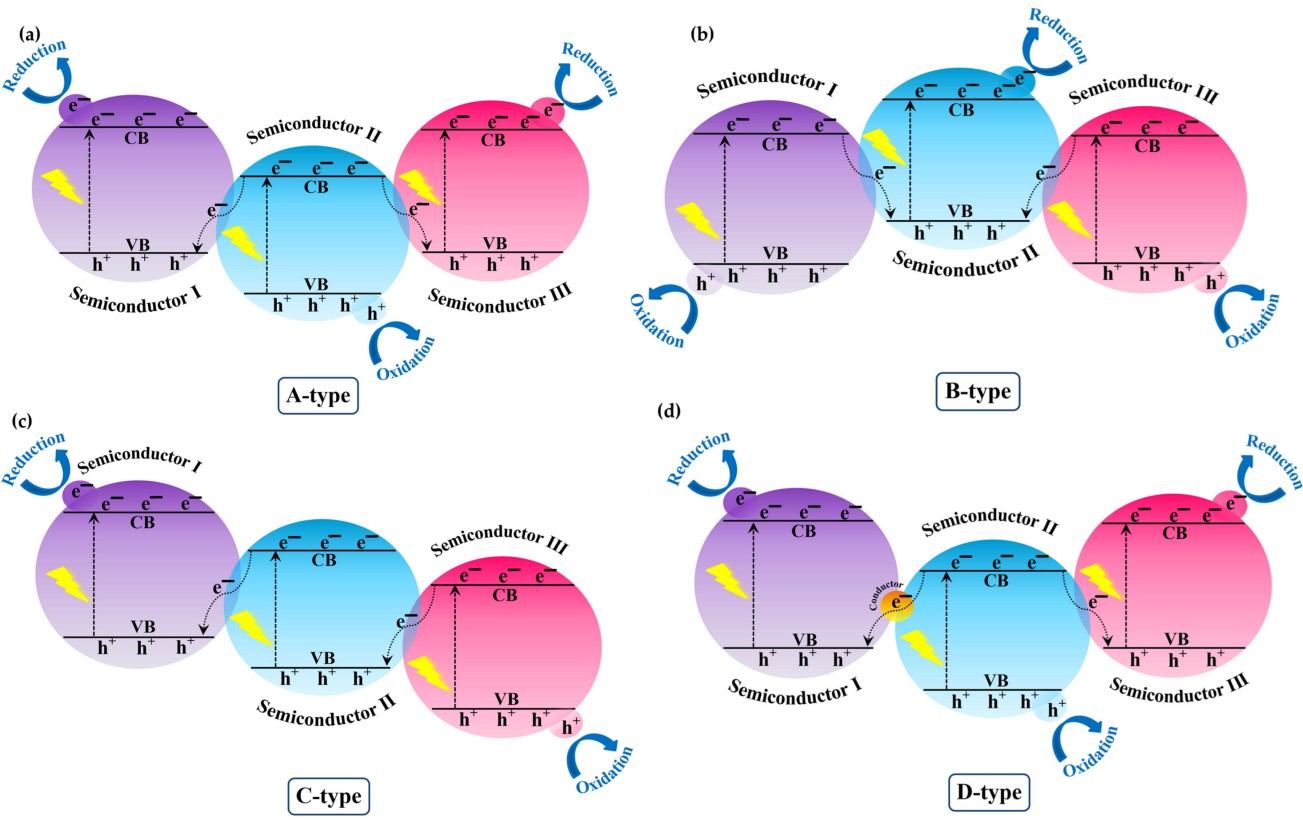

**Figure 20.** Schematic illustration of charge carrier transfer mechanism of the dual Z-scheme hetero-junctions: (**a**) A-type; (**b**) B-type; (**c**) C-type; (**d**) D-type dual Z-scheme heterojunctions.

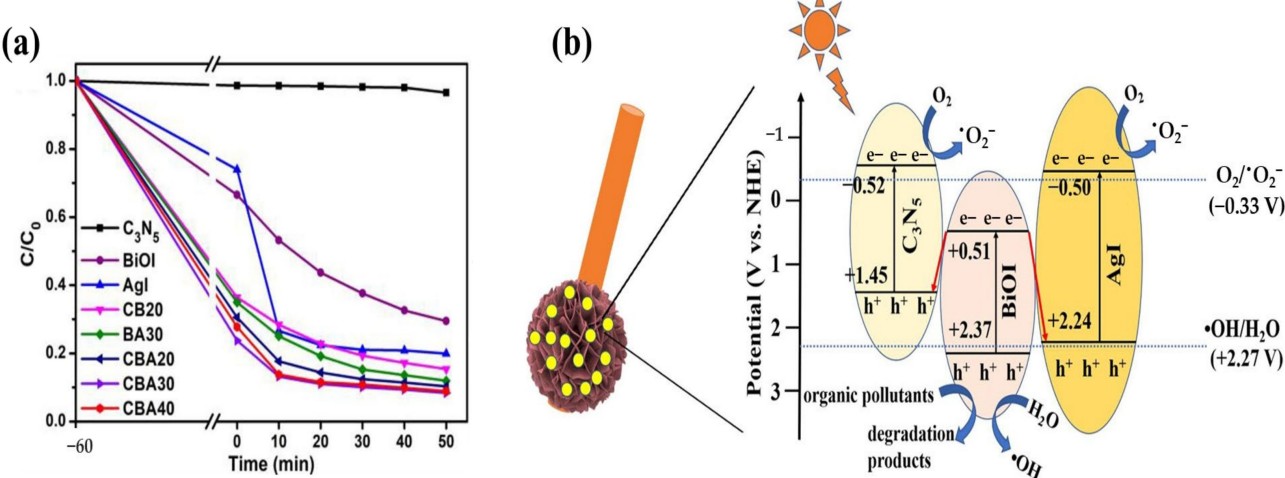

**Figure 21.** (**a**) Photocatalytic TCH degradation over AgI/BiOI/C$_3$N$_5$; (**b**) Schematic illustration of photocatalytic charge transfer mechanism of dual Z-scheme AgI/BiOI/C$_3$N$_5$. Reproduced with permission from ref. [176].

S-Scheme Heterojunction

In 2019, Fu et al. [177] introduced the concept of an S-scheme heterojunction to solve the limitations of a type-II, traditional Z-scheme, and all-solid-state Z-scheme heterojunction. In an S-scheme heterojunction two semiconductors with different work function values are in contact with each other. The CB bottom, Fermi level, and VB top levels of semiconductor II are higher than those of semiconductor I, as demonstrated in Figure 22a. When the two semiconductors are combined, the charge distribution will occur at the

interface, which results in the formation of a built-in electric field in the direction of semiconductor II to semiconductor I (Figure 22b). The presence of the built-in electric field induces an upward and downward bending of the energy bands in the semiconductor II and I, respectively. After illumination, the built-in electric field and electrostatic attraction accelerate the recombination of the photogenerated electrons and holes with weak redox potentials. Consequently, the photogenerated electrons and holes with strong redox abilities are separately remain in the semiconductor II and I, respectively, as shown in Figure 22c. The broad applicability of the S-scheme heterojunction makes it highly suitable for photocatalytic clean energy production and environmental remediation [178].

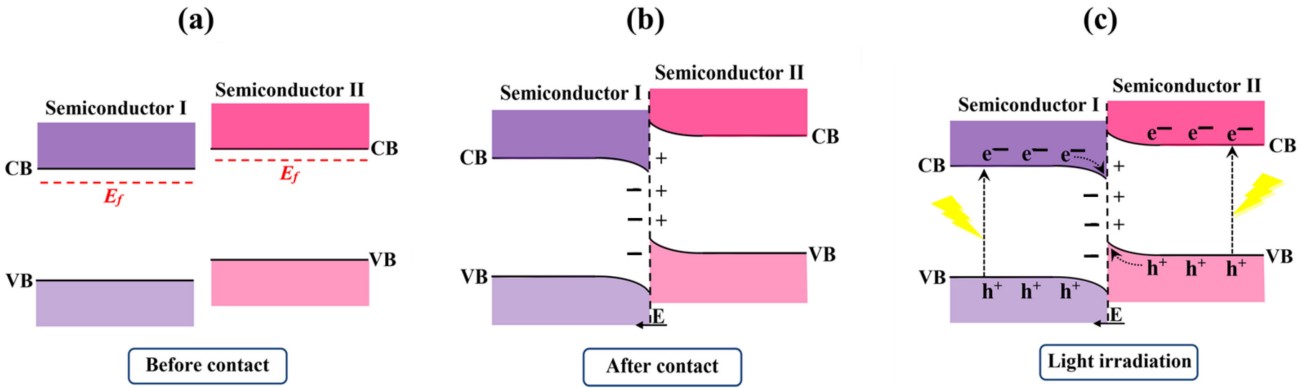

**Figure 22.** Schematic illustration of S-scheme heterojunction.

Cai et al. [179] fabricated $Cd_{0.5}Zn_{0.5}S/Bi_2MoO_6$ (CZS/BMO) S-scheme heterojunction by coupling $Cd_{0.5}Zn_{0.5}S$ nanoparticles and $Bi_2MoO_6$ microspheres as an efficient photocatalyst for photodegradation of Enrofloxacin. The fabrication of S-scheme heterojunction between $Cd_{0.5}Zn_{0.5}S$ and $Bi_2MoO_6$ photocatalysts, optimized the light absorption and charge separation. The charge-migration pathway of the CZS/BMO heterojunction was examined using the electron spin resonance (ESR) technique. As depicted in Figure 23a, distinctive signals of DMPO$-{}^{\bullet}O_2{}^-$ were observed for the BMO, CZS, and 1.0CZS/BMO, indicating their sufficient photoredox capacity to generate ${}^{\bullet}O_2{}^-$ under visible-light irradiation. Notably, the signal intensity of DMPO$-{}^{\bullet}O_2{}^-$ for 1.0CZS/BMO was significantly enhanced compared to the BMO and CZS, indicating electron aggregation on the CB of CZS, resulting in abundant ${}^{\bullet}O_2{}^-$ production. In Figure 23b, no noticeable peaks of DMPO$-{}^{\bullet}OH$ were detected for CZS due to its VB potential (1.34 V), which was lower than those of ${}^{\bullet}OH/OH^-$ (1.99 V) and ${}^{\bullet}OH/H_2O$ (2.38 V). However, the 1.0CZS/BMO photocatalyst exhibited a remarkable intensity of the DMPO$-{}^{\bullet}OH$ signal, surpassing that of BMO, indicating an enhanced oxidation capacity achieved through the S-scheme heterojunction construction. These observations confirmed that photocarriers were separated and transferred in the S-scheme mechanism. The photocatalytic of this process is schematically illustrated in Figure 23c,d. Considering the lower position of the Fermi level of the BMO compared with that of the CZS, the electrons transferred from the CZS to BMO, which resulted in the creation of a built-in electric field. The presence of strong internal electric field (IEF) can effectively ensure that the photocarriers follow the separation pathway of the S-scheme. The built-in electric field induced an upward and downward band bending in the energy levels. Under the influence of IEF and bands bending, the photo-generated electrons in the CB of the BMO automatically migrated to the VB of the CZS and combined with the holes. In contrast, the electrons of the CZS and holes of the BMO were well preserved and participated in the reaction. The constructed S-scheme in this study provided a suitable pathway for electron transfer, resulting in a superior photocatalytic activity that yields the highest degradation efficiency of 76.3% within 40 min [179].

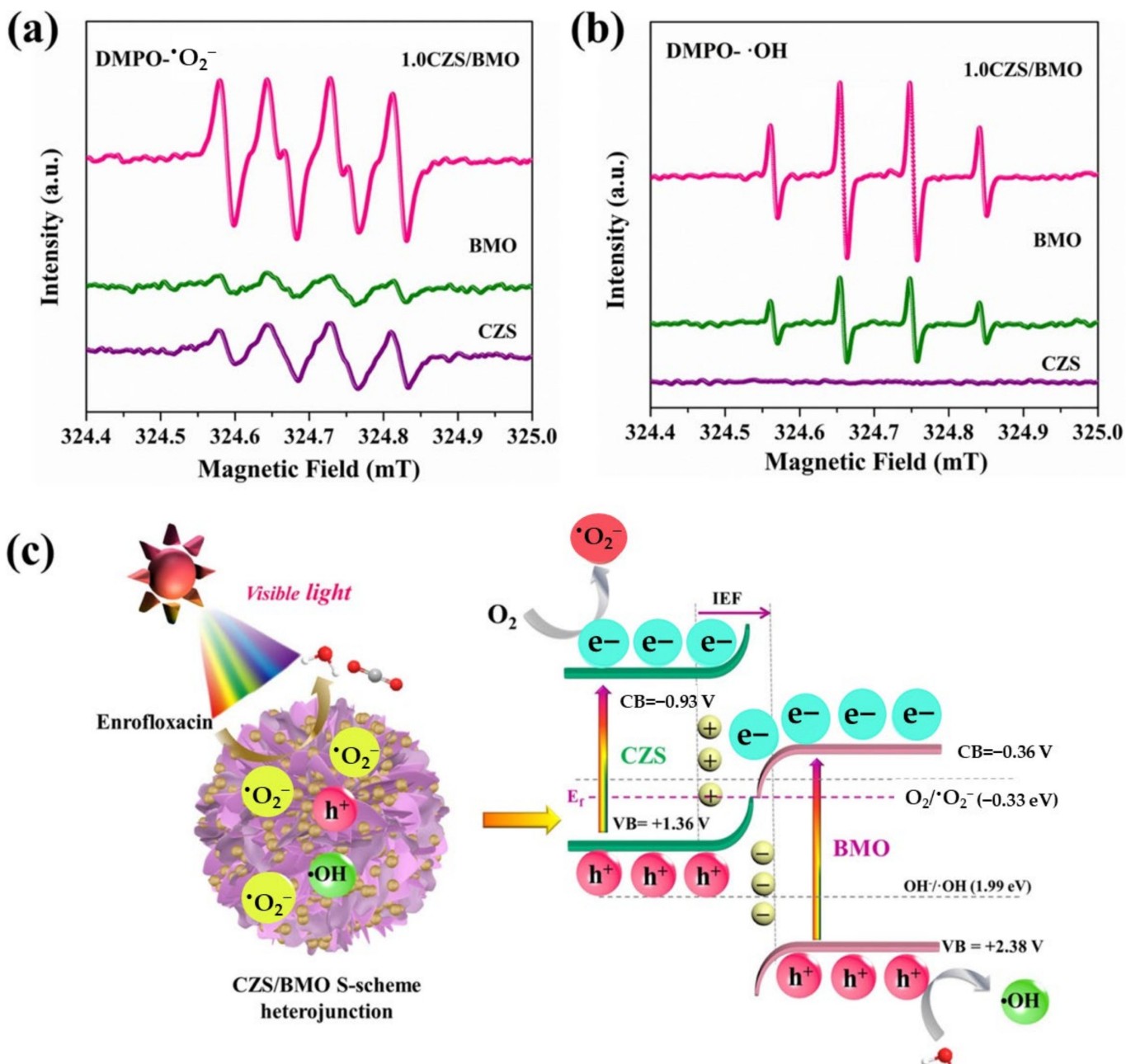

**Figure 23.** Spectra of (**a**) •$O_2^-$ and (**b**) •OH for BMO, CZS and 1.0CZS/BMO under visible-light illumination; (**c**) Schematic illustration of the $Cd_{0.5}Zn_{0.5}S/Bi_2MoO_6$ S-scheme heterojunction. Reproduced with permission from ref. [179].

### 3.2. Reaction Conditions

### 3.2.1. Photocatalyst Concentration

One of the crucial factors impacting the photocatalytic efficiency is the photocatalyst concentration. Generally, the higher the catalyst loading, the more surface active sites will be available, and the more electron–hole pairs will be generated, leading to improved photocatalytic activity [180]. Nevertheless, an excessive increase in the photocatalyst concentration decreases the reaction rate due to the reflection of a significant amount of incident radiation by scattering and preventing the catalyst particles from receiving more photons [181]. Hu et al. [182] illustrated that by increasing the $PeCoFe_2O_4$@GCN-1.0 dosage from 15 to 40 mg, the photocatalytic degradation of Tetracycline (TC) increased

from 65.9% to 98.2%, but the corresponding average degradation rate decreased from 0.0329 to 0.0185 h$^{-1}$ because of decreasing in the light transmission of the solution at higher concentration of photocatalyst (Figure 24). Therefore, it is essential to optimize the amount of photocatalyst concentration in order to maximize the rate of photodegradation.

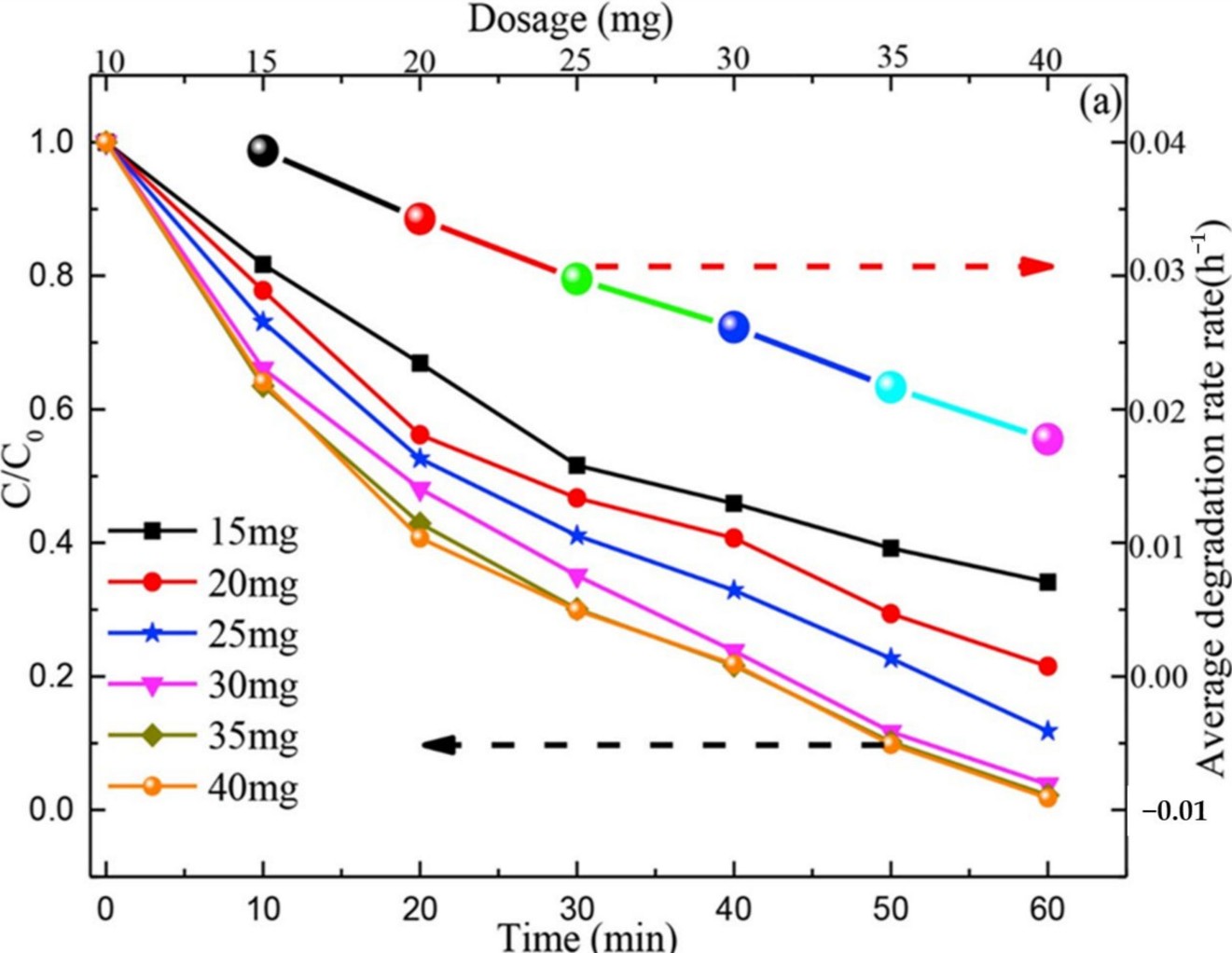

**Figure 24.** The effect of the photocatalyst dosage on the degradation of tetracycline. Reproduced with permission from ref. [182].

### 3.2.2. Initial Contaminant Concentration

In general, the photocatalytic degradation efficiency increases with increasing pollutant concentration but decreases after reaching a specific concentration level [183]. Screening the irradiated light before reaching the photocatalyst surface by contaminant molecules, blocking active sites of the photocatalyst surface with pollutant molecules as well as reducing the number of absorbed photons on the photocatalyst surface followed by a reduction in the number of ROS, leads to decreasing the photocatalytic efficiency at a very high pollutant concentration [184]. Nguyen et al. [185] showed that the photocatalytic performance of the $ZnFe_2O_4@ZnO$ nanocomposites in the degradation of Congo red dye decreases (95–42%) at initial concentrations of 2.5–15 mg·L$^{-1}$. Zhu et al. [186] observed that the optimal removal efficiencies of levofloxacin and Cr (VI) over the Sn(IV)-doped $Bi_2O_2CO_3$ photocatalyst were obtained with initial concentrations of both pollutants of 10 mg·L$^{-1}$ (Figure 25a,b). Therefore, utilizing an appropriate concentration level of the contaminant plays a crucial role in enhancing photocatalytic efficiency.

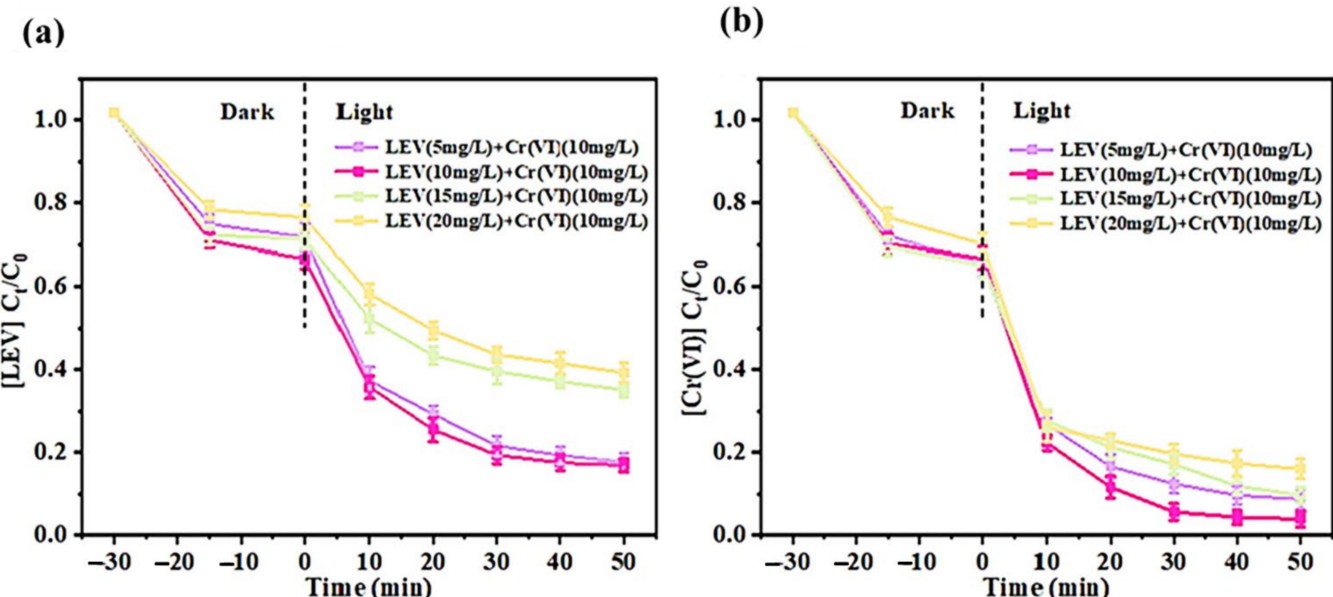

**Figure 25.** The effect of pollutants concentrations on the removal of (**a**) levofloxacin; (**b**) Cr (VI). Reproduced with permission from ref. [186].

### 3.2.3. pH

The photocatalytic reaction efficiency is significantly affected by pH changes because of changing the surface charge of a photocatalyst with pH changes [187]. pH level also has effects on the predominant oxidation species. The essential oxidation species at higher and lower pH values are hydroxyl radicals and holes, respectively [183]. Ahmad et al. [188] reported that an optimum pH value of 4 resulted in 100% degradation of Congo red by $CeFeO_3$. As the pH raised to 7 and 10, the degradation yield reduced to 97.3% and 18.52%, respectively (Figure 26). The reasons for these observations are attributed to two facts: firstly, the electrostatic attraction between the photocatalyst and dye will increase because of the positively charged of the photocatalyst surface at pH = 4. Secondly, the production of hydroxyl radicals will be facilitated in acidic media. For hydrogen evolution reaction (HER), decreasing the pH value results in the production of higher $H^+$ ions, which is beneficial for a reaction [189].

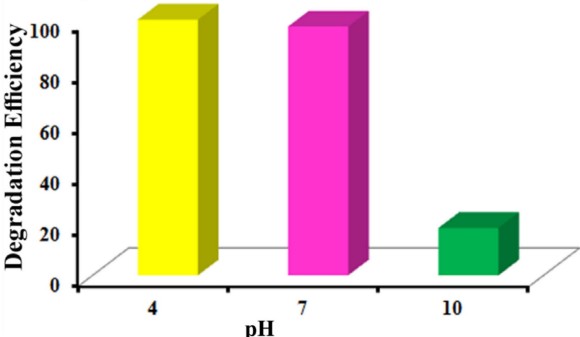

**Figure 26.** The effect of pH on the photodegradation of Congo red. Reproduced with permission from ref. [188].

### 3.2.4. Reaction Temperature

Another critical factor in the photocatalytic reaction is temperature. In general, the photocatalytic degradation rate accelerates by increasing the reaction temperature. Enhancing the formation of active free radicals in solution, increasing the oxidation rate of pollutant molecules at the interface, and reducing the electron–hole pair recombination rate

lead to significant improvement in the photocatalytic efficiency at higher reaction temperatures [184,190]. For example, Wang et al. [191] reported that the rate of the photocatalytic hydrogen production over $C_3N_4/Ni_xP$/Red phosphorus Z-Scheme photocatalyst during water splitting reaction enhanced from 1.78 to 6.16 μmol h$^{-1}$ by increasing the reaction temperature up to 70 °C under visible light irradiation (Figure 27a). In another study, Lais et al. [192] investigated the effect of reaction temperature on the rate of methanol formation during $CO_2$ reduction. They observed that the reaction rate constant of methanol production increased with increasing temperature. This result was attributed to the enhanced diffusion and collision frequencies of the reactants at higher temperatures. The effect of the reaction temperature on the environmental remediation process was also studied by several researchers. For instance, very recently, Rahmani et al. [193] demonstrated that the photocatalytic degradation rate of enrofloxacin over CdS/CuAg photocatalyst increased by increasing the reaction temperature from 10 °C to 40 °C due to increasing its rate constant and the higher efficiency obtained at 40 °C (Figure 27b), which confirms the positive effect of increasing temperature on photodegradation efficiency.

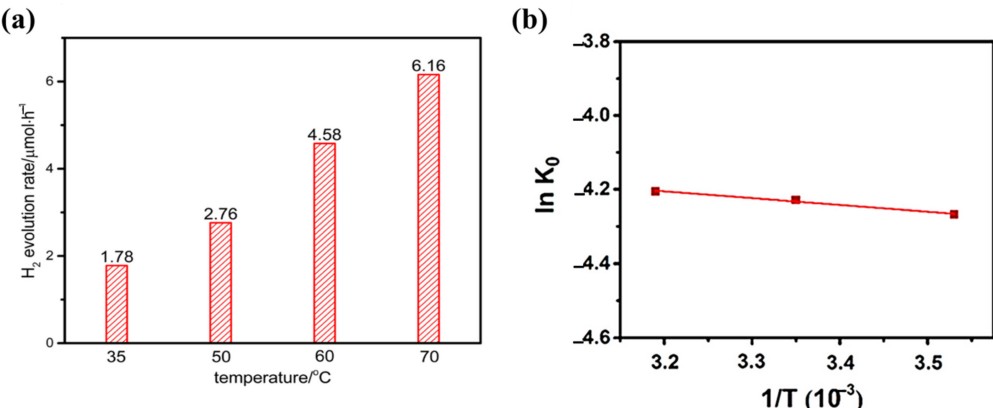

**Figure 27.** (**a**) $H_2$ production efficiency over the $C_3N_4/Ni_xP$/RP photocatalyst under visible light irradiation at different temperatures. Reproduced with permission from ref. [191]; (**b**)Arrhenius plot of ln ($k_0$) vs. (1/T) for the photocatalytic degradation of Enrofloxacin over CdS/CuAg bi-alloy composite. Reproduced with permission from ref. [193].

### 3.2.5. Light Intensity

The efficiency of a photocatalytic reaction improves with increasing light intensity due to increasing the formation of charge carriers and reducing the possibility of recombination [194]. A linear correlation exists between the measured photo-current density (J) and light intensity (I) (J–I) for an ideal photo-responsive material. However, in an actual material, the relationship is not perfectly linear. It is often modeled as a power law dependence, where J is approximately equal to I raised to the power β ($0 \leq β \leq 1$), where the higher the scaling exponent, β, approaches 1, the greater the photocatalytic activity of a material. The β = 1 implies that as the light intensity increases, all incident photons are converted to photo-current with equal efficiency. In other terms, β = 1 indicates the absence of any nonlinear effects, including direct recombination of charge carriers in the layer, which not only verifies the critical role of separated charge carriers in the reaction, but also confirms that J is independent of the series resistance of the layer. Both bimolecular recombination and series resistance cause a sub-linear behavior and result in photocurrent saturation at high light intensities [195]. Moshfegh's research group [196], investigated the relationship between the photogenerated current density ($I_{Ph}$) and the light intensity ($I_o$) in $H_2$ production over Ag:$TiO_2$ nanocomposite thin film photoanode. As shown in Figure 28, there is a linear relation between $I_{Ph}$ and $I_o$, so that, the photoresponse of Ag:$TiO_2$ increased with increasing the light intensity in all selected potentials in the saturation regions (from −0.1 to 0.7 V).

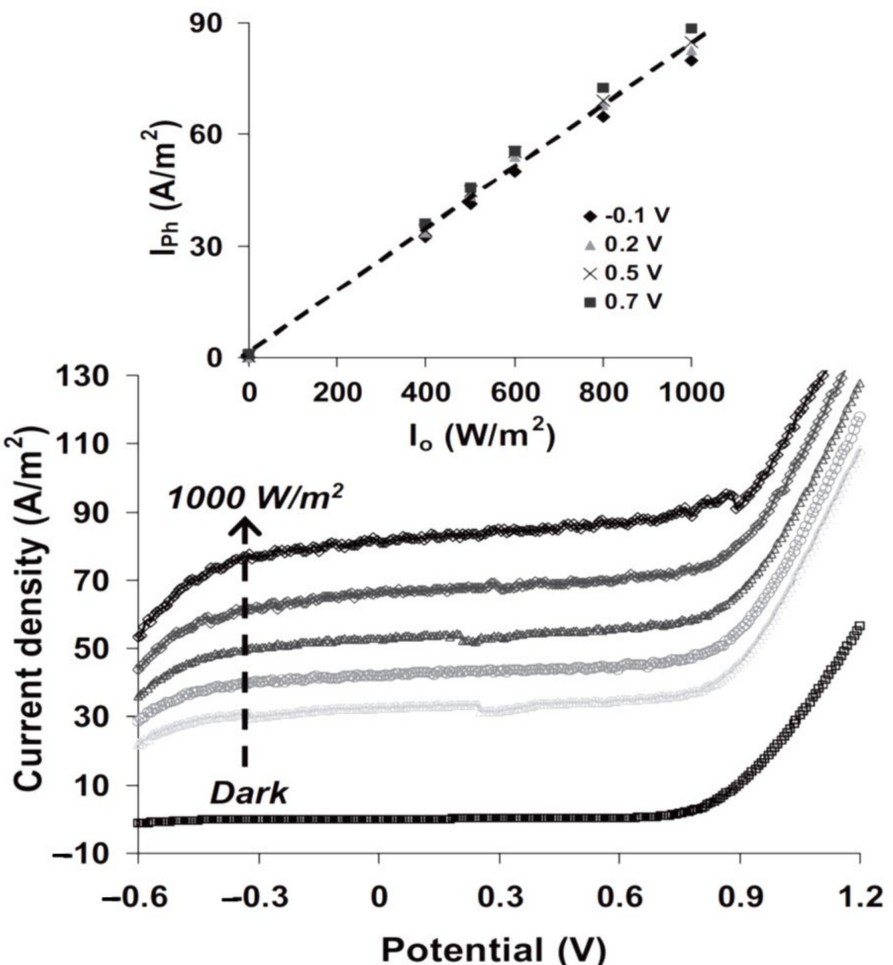

**Figure 28.** The curve of the photogenerated current density ($I_{Ph}$) over annealed Ag:TiO$_2$ thin films with 1 mol% Ag, vs. applied potential and light intensity ($I_o$). Reproduced with permission from ref. [196].

In a recent study, Pan et al. [197] reported that the performance of CoPi/Mo:BiVO$_4$ photoanode increased (Figure 29a,b), because of increasing the photocurrent density under high light intensity and increased reaction temperature that attributed to the increased amount of photoexcited carriers under high light intensity.

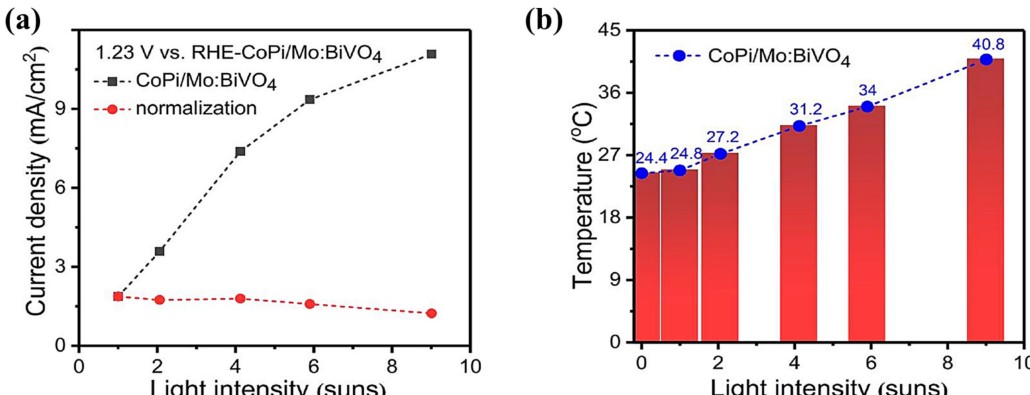

**Figure 29.** (**a**,**b**) The photocurrent density and the normalized photocurrent density of CoPi/Mo:BiVO$_4$ at 1.23 V vs. RHE and the temperature of CoPi/Mo:BiVO$_4$ photoanodes at various light intensities (1–9 suns illumination). Reproduced with permission from ref. [197].

### 3.2.6. Charge-Carrier Scavengers

In order to remove unwanted reaction products such as oxygen during a photocatalytic process and thus avoid undesirable reactions occurrence and to examine a photocatalytic activity by inactivating the reactive species, a scavenger is used [184]. The most commonly used scavengers in a photocatalytic reaction are ammonium oxalate and disodium ethylenediaminetetraacetate (EDTA-2Na) as the hole scavengers, dimethylsulfoxide (DMSO) and $AgNO_3$ as electron scavengers, 1,4-benzoquinone (BQ) as a $^\bullet O_2^-$ scavenger, methanol, 2-propanol (IPA), and *tert*-butanol (*t*-BuOH) as $^\bullet OH$ scavengers [53]. Moshfegh's group [87] recently employed different scavenger tests by using ethylenediaminetetraacetic acid disodium salt (EDTA), $AgNO_3$, and t-BuOH as holes, $^\bullet OH$, and electron scavengers to investigate the photodegradation mechanism of methylene blue (MB) using $ZnO/g$-$C_3N_4$ composite nanofibers. In this work, the oxygen of the pollutant solution was removed by passing $N_2$ gas to prevent the formation of $^\bullet O_2^-$. As shown in Figure 30, after adding $AgNO_3$, the photocatalytic efficiency tremendously decreased, implying that the electron is the primary reactive species in the reaction.

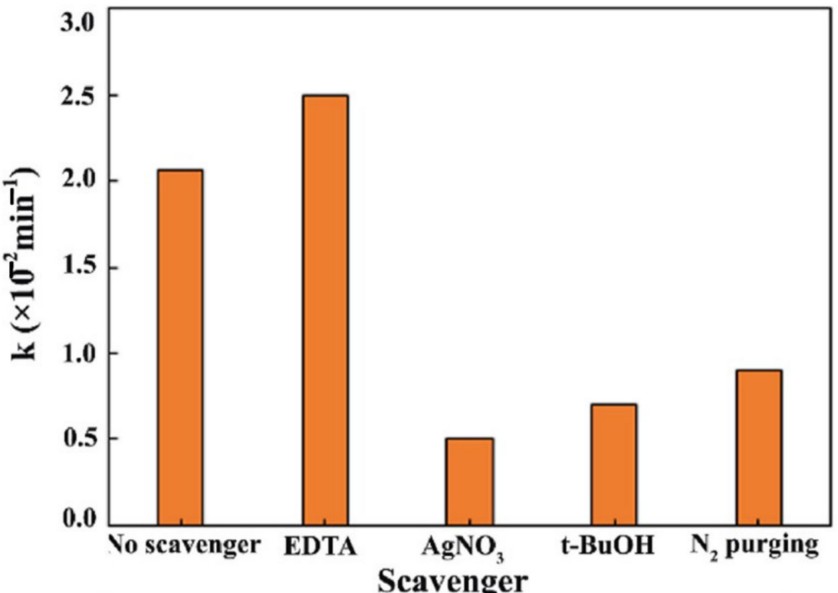

**Figure 30.** k values of the $ZnO/g$-$C_3N_4$ composite nanofibers photocatalyst with different scavengers Reproduced with permission from ref. [87].

For better understanding and realizing different photocatalytic processes on nanostructure semiconductor surfaces/interfaces, several photocatalytic reactions will be discussed. Three types of photocatalytic reactions, namely photoelectrochemical $H_2$ production from water splitting, photocatalytic $CO_2$ reduction, and photocatalytic dye/drug degradation, will be presented in Sections 4–6, respectively.

## 4. Photoelectrochemical $H_2$ Production from Water Splitting

It is well known that the efficient conversion of solar energy to chemical energy for producing clean energy is based on utilizing an appropriate semiconductor photocatalyst. Thus, hydrogen can be produced via photoelectrochemical (PEC) water splitting on semiconductor photocatalysts with effective nanostructures with high surface-to-volume ratio and high light absorption ability. A typical PEC cell consists of two half-reactions (a) oxygen evolution reaction (OER) usually occurs on an n-type semiconductor as a photoanode, and (b) hydrogen evolution reaction (HER) occurs on a cathode as a counter electrode, as illustrated in Figure 31.

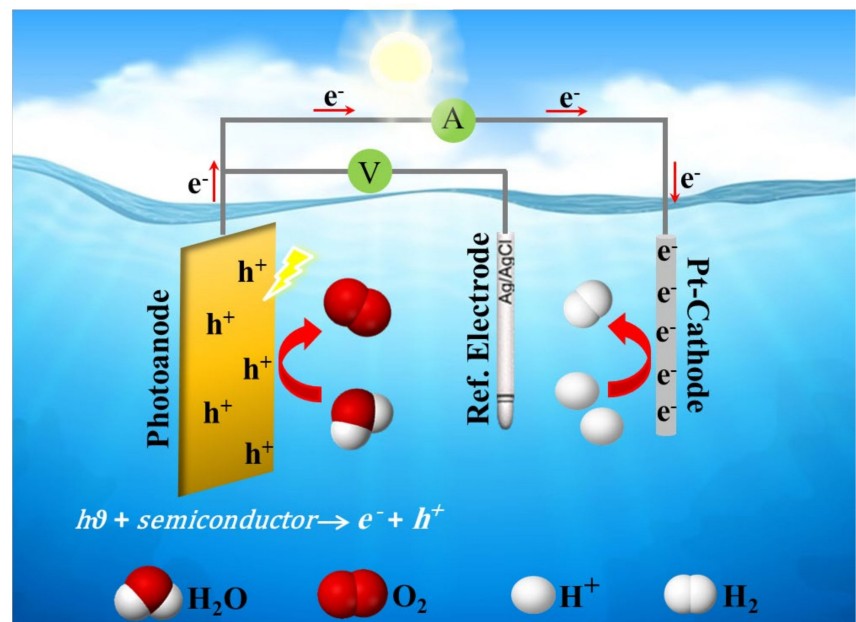

**Figure 31.** A schematic diagram of a typical photoelectrochemical (PEC) cell.

There are extensive efforts conducted by many researchers to maximize the efficiency of hydrogen production rate under solar irradiation by examining different nanostructured semiconductor photocatalysts. Table 1 presents some of the results obtained on different photoanode semiconductors.

**Table 1.** Some of the newest synthesized photocatalysts for $H_2$ production from water splitting.

| Entry | Photocatalyst | Synthesis Method | Preparation Condition | Photocatalytic Test Condition | Photocurrent Density, Efficiency | Ref. |
|---|---|---|---|---|---|---|
| 1 | Sb-SnS | hydrothermal | 180 °C for 16 h. | Simulated solar, AM 1.5 G, 100 mW cm$^{-2}$, 0.5 M Na$_2$SO$_4$ | 3.3 mA cm$^{-2}$ | [198] |
| 2 | BiVO$_4$, Bi$_2$O$_3$, TiO$_2$ | spin-coating and calcination | sol-gel-based spin coating method | Na$_2$SO$_4$ solution without hole scavengers | 5.1, 3.4, and 2.1 mA cm$^{-2}$ | [199] |
| 3 | CN/Mn$_2$O$_3$ | plasma-assisted liquid-based | melamine and C$_3$H$_3$N$_3$O$_3$ | 0.1 M Na$_2$SO$_4$, pH = 6.5 | 25 µA cm$^{-2}$ | [200] |
| 4 | TiB$_2$@AuNPs | spin-coating deposition | Ultrasonic assisted liquid-phase exfoliation | sunlight simulator (1 Sun), 0.1 M KOH | 10 mA cm$^{-2}$ | [201] |
| 5 | NiO/BiVO$_4$ | dip-coating | Ink: Ni nanocrystals | 1 M KBi, hole scavenger: 0.2 M Na$_2$SO$_3$ | 4.41 mA cm$^{-2}$ | [202] |
| 6 | β-FeOOH/CN | particle-to-substrate contacts | precursor: DCDA and TCY | AM 1.5G illumination, 0.1 M NaOH | 320 µA cm$^{-2}$ | [203] |
| 7 | Fe$_2$O$_3$ (Hematite) | CVD | precursor: ferrocene organometallic compound | Simulator solar AM 1.5 G filter, 1 M NaOH | 2.5 mA cm$^{-2}$ | [204] |
| 8 | Ir SAs/NiO/Ni/ ZrO$_2$/n-Si | ALD process | precursor: TICP | simulated solar AM 1.5 G, 1 M NaOH | 27.7 mA cm$^{-2}$ | [205] |
| 9 | Ni(OH)$_2$/Cl-BiVO$_4$ | impregnation method | precursor: 0.5 M NH$_4$Cl | AM1.5 G filter, 100 mW cm$^{-2}$, 0.5 M K$_3$BO$_3$, pH = 9.5 | 4.33 mA cm$^{-2}$ | [206] |
| 10 | Bi$_2$S$_3$-Cu$_3$BiS$_3$ | in-situ decoration | precursor: Cu:Bi 3:1 ink | simulated solar light (AM 1.5G), K–Pi buffer | 7.8 mA cm$^{-2}$ | [207] |
| 11 | BiVO$_4$/Vo$_x$ | electrodeposition | precursor: BiOI | simulated solar light (AM 1.5G), 1 M borate buffer | 6.29 mA cm$^{-2}$ | [208] |

## 5. Photocatalytic CO$_2$ Reduction

The substantial fossil fuels consumption resulted in vast carbon emissions with an atmospheric concentration of over 400 ppm, as reported in 2019. The over usage of fossil fuels ((e.g., petroleum, gas, and coal)) consumption is responsible for discharging a large amount of CO$_2$ into the environment after their combustion. The release of CO$_2$ accounts for around 76% of annual greenhouse gas (GHG) emissions which creates severe problems for the environment and public health, such as global warming and climate changes, seawater acidification, and the upsurge of ocean levels. To solve these severe ecological issues, the conversion of CO$_2$ into valued small-molecule chemical products or energy fuels (such as CO, CH$_4$, HCOOH, and other chemicals) through photocatalytic technology, can solve the energy crisis as well as environmental issues. Extensive efforts have been conducted by many researchers to convert GHG, such as CO$_2$, into valuable products and reduce its catastrophic emission. As a result of this process, environmental problems such as climatic turbulences, which comprise ecological deterioration, seawater acidification, and the upsurge of ocean levels will be reduced [59,209–214]. Table 2 presents some results obtained on different photocatalyst semiconductors along with preparation methods, photocatalytic conditions, and product yield.

**Table 2.** A summary of the synthesized photocatalysts for CO$_2$ reaction.

| Entry | photocatalyst | Synthesis Method | Preparation Condition | Photocatalytic Conditions | Photocatalytic Activity (Product Yield) | Ref. |
|---|---|---|---|---|---|---|
| 1 | MOF [Cu$_3$Th$_6$(μ$_3$-O)$_4$(μ$_3$-OH)$_4$(cpb•)$_{12}$] [Fe$^{III}$(CN)$_6$]$_6$ | mixed, stirred and centrifuged | precursors: Th(NO$_3$)$_4$, formic acid, | 300 W Xe lamp, 420 or 800 nm filter | CO production rate: 570.3 μmol g$^{-1}$h$^{-1}$ | [215] |
| 2 | Porphyrin-based MOFs | solvothermal | precursor: TPP and MnCl$_2$ | 300 W Xe lamp, 1 sun (AM1.5 G) | CO production rate: 46.148 μmol g$^{-1}$h$^{-1}$ | [214] |
| 3 | boron-doped g-C$_3$N$_4$/TiO$_{2-x}$ | thermal reduction process | precursor: mixture of urea and NaBH$_4$ | 300 W Xe lamp | CO production rate: 265.2 μmol g$^{-1}$h$^{-1}$ | [210] |
| 4 | Co-COF: | mix and sonication | precursor: TAPT, 120 °C, 96 h | 300 W Xe lamp (Microsolar 300, cut-off 420 nm), (λ ≥ 420 nm) | CO production rate: 18,000 μmol g$^{-1}$h$^{-1}$ H$_2$ production rate: 800 μmol g$^{-1}$h$^{-1}$ | [211] |
| 5 | GQDs/C-BN | hydrothermal | precursors: boric acid, melamine and glucose | 300 W Xe lamp (λ = 200–420 nm or λ > 700 nm) | CO production rate: 33.47 μmol g$^{-1}$h$^{-1}$ | [216] |
| 6 | Ni(OH)$_2$ | hydrolyzation of CaO | stirring for 24 h at 298.15 K. | LED lamps (5 W, λ ≥ 420 nm) | CO production rate: 9.2 μmol g$^{-1}$h$^{-1}$ | [217] |
| 7 | S-ZnS/ZnIn$_2$S$_4$ | hydrothermal | precursor: Methylimidazole | 300 W Xe lamp cut-off filter (λ > 420 nm) | CO production: 2075.7 ± 63.0 μmol g$^{-1}$h$^{-1}$ H$_2$ production: 2912.3 ± 185.9 μmol g$^{-1}$h$^{-1}$ | [218] |
| 8 | BP-CN | mix and sonication | Precursor: [NH$_3$OH]$^+$Cl$^-$ and melamine | 300 W Xe lamp | CO production rate: 44.6 μmol g$^{-1}$h$^{-1}$ | [219] |
| 9 | P-doped PCN | thermal polymerization method | precursor: Melamine and NaH$_2$PO$_2$·H$_2$O | 300 W Xe lamp, cut-off filter 420 nm | CH$_4$ production rate: 1.1 μmol g$^{-1}$ h$^{-1}$ | [220] |
| 10 | Ni-BiOBr | hydrothermal | precursor: Bi(NO$_3$)$_3$·5H$_2$O | 300 W xenon lamp cut-off 380 nm filter (λ ≥ 380 nm) | CO production rate: 378.7 μmol g$^{-1}$h$^{-1}$ | [221] |

## 6. Photocatalytic Dye/Drug Degradation

Based on a World Bank report, dyeing and textile industries have contributed to about 17–20% of water pollution [222]. According to a recently published article, the textiles, leathers, food, and paper industries are primarily in charge of producing dye wastewater, with an annual global production of 8 × 10$^5$ tons of dyes, of which around 200,000 tones are textiles and dyes [223]. Numerous synthetic dyes used in textile, including both cationic dyes (e.g., safranin O (SO), rhodamine B(RhB), malachite green (MG), rhodamine 6G (Rh6G), methylene blue (MB), and crystal violet (CV)) and anionic dyes (e.g., eosin Y (EY), Eriochrome black T (EBT), phenol red (PR), methylene orange (MO), and Congo red (CR))

are toxic and harmful organic contaminants that will impede the photosynthesis process of aquatic plants and pose a threat to living creature health via the channels of food and drinking water supply [224–226].

Other organic wastes are produced by chemical and pharmaceutical industries that are harmful to society and the environment. During past decades, there has been a significant increase in the production and utilization of Pharmaceutical and Personal Care Products (PPCPs) to accommodate the demands of contemporary lifestyles and increased health care [227]. According to the European Union market, there are approximately 3000 frequently used drugs, and their usage is still on the rise worldwide [228]. Based on a recent report, the worldwide usage of antibiotics is believed to range from 100,000 to 200,000 metric tons. When excreted from the body, 70–90% of consumed antibiotics remain chemically unchanged or as active metabolites [229]. Moreover, the global production and consumption of PPCPs have significantly increased as a result of the COVID-19 pandemic in recent years. The findings of the National Health Commission of the People's Republic of China indicate a dramatic increase in the consumption of antiviral and antibiotic drugs during the pandemic [230]. Pharmaceutical contaminants are found in water at concentrations as trace as $ng \cdot L^{-1}$ to $\mu g \cdot L^{-1}$. However, owning their chemical and physical properties, they can pose a serious threat to living organisms' health even at these low concentration levels [231]. Given the severe problems resulting from water pollution caused by dye and pharmaceutical pollutants, as well as regarding the significant advantages of photocatalysis processes in removing harmful pollutants from water, numerous attempts have been undertaken to develop highly effective semiconductor-based photocatalysts to photodegrade of dye and pharmaceutical pollutants globally. A summary of various photocatalysts used for the degradation of dye and pharmaceutical contaminants in water, along with their synthesis method and reaction conditions, is presented in Table 3.

**Table 3.** A summary of various synthesized photocatalysts used for the degradation of dye and pharmaceutical contaminants in water.

| Entry | Photocatalyst | Synthesis Method | Preparation Condition | Pollutant Type | Light Condition | Efficiency (%) | Time (min) | Ref. |
|---|---|---|---|---|---|---|---|---|
| 1 | $CN/CQD/BiOCl_{0.75}Br_{0.25}$ | hydrothermal | pH = 5.8 [TC] = 100 mg·L$^{-1}$ Catalyst dose = 0.1 g·L$^{-1}$ | Tetracycline(TC) | 500 W Xe lamp ($\lambda$ > 400 nm) | 83.4 | 30 | [232] |
| 2 | $\alpha$-Fe$_2$O$_3$@TiO$_2$ | sonication and wet impregnation | pH = 4.76 [CFX] = 20.5 mg·L$^{-1}$ Catalyst dose = 0.012 g·L$^{-1}$ | Cefixime (CFX) | 500 W halogen visible light (>400 nm) | 98.8 | 103 | [233] |
| 3 | DBS/CNNS | calcination | pH = 5.5 [MOX] = 50 mg·L$^{-1}$ Catalyst dose = 1 g·L$^{-1}$ | moxifloxacin (MOX) | 300 W xenon lamp ($\lambda$ > 420 nm) | ~100.0% | 30 | [234] |
| 4 | 0.1 chl/0.1 SA-TiO$_2$ | incipient wetness impregnation | pH = 6 [CPX] = 10 mg·L$^{-1}$ Catalyst dose = 0.75 g·L$^{-1}$ | Ciprofloxacin (CPX) | Blue LED light ($\lambda$ = 457 nm) | ~75% | 120 | [235] |
| 5 | ZnO/ZnIn$_2$S$_4$ (ZnO/ZIS) | hydrothermal | pH = 3 [CS] = 10 mg·L$^{-1}$ Catalyst dose = 0.40 g·L$^{-1}$ | ceftriaxone sodium (CS) | 500 W xenon lamp. | 85.3% | 150 | [236] |
| 6 | In$_2$S$_3$/MQDs/SmFeO$_3$ (IMS) | sonication | [SMX] = 10 mg·L$^{-1}$, catalyst dose = 0.6 g·L$^{-1}$ pH = 5.3 | sulfamethoxazole (SMX) | 300 W Xe lamp | 98.0% and 95.4% of SMX | 120 and 90 min, | [237] |
| 7 | MgCr-LDH | formamide-assisted co-precipitation and mild hydrothermal | pH = 7 [MB] = 20 mg·L$^{-1}$ Catalyst dose = 30 mg | methylene blue (MB) | solar light | 90.6 | 120 | [238] |
| 8 | ZnO-CT | a green synthesis route using lemon leaf extract | pH = 7 [CR] = 20 mg·L$^{-1}$ Catalyst dose = 0.4 g·L$^{-1}$ | Congo red (CR) | Natural sunlight, ($\lambda$ = 408 nm) | 97 | 90 | [239] |
| 9 | Au/La$_2$Ti$_2$O$_7$/Ag$_3$PO$_4$ | The in-situ precipitation | pH = 9.6 [RhB] = 10 mg·L$^{-1}$ Catalyst dose = 1 g·L$^{-1}$ | Rhodamine (BRhB) | Natural sunlight | 100 | 6 | [240] |

**Table 3.** *Cont.*

| Entry | Photocatalyst | Synthesis Method | Preparation Condition | Pollutant Type | Light Condition | Efficiency (%) | Time (min) | Ref. |
|---|---|---|---|---|---|---|---|---|
| 10 | UiO-66-NH$_2$/PhC$_2$Cu | hydrothermal | pH = 9 [NOR] = 10 mg·L$^{-1}$ Catalyst dose = 0.2 g·L$^{-1}$ | norfloxacin (NOR) | 9 W LED lamp (455 nm) | 97.9 | 60 | [241] |
| 11 | CuPd/ZnO | hydrothermal and chemical reduction | pH = 2 [OM] = 40 mg·L$^{-1}$ Catalyst dose = 0.5 wt.% | methyl orange (MO) | solar simulator ($\lambda$ = 440 nm) | 95.3 | 45 | [242] |
| 12 | mesoporous Fe/Al/La trimetallic nano-oxide(FAL) | chemical route | pH = 7 [dyes] = 10$^{-5}$ M Catalyst dose = 0.30 g/100 mL | black 5 (RB5) methylene blue (MB) direct blue 71 (DB71) and | Sunlight | 93.85 ± 2 90.51 ± 2 91.16 ± 2 | 90 45 60 | [243] |
| 13 | Fe$_2$O$_3$/CNT/MIL | hydrothermal | pH = 7 [OFX] = 20 mg·L$^{-1}$, Catalyst dose = 100 mg·L$^{-1}$ | ofloxacin (OFX) | 300 W Xe lamp ($\lambda$ > 420 nm) | 99.3 | 60 | [244] |
| 14 | $\alpha$-NiMoO$_4$/ ZnFe$_2$O$_4$/BC | Pyrolysis and hydrothermal | pH = 10 [KP] = 10 mg·L$^{-1}$, Catalyst dose = 100 mg·L$^{-1}$ | ketoprofen (KP) | visible light (UV cutoff 150 W LS xenon arc lamp) | 98.65 | 180 | [245] |
| 15 | 0D/2D AgI/CAU-17 | deposition-precipitation | pH(for RhB degradation) = 3 [RhB] = 10 mg·L$^{-1}$, [KP] = 10 mg·L$^{-1}$, [MO] = 5 mg·L$^{-1}$, Catalyst dose = 0.25 mg·L$^{-1}$ | Rhodamine B (RhB) Tetracycline (TC) methyl orange (MO) | 500 W Xe lamp | 96.7 81.3 50.3 | 90 | [246] |

## 7. Simultaneous Photocatalysis

In the last decade, heterogeneous photocatalytic reactions have experienced many efforts and devoted studies in developing advanced and innovative materials to confront both environmental and energy crises via the utilization of appropriate semiconducting photocatalysts in many essential chemical reactions, including wastewater treatment, H$_2$ generation, CO$_2$ reduction, organic transformations, N$_2$ photofixation, and biomass conversion to valuable products. It is well known in a conventional photocatalytic investigation, and these processes are occurred in a controlled condition and are discussed in the literature separately. However, a recent innovative approach is to engage two or more functions in one photocatalytic system simultaneously. The challenging point is that combining of two functions in one photocatalytic system requires a novel design, control, and engineering of an appropriate semiconductor photocatalyst with unique characteristics for each application in a particular environment. The concept of dual-purpose photocatalysis was discussed in the pioneering work of Kim et al. [247] on simultaneous hydrogen production and phenolic compounds' degradation by using a TiO$_2$ surface decorated with platinum nanoparticles and fluorine atoms (F-TiO$_2$/Pt) as a photocatalyst. They obtained complete mineralization of organic compounds and found anoxic degradation of 4-chlorophenol accompanied by H$_2$ production. Because an appropriate photocatalyst characteristic is required for each process, thus the primary challenge is to conduct two or more kinds of applications simultaneously over one photocatalyst. Thus, it is an ideal approach to design and utilizes a special photocatalyst in at least two different concurrent applications. For example, 2D semiconductors (e.g., GO, rGO, and MXenes) and their hybrid combinations can be applied to produce hydrogen generation and pollutant degradation simultaneously. Nugmanova A. G. et al. [248] synthesized zinc porphyrin metal–organic frameworks non-covalently attached to graphene oxide (SURMOF/GO) in Pickering emulsions, and investigated its photocatalytic activity during photodegradation of rhodamine 6G (Rh6G) and 1,5-dihydroxynaphtalene (DHN). In another study, Nikoloudakis et al. [249] fabricated a covalently linked nickel (II) porphyrin–ruthenium(II) tris(bipyridyl) dyad for photocatalytic water oxidation reaction in dimethylformamide (DMF) using methyl viologen as a sacrificial electron acceptor. As depicted in Figure 32, the NiP-Ru dyad demonstrated

notably higher $O_2$ evolution in comparison with the non-covalent system consisting of NiP and Rubpy and the turnover number (TON) of 18 was achieved for the NiP-Ru dyad after one hour of continuous visible light irradiation. This group acknowledged that utilizing an organic solvent (DMF) in this photocatalytic system offers the advantage of simultaneously coupling water oxidation and $CO_2$ reduction catalysts within the same cell for efficient $CO_2$ conversion.

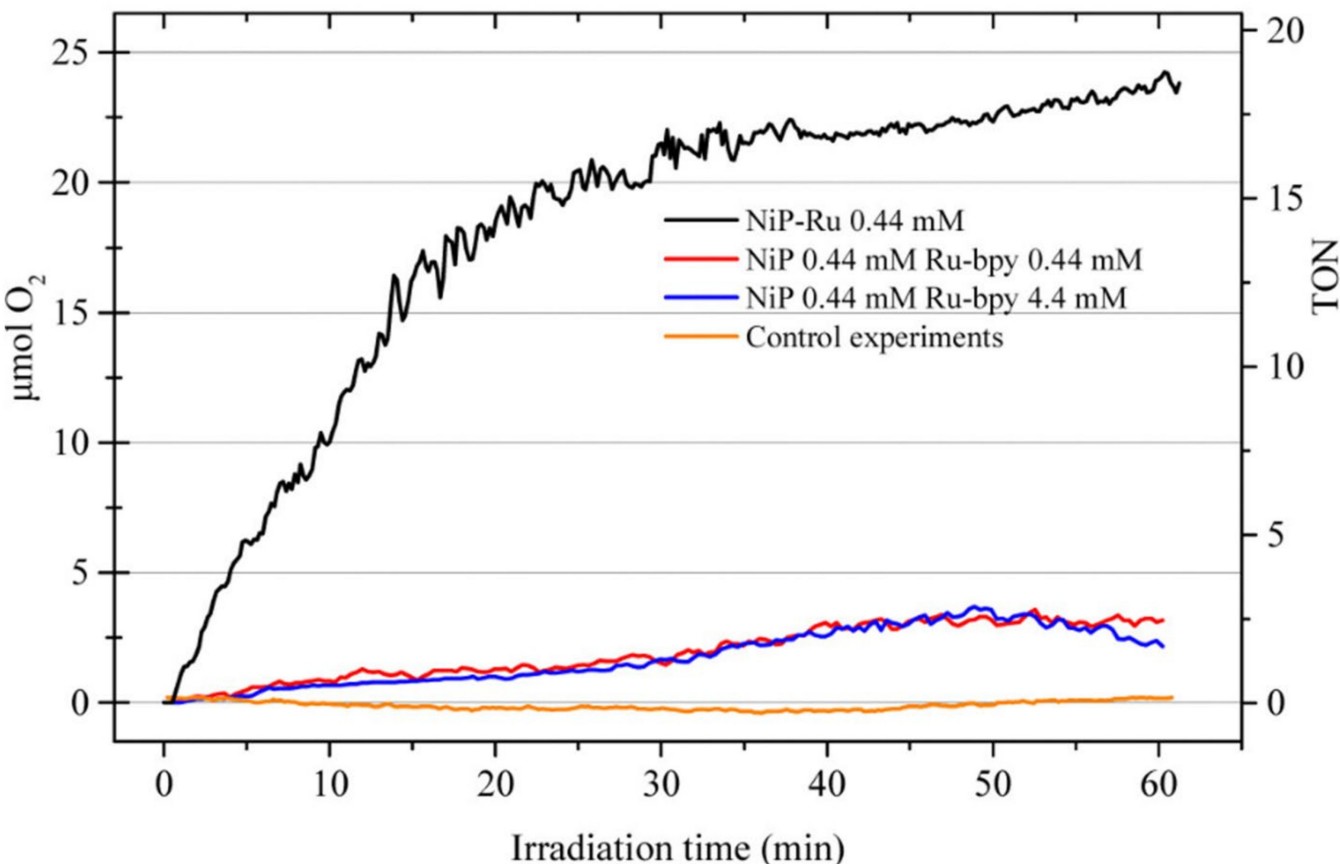

**Figure 32.** $O_2$ evolution through photocatalytic $H_2O$ oxidation after irradiation with a 450 W Xenon lamp (with a λ > 420 nm cutoff filter) in the presence of 50 mM of Methyl Viologen in DMF and 4% $H_2O$. Reproduced with permission from ref. [249].

In another study, Wu et al. [250] reported that the photodegradation yield of 4-chlorophenol was 99.9% after 60 min, and the $H_2$ production rate by oxygen vacancies-enriched titanium dioxide ($O_v$-$TiO_2$) photoanode was 198.2 μmol h$^{-1}$ cm$^{-2}$ that is significantly higher than that on the $TiO_2$ working electrode. For example, in the case of simultaneous organic pollutant oxidation and $H_2$ evolution, the chemical potential of pollutant oxidation can provide the chemical potential needed for hydrogen evolution ($H^+$/$H_2$; 0 V vs. NHE) from water splitting [50]. In another study, dual functional photocatalysis was conducted to tackle both environmental and energy challenges. Figure 33 illustrates different simultaneous photocatalytic reactions involved in clean energy production and environmental remediation [251].

Xu et al. [252] applied nitrogen-rich carbon nitride nanotubes (CNNTs) coupled with 3 wt.% Pt as a co-catalyst for simultaneous degradation of bisphenol A (BPA) and $H_2$ generation resulting in 92% photodegradation rate after 5 h and 18.06 mmol h$^{-1}$ g$^{-1}$ hydrogen production under visible-light irradiation as shown in Figure 34.

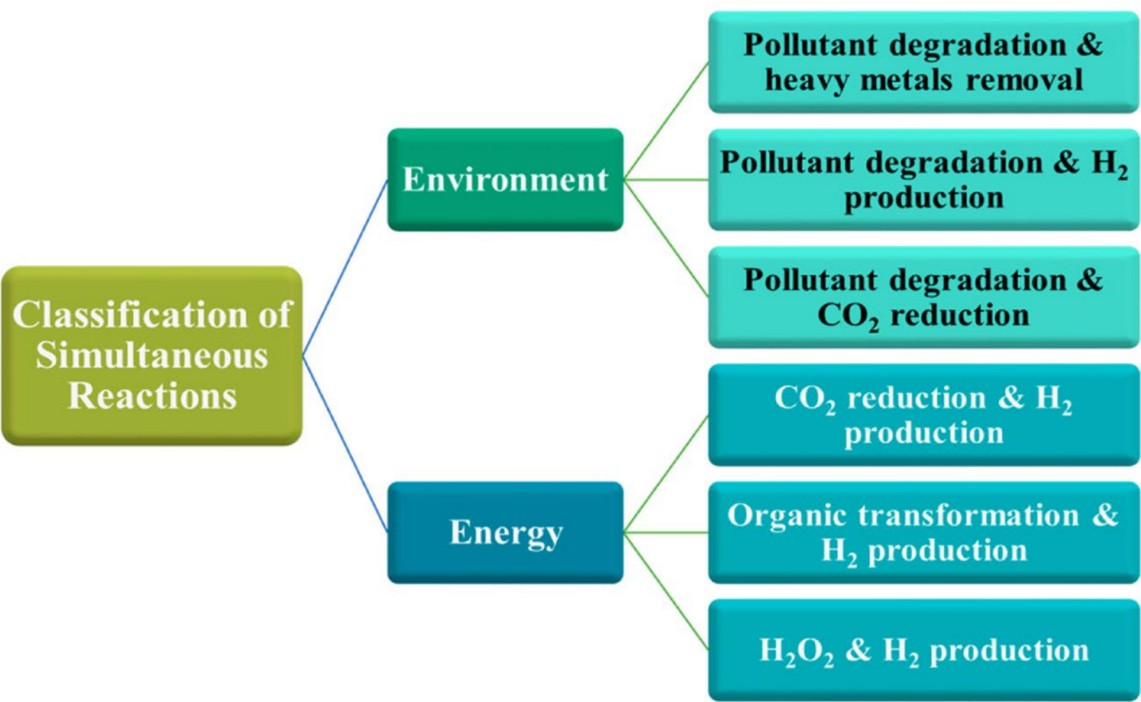

**Figure 33.** Different simultaneous photocatalytic reactions involved in clean energy production and environmental remediation. Reproduced with permission from ref. [251].

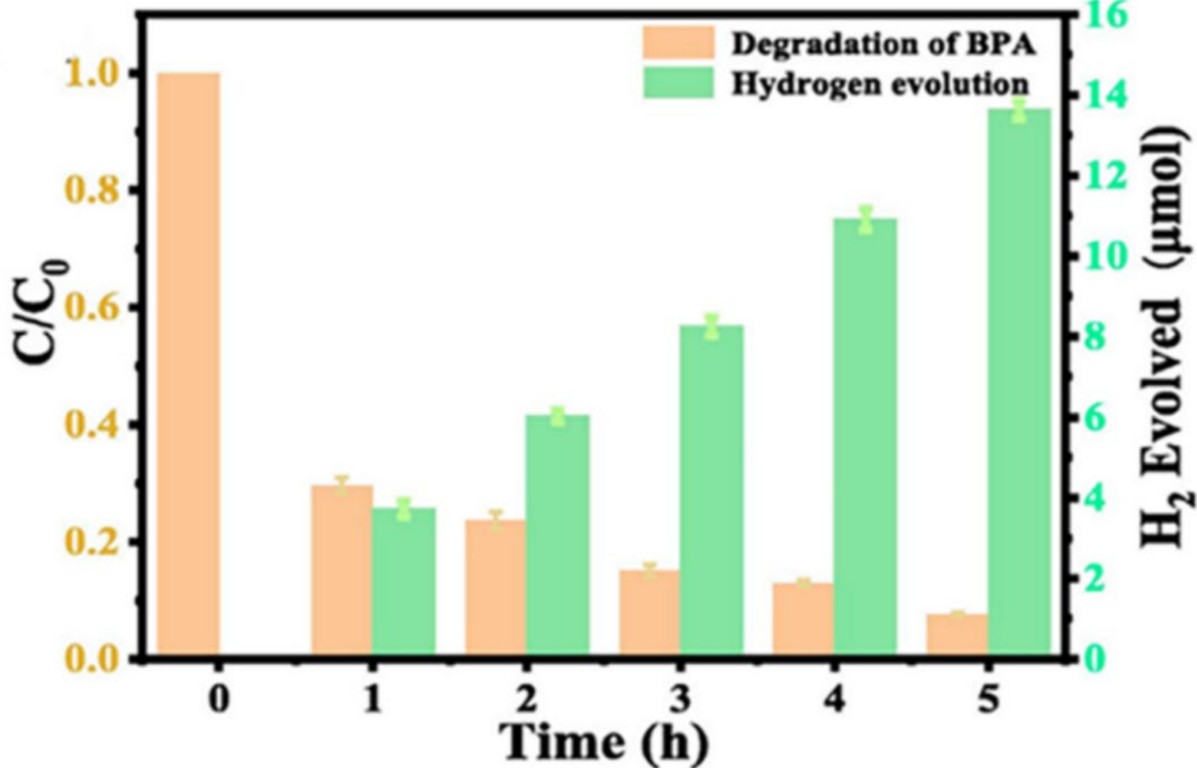

**Figure 34.** Rate of simultaneous photodegradation of bisphenol A (BPA) and $H_2$ generation over CNNTs. Reproduced with permission from ref. [251].

Additional examples of various photocatalysts used for simultaneous pollutant photodegradation and $H_2$ evolution are presented in Table 4.

**Table 4.** A summary of various photocatalysts along with their reaction condition applied in simultaneous photodegradation of pollutant and $H_2$ evolution from water splitting.

| Entry | Photocatalyst | Synthesis Method | Photocatalytic Test Condition | $H_2$ Activity ($\mu$mol h$^{-1}$ g$^{-1}$) | Pollutant: Photodegradation Efficiency (%) | Ref. |
|---|---|---|---|---|---|---|
| 1 | 0.5 wt.% Pt/Zn-V-20 | Calcination, hydrothermal | 300 W Xe lamp AM 1.5 G filter 0.5 M Na$_2$SO$_4$, 15% CH$_3$OH | 5230.4 | 2,2′,4,4′-tetrahydroxybenzophenone (BP-2): (99.6), methylene blue (MB): (99.4), acetaminophen (AAP): (92.0) | [253] |
| 2 | g-C$_3$N$_4$/BiOI/CdS | calcination, solvothermal, and solution chemical deposition | 300-W Xe lamp with a $\lambda$ > 420 nm cutoff filter | 863.44 | bisphenol A: (98.62) | [254] |
| 3 | Mo$^1$@CNNTs | template free polymerization | 300 W xenon lamp coupled with $\lambda$ = 420 nm cutoff filter. | 4861 | tetracycline hydrochloride: (97.3) | [255] |
| 4 | [g-C$_3$N$_4$/polymethylmethacrylate (PMMA)]//[TiO$_2$/polyaniline (PANI)/PMMA]//[self-assembled 3, 4, 9, 10-perylene tetraformyl diimide (PDI)/PMMA] (TMOP) | tri-axial parallel electrospinning | simulated sunlight | 536.7 | Ciprofloxacin: (88.99), tetracycline hydrochloride: (91.15), chlortetracycline hydrochloride: (77.55), levofloxacin: (69.51), and colored dye methylene blue: (92.50) | [256] |
| 5 | SCN/NiS-1 | hydrothermal | visible light (400 nm filter) irradiation | 700.9 | Rhodamine B (RhB): (98.5) | [257] |
| 6 | ZnS@Zn$_{0.58}$Cd$_{0.42}$S | hydrothermal | Xe lamp (CEL-PF300-T9, CEAU) with an AM1.5G filter | 36,000 | Helianthine: (94.2) | [258] |
| 7 | NbO-BRGO | hydrothermal | 300 W Xe > 400 nm | 1742 | crystal violet (CV): (97.6) | [259] |
| 8 | ZnIn$_2$S$_4$@SiO$_2$@TiO$_2$ | sol–gel and solvothermal | 300 W xenon lamp | 618.3 | methylene blue: (99.7) | [260] |
| 9 | Ag@TiO$_2$-P25-5%MoS$_2$ | Combination of photocatalysts | solar simulator composed of two white light bulbs (60 watts) | 1792 | Ciprofloxacin: (75) | [261] |
| 10 | MoS$_2$/ZnO | hydrothermal | 250 W metal halide lamp | 235 | Ciprofloxacin: (89) | [262] |

## 8. Industrial Photocatalyst Application

It is widely accepted that photocatalysts with promising industrial applications must possess three critical features: (i) low-cost production, (ii) effective charge carrier separation, and (iii) an ideal band gap to facilitate optimal utilization of the solar spectrum [263]. Considering the promising results obtained by several photocatalysts in energy production and environmental remediation at the laboratory scale, researchers are intensifying their efforts to advance the industrialization of photocatalyst technology for the applications mentioned above. In the field of clean energy production, Lee et al. [264] synthesized a floatable photocatalytic platform fabricated from porous elastomer–hydrogel nanocomposites for scalable solar $H_2$ production. The platform's scalability, as an essential factor for industrialization, was demonstrated under natural sunlight. It was confirmed that the floatable photocatalytic platform shows an excellent $H_2$ evolution rate of 163 mmol h$^{-1}$ m$^{-2}$ in the presence of Pt/TiO$_2$ cryoaerogel with an area of 1 m$^2$ using Cu/TiO$_2$ photocatalysts. Furthermore, the system generates 79.2 mL of hydrogen per day under natural sunlight. The schematic of the nanocomposite and its advantages for photocatalytic hydrogen production (i–vi) has been shown in Figure 35.

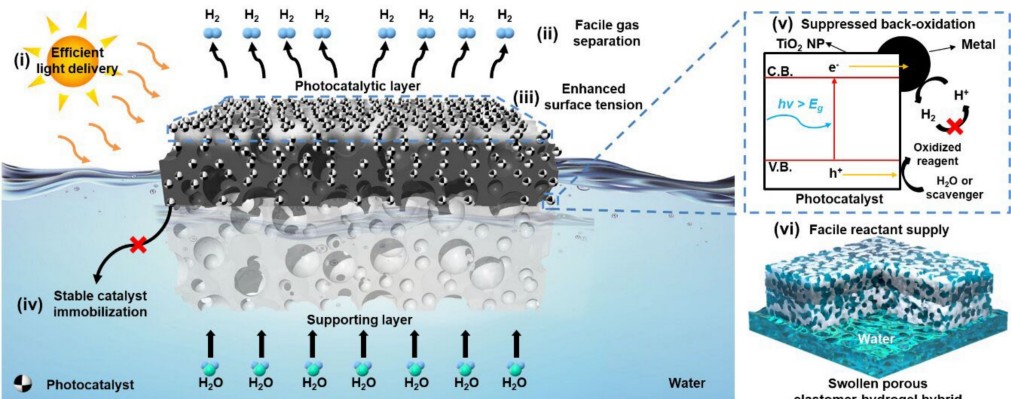

**Figure 35.** A schematic illustration floatable photocatalytic platforms along with its advantages in hydrogen evolution reaction (**i**) light delivery; (**ii**) facile gas separation; (**iii**) enhanced surface tension; (**iv**) stable catalyst immobilization; (**v**) suppressed back-oxidation (reverse reaction); and (**vi**) facile supply of water. Reproduced with permission from ref. [264].

In the field of environmental remediation, Chaudhuri et al. [265] developed a photochemical rotor−stator spinning disk reactor (pRS-SDR) capable of handling and scaling solid-containing photochemical reactions on the flow condition. They investigated the efficiency of the reactor in handling slurries for the $TiO_2$-mediated aerobic photodegradation of methylene blue (MB). By utilizing a high-speed rotating disk, the pRS-SDR achieved uniform mixing of the reaction components, enhanced mass transfer, and optimized irradiation throughout the reaction mixture. This innovative system is expected to facilitate the shift from batch to continuous-flow processes in the pharmaceutical and agrochemical sectors. Its ability to scale up complex multiphase synthetic reactions makes it a valuable tool in advancing industrial operations. The schematics of photocatalytic degradation of MB using $TiO_2$ photocatalyst in the pRS-SDR are illustrated in Figure 36. The highest MB photodegradation rate of 90.0% was reported at a liquid flow rate of 19.5 mL/min [265].

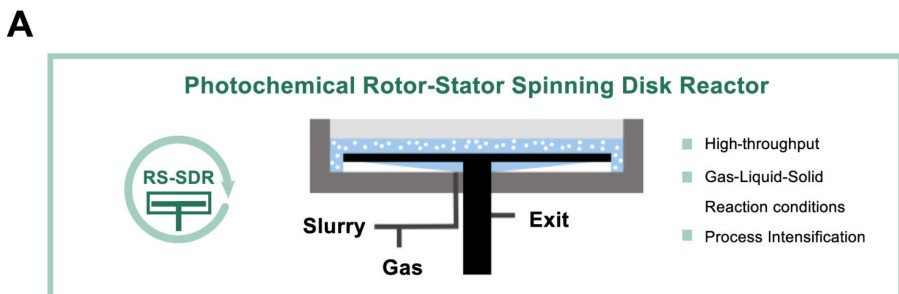

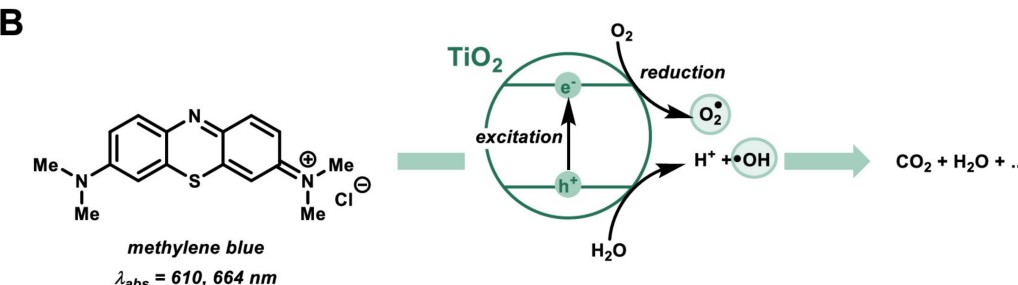

**Figure 36.** A schematic of the photocatalytic degradation of MB over $TiO_2$ photocatalyst using a photochemical rotor-stator spinning disk reactor (pRS-SDR). (**A**) The pRS-SDR facilitates the scaling up of complex heterogeneous reaction conditions. (**B**) $TiO_2$ photocatalyst enabled the degradation of MB. Reproduced with permission from ref. [265].

Despite all extensive efforts in the field of industrial photocatalysis, there is still a strong desire to develop appropriate semiconductor photocatalysts with high quantum efficiency and a broad spectrum response [266]. In this context, continuous studies have been carried out to maximize the utilization of solar energy and improve the charge carrier separation. A recent and highly promising approach in photocatalysis involves the integration of up-conversion (UC) photocatalyst particles, which convert near-infrared (NIR) photons into UV-visible light. This integration shows the potential ability to harness NIR light in a photocatalytic process. As a result, using up-conversion materials in designing efficient photocatalysts is a promising approach to improve the utilization of a wide range of solar energy by semiconductors [267]. Zhang et al. [266] synthesized $Lu_3NbO_7$:Yb, Ho/CQDs/$AgInS_2$/$In_2S_3$ nanocomposite by using carbon quantum dots (CQDs) as an electron mediator and $Lu_3NbO_7$:Yb, Ho as an up-conversion material, and studied its photocatalytic activities in Cr(VI) reduction and $H_2O_2$ production, simultaneously. Under visible light irradiation, the as-prepared photocatalyst achieved a remarkable efficiency of 99.9% and 78.5% within 15 and 30 min for Cr(VI) reduction at 20 and 40 ppm, respectively. Additionally, under NIR light irradiation, they achieved a significant efficiency of 94.0% Cr(VI) reduction at 20 ppm within 39 min. Simultaneously, the photocatalyst effectively generated 902.9 μM $H_2O_2$ within a 5-hour under visible light illumination. They proposed the Z-scheme electron transfer mechanism of the process, as illustrated in Figure 37. As can be seen, CQDs enhanced light absorption in the $Lu_3NbO_7$:Yb, Ho up-conversion material, leading to transitions between energy levels and emission of green and red light. Subsequently, the emitted light activated heterojunctions, generating electron–hole pairs. Electrons in the CB of the $AgInS_2$ were transferred to the CQDs and combined with holes in the VB of the $In_2S_3$, resulting in stronger fluorescence. The CB potential facilitated the reduction of Cr(VI) to Cr(III) and generated $H_2O_2$ through electron transfer with $O_2$. The enhanced photocatalytic performance has been attributed to improved light absorption, efficient solar-to-energy conversion via $Lu_3NbO_7$:Yb, Ho/CQDs up-conversion property, and accelerated charge transfer in the Z-schematic pathway.

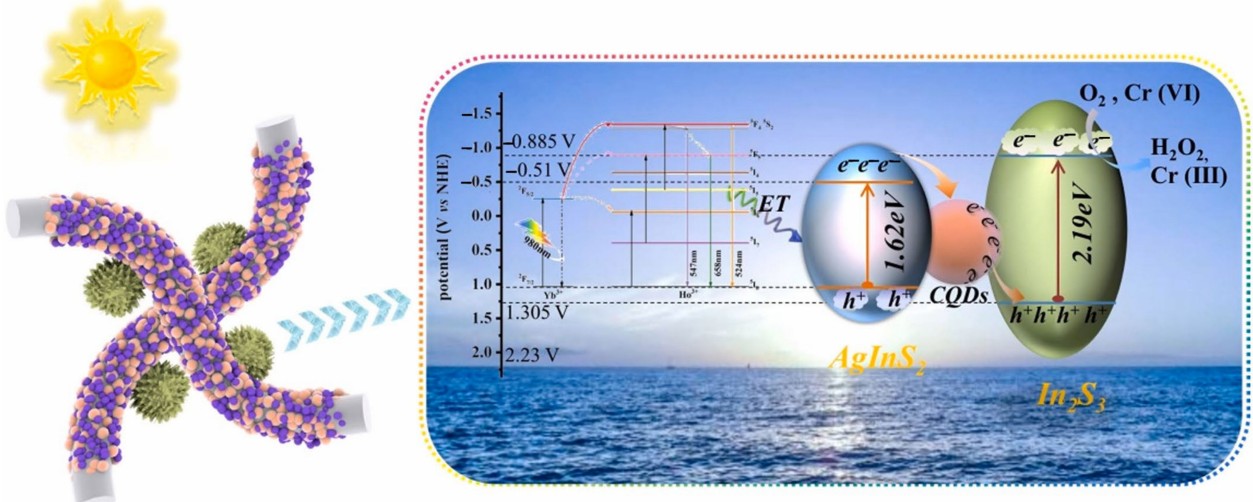

**Figure 37.** The Z-scheme electron transfer mechanism for $H_2O_2$ production and removal of Cr (VI) over $Lu_3NbO_7$: Yb, Ho/CQDs/$AgInS_2$/$In_2S_3$ heterostructure. Reproduced with permission from ref. [266].

Another effective strategy for improving charge separation involves the generation of an internal electric field in a semiconductor. The induced electric polarization facilitates the separation of photogenerated charge carriers, allowing them to migrate in opposite directions and effectively preventing charge recombination [268]. Yu et al. [269] designed the polarization and built-in electric field in the bulk structure of NiO by introducing

highly dispersed erbium (Er) atoms into the unit cell of NiO. The density functional theory (DFT) calculations and experimental characterizations revealed that introducing Er dopant atoms induced localized lattice distortion. These properties facilitated changes in dipole moments and polarization and increased the surface potential difference, leading to efficient charge separation. Furthermore, introducing highly dispersed Er atoms in the n-type NiO exhibited a significant enhancement in both adsorption and activation of $CO_2$, leading to a remarkable reduction in the energy barrier of $CO_2$ photoreduction resulting in a CO yield of 368 $\mu$mol g$^{-1}$ h$^{-1}$ at an optimized ($\approx$2%) Er-doped NiO.

From an economical and industrial viewpoint, the combination of the up-conversion (UC) active materials with an appropriate semiconductor photocatalyst, along with the application of external electric or magnetic fields, can provide state-of-the-art future perspectives in designing highly efficient and practical semiconductor photocatalysts. These advances in photocatalyst technology can remarkably provide the projected growth of the global photocatalyst market. According to a document reported in ref. [270], the global photocatalyst market was valued at USD 2.51 billion in 2021. It is further projected to show a compound annual growth rate (CAGR) of 9% during the period 2023–2032 (Figure 38). These findings highlight the potential opportunities in the photocatalyst market and emphasize the importance of continued research progress in this field to meet the growing demand for clean energy production and environmental remediation.

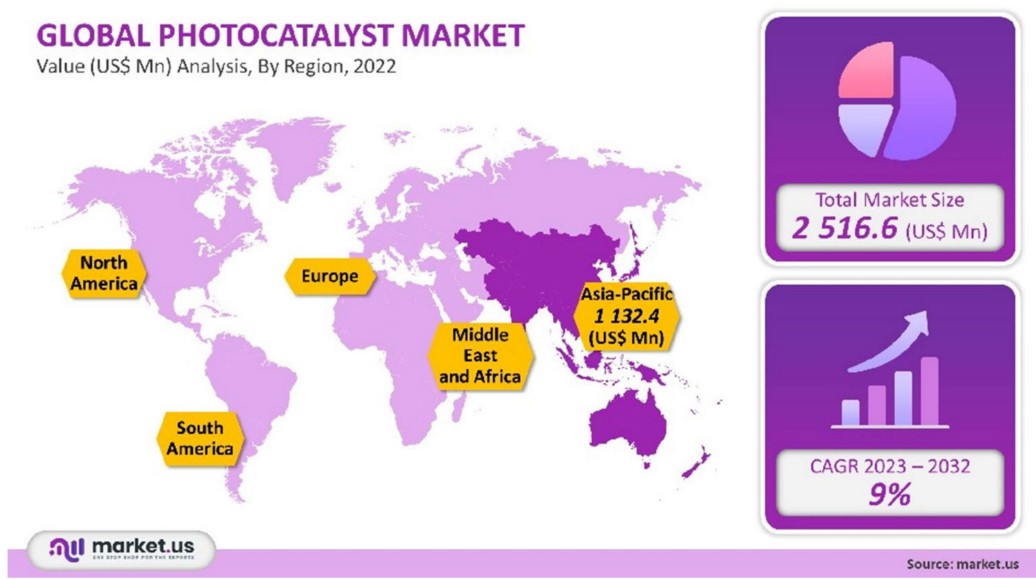

**Figure 38.** Global photocatalyst market prediction during 2023–2032 [270].

## 9. AI-Assisted Photocatalyst Design

Very recently, artificial intelligence (AI) and machine learning (ML) methods have evolved as powerful tools to revolutionize the discovery of electrocatalysts [271] and photocatalysts [272]. The term "machine learning" often refers to the methods where a simulation of the correlation between a designated reference or input characteristics and the parameters of the output to be anticipated from the input is "learned" from an appropriate training dataset. Oral et al. [273] used machine learning method to analyze a vast dataset comprising 10,560 data points from 584 experiments reported in 180 academic articles concerning photoelectrochemical water splitting over n-type semiconductors. They have used a predictive model developed by random forest statistics to identify the patterns in the data for establishing a relationship between photocurrent density and 33 descriptors, including the type of electrode, preparation methods, the light irradiation condition, and electrolyte solution. The obtaining band gap of the electrode was remarkably good, with the root mean square error of validation and testing 0.24 and 0.27, respectively.

In general, as the complexity of an ML model increases, the requirement for a more extensive training dataset also increases. As the models have been trained, it is possible to infer catalyst activity without performing actual experiments or simulations. Figure 39a illustrates the application of data-driven ML to speed the active photocatalysts finding within the field of photocatalysis. Predictive ML models have the potential to provide cost-effective means of determining photocatalytic activity based on catalyst properties, thus reducing the need for experimental and, or computational methods that are conventionally employed. This process can significantly minimize the efforts and resources required to determine the photocatalytic activity. In addition, merging domain knowledge with data-driven ML model training is one strategy for dealing with a limited amount of data.

From the heterogeneous catalysis perspective, many kinds of photocatalysis domain knowledge are presented, followed by current data-driven ML developments. Figure 39b shows how combining domain knowledge with data-driven ML at different phases adds to and matches the process of discovering photocatalysts [272]. As a result, the training of precise predictive models leads to a reduction in the dependence on experiments and simulations, which in turn allows for efficient screening of photocatalysts.

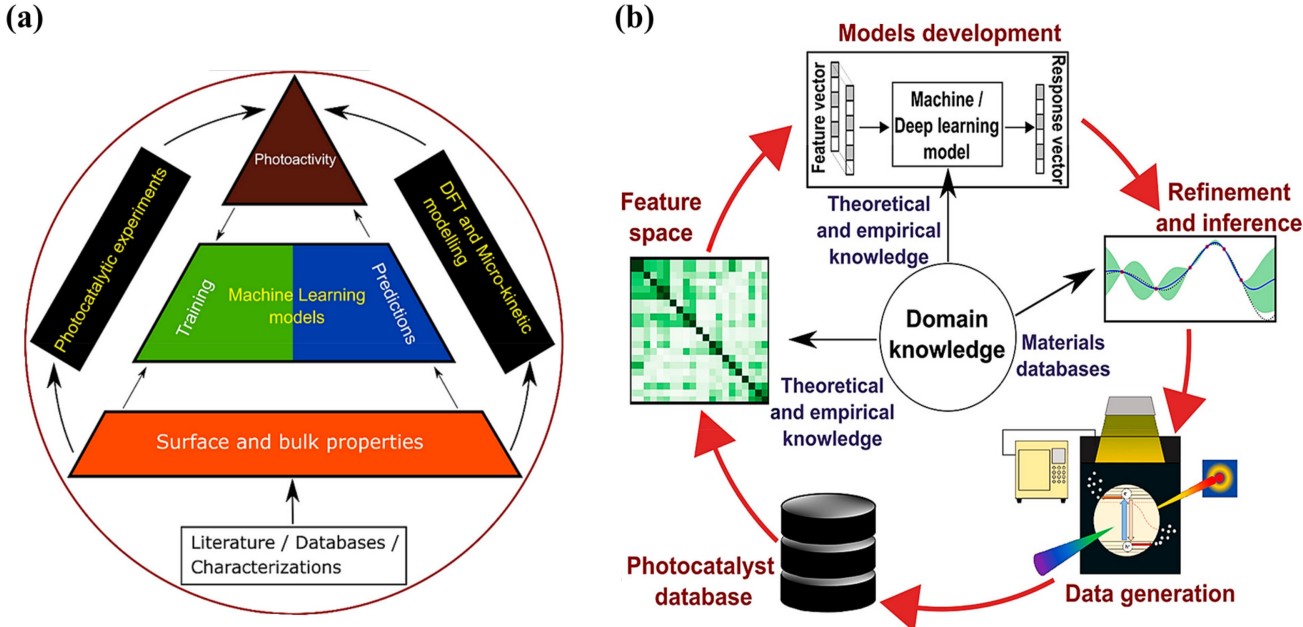

**Figure 39.** (**a**) Depicts the utilization of ML as a means to accelerate the process of determining property-activity; (**b**) integration of photocatalysis domain knowledge into the ML framework adds and matches the entire photocatalyst discovery process. Reproduced with permission from ref. [272].

ML shows potential in offering reliable predictions regarding the selection of dopants for PEC systems that exhibit superior performance. The analysis of correlations between numerous dopant features and the photoelectrochemical performance of doped photoelectrodes is a valuable method for revealing previously unclear linkages [274]. Wang et al. [274] successfully constructed an ML model capable of predicting the doping impact of 17 metal dopants on hematite ($Fe_2O_3$), a representative photoelectrode substance. Figure 40 outlines a framework for examining the impact of dopants for their inherent structural characteristics, as documented in database S. The database S comprises 11 descriptors, such as atomic number (N), ionic radius ($r_i$), atomic radius ($r_a$), single molecular bond covalent radius ($r_c$), chemical valence (Z), M-O bond formation enthalpy based on metal and oxygen, electronegativity ($\chi$), and melting point of pure metal ($T_m$).

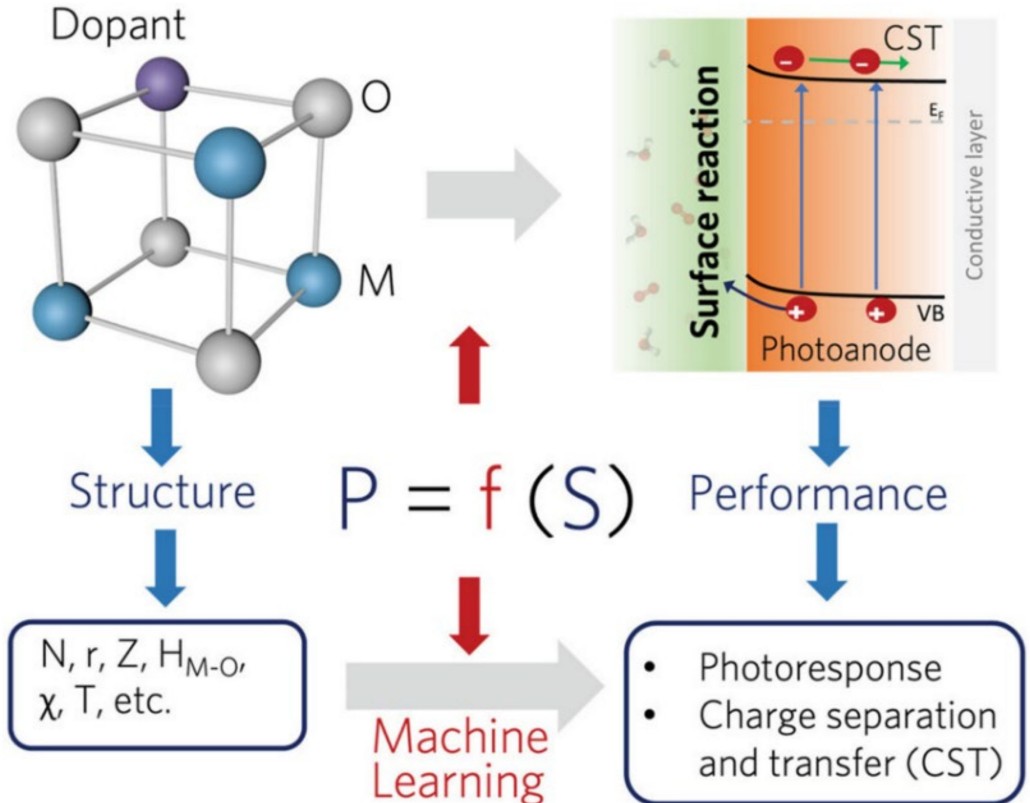

**Figure 40.** A schematic illustration of ML guided dopant selection for an efficient photoelectrochemical process. Reproduced with permission from ref. [274].

Additionally, Scientists have used various ML methods to predict pollutant removal through photocatalytic reactions, using different operational variables as input. Gheytanzadeh et al. [275] proposed the ML method of Gaussian process regression (GPR) technique with four kernel functions to calculate the photodegradation of tetracycline (TC) from wastewater using metal–organic frameworks (MOFs). The properties of their investigated MOFs are considered with the following parameters: surface area and pore volume, as well as operational factors such as light exposure duration, the dosage of catalyst, the concentration of TC, and pH. All of these factors were used as input parameters, and the results confirmed that the GPR-Matern model, which provides the most reliable predictions for experimental TC photodegradation. The sensitivity analysis also showed that light exposure duration and surface area are the most influential parameters in TC photodegradation. Overall, this study provides valuable insights for researchers to optimize conditions for eliminating TC in the wastewater. Moreover, ML is a predictive approach obtaining patterns in complex datasets without basic knowledge. For example, ML can identify a relationship between increased co-catalyst loading and improved alcohol reforming performance without explaining for its superior activity [272]. Based on the above discussion, ML demonstrates a significant potential in evolving photocatalysts discovery practices, accelerating photocatalyst selection, and optimizing photocatalytic reaction conditions that provide tremendous cost and time savings for a desired photocatalytic process. However, this predictive model is still at its infancy stage [272,276].

## 10. Conclusions and Future Research Directions

Global population growth and industrialization have significantly increased the demand for obtaining clean energy and removing organic and inorganic pollutants for a safer environment, thus achieving a sustainable society. In this review, the recent advancement and progress in designing semiconductors-based photocatalysis to clean energy production and environmental remediation by using solar energy have been

presented. We discussed various semiconductor structure modification strategies, as well as reaction conditions, as the most critical parameters affecting the efficiency of photocatalytic processes. We introduced several potential areas for future research directions: Firstly, there is the opportunity to develop new photocatalytic materials with improved efficiency, selectivity, and reusability through the synthesis of novel materials or modification of existing materials. Secondly, optimizing the semiconductor structure for the synthesis of flexible and more stable photocatalysts with self-cleaning and flame-resistance properties is of an innovative strategy to improve their photocatalytic performance in a wide range of applications. Thirdly, the design of photocatalytic systems that are not only active under natural sunlight, but can maintain their activity in the absence of light is of promising outlook to extend the photocatalyst lifetime for a long time. Fourthly, boosting photocatalytic reactions by applying various external fields such as magnetic, electric, and piezoelectric fields could lead to the development of more efficient photocatalysts via enhancing light absorption, charge separation, and surface reactions. Fifthly, advanced characterization analyses could also be performed for a better understanding of the kinetics and mechanisms of the photocatalytic reactions in order to design more effective photocatalytic systems for clean energy production and environmental remediation. Another noteworthy future trend that has attracted a great deal of attention very recently is the design of single-atom catalysts to achieve high catalytic activity and selectivity and reduce practical costs. Isolation of dispersed atoms or coordination atoms with surface atoms on a suitable support is the main reason for maximizing the atomic efficiency of metals in these systems. Finally, simultaneous online analysis of photocatalyst performance using in-situ imaging techniques such as scanning tunneling microscopy (STM) during its working under actual conditions, called operando characterization, helps to achieve a comprehensive understanding of the photocatalytic reaction. Figure 41 shows future research directions in a photocatalytic process.

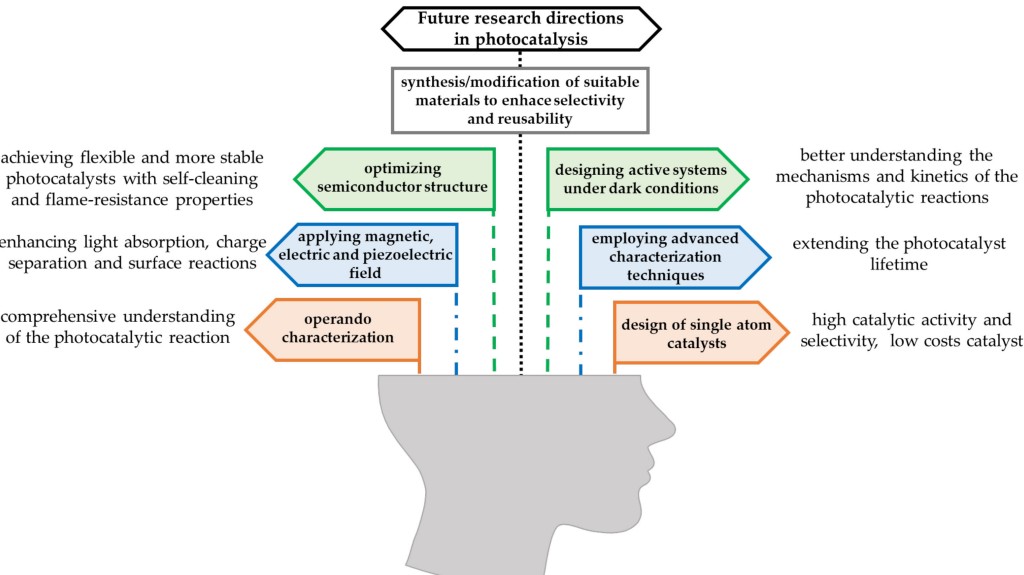

**Figure 41.** A schematic illustration of future research direction in a photocatalytic process.

**Author Contributions:** N.G.: Conceptualization, discussion, writing—original draft preparation, review and editing; Z.A.-P.: Drew the schematic diagrams, discussion, writing—review and editing; E.K.: Discussion, writing—review and editing; A.Z.M.: Supervision, conceptualization, writing—review and editing; All authors have read and agreed to the published version of the manuscript.

**Funding:** A.Z.M. thanks Iran National Science Foundation (Research Chair Award of Surface and Interface Physics, Grant No. 940009) for financial assistance.

**Acknowledgments:** The authors would like to thank Research Council of the Sharif University of Technology for supporting this project.

**Conflicts of Interest:** The authors declare no conflict of interest.

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
