# Peer review of "Recent Progress on Semiconductor Heterogeneous Photocatalysts in Clean Energy Production and Environmental Remediation"

_catalysts, doi:10.3390/catal13071102_

Round 1

Reviewer 1 Report

In this work the authors presented a systematic comprehensive review article about the progress of semiconductor heterogeneous photocatalysts in clean energy production and environmental remediation. The manuscript is well written, and the authors include many parameters that can affect the photocatalyst performance and its working process. This manuscript can be a good source for researchers in the field of photocatalysts. In addition, the authors did great work by choosing the most recent published articles. Therefore, I recommend this article for publication after minor revision.  

My comments as below:

1.      Please add more schematics.

2.      Can the authors add schematics to show the history of photocatalyst. When did it develop and where it was reached today?

3.      Please add a section about the real use/commercial use of the photocatalyst in industry. Same as well please add the market prediction of the photocatalyst. these information is available online and can strengthen the article and can attract the researchers.

Minor editing of English language required

Author Response

Reviewer 1:

Dear reviewer:

Thank you very much for devoted time and efforts for reviewing the manuscript. We greatly appreciate your valuable comments that helped to improve the manuscript. We have tried to answer point by point and modified the manuscript.

Point 1: Please add more schematics.

Response 1:

Thank you very much for the comment. According to your effective comment, we added one new schematic consists of two parts (H2 production from water splitting and CO2 reduction) in the introduction section of the revised manuscript.

“ Inspiring natural photosynthesis, Figure 1 illustrates the application of solar energy by using semiconductor photocatalysts toward hydrogen generation from water splitting (Figure 1 (a)) and CO2 reduction to valuable chemicals (Figure 1 (b)).”

Figure 1. A schematic diagram of (a) hydrogen production from water splitting; and (b) reduction of CO2 to useful chemicals using solar light.(attached in the file)

Point 2: Can the authors add schematics to show the history of photocatalyst. When did it develop and where it was reached today?

Response 2:

Thank you very much for the valuable comment. As requested we provided a timeline of hydrogen energy, highlighting its evolution from atom to fuel cell and its potential as a chemical fuel.

The following ststement is added and highlighted in the introduction section “A timeline of hydrogen energy, highlighting its evolution from atom to fuel cell, and its potential application as a chemical fuel is schematically shown in Figure 2. Several research organizations have reported the practical feasibility of hydrogen-based fuel cells. Between 1950 and 2000, extensive research was carried out in developing of infrastructure and storage mechanisms from small to large-scale applications. Since 2000, the focus has shifted towards the commercializing of hydrogen energy and conducting system integration research. Based on this trend, it is predicted that by 2040, there will be significant growth in the hydrogen energy market and an increasing number of initiatives to replace conventional fuels with hydrogen energy[10,20]. “

Figure 2. Timeline of hydrogen energy. Reproduced with permission from ref.[10].(attached in the file)

Then a a schematic diagram of the historical evolution of photocatalysts for water contaminant removal with brief discussion is provided.

“Concerning the history of photocatalysis, it is established that the idea of artificial photosynthesis was first proposed by Verne[46] in 1874, but it took over 100 years for its realization in the lab. The light-driven oxygen evolution was first carried out by Boddy [47] in 1968 using n-type rutile (TiO2). Müller's report in 1969 on the ability of ZnO to decompose isopropanol under UV light, demonstrated the potential of photo-catalysts to degrade organic contaminants in water. Subsequently, a TiO2-based semiconductor and a platinum-black electrode were historically employed by Fujishima and Honda in 1972 for photoelectrochemical (PEC) water splitting. In this period, ex-tensive research was done on a wide range of metal oxides (e.g., ZnO, NiO, and WO3), metal sulfides (e.g., CdS, and ZnS), and mixed metal oxides (e.g., SrTiO3) to degrade a wide range of organic contaminant. Between 1994 and 2000, significant progress was made in understanding the phenomenon and mechanism of photocatalysis due to the development of semiconductor band theory and the utilization of advanced charac-terization techniques. Since 2000, nanotechnology has attracted considerable atten-tion, leading to the development of various nano-sized photocatalysts with several advantages, including increased charge transfer rates, larger specific surface area, and increased active sites. In 2011-present photocatalytic recycling attracted increasing attention due to the biosafety aspects of applying nanomaterials. The subsequent steps focused on developing many visible light photocatalysts to enhance photoconversion efficiency. Notably, research focus has shifted towards areas such as black TiO2, non-TiO2 junctions, plasmon-based photocatalysts, and catalyst design and tailoring its structure for a specific need [48,49]. Figure 5 shows a schematic diagram of historical evolution of photocatalysts to remove water contaminants. Hence, the development of the photocatalytic process by using highly effective semiconductors with suitable bandgap positions has attracted a great deal of interest concerning clean energy generation, and environmental restoration in recent years. Based on the Scopus citation database during 2005–2023 through keywords, “photocatal*”, “environment*” and “energy” in title, abstract, and keywords, the number of publications on photocatalytic energy production and environmental remediation shows a continuous increasing trend which indicates the importance of this active field (Figure 6).”  

Figure 5. A schematic diagram of the historical evolution of photocatalysts for water contaminant removal. Reproduced with permission from ref [48].

Point 3: Please add a section about the real use/commercial use of the photocatalyst in industry. Same as well please add the market prediction of the photocatalyst. these information is available online and can strengthen the article and can attract the researchers.

Response 3:

Thank you very much for the attractive comment. we added a new section (section 9) in the manuscript to illustrate the commercial and real applications of photocatalysts with several examples in the both field of clean energy production and environmental remediation. At the end of this section, market prediction of the photocatalyst based on Data Bridge Market Research report is provided.

“8. Industrial photocatalyst application

It is widely accepted that photocatalysts with promising industrial applications must possess three critical features: (i) low-cost production, (ii) effective charge carrier separation (iii) an ideal band gap to facilitate optimal utilization of the solar spectrum [263]. Considering the promising results obtained by several photocatalysts in energy production and environmental remediation at the laboratory scale, researchers are intensifying their efforts to advance the industrialization of photocatalyst technology for the applications mentioned above. In the field of clean energy production, Lee et al. [264] synthesized a floatable photocatalytic platform fabricated from porous elastomer–hydrogel nanocomposites for scalable solar H2 production. The platform's scalability, as an essential factor for industrialization, was demonstrated under natural sunlight. It was confirmed that the floatable photocatalytic platform shows an excellent H2 evolution rate of 163 mmol h–1 m–2 in the presence of Pt/TiO2 cryoaerogel with an area of 1 m2 using Cu/TiO2 photocatalysts. Furthermore, the system generates 79.2 mL of hydrogen per day under natural sunlight. The schematic of the nanocomposite and its advantages for photocatalytic hydrogen production (i-vi) has been shown in Figure 35.

Figure 35. A schematic illustration floatable photocatalytic platforms along with its advantages in hydrogen evolution reaction (i) light delivery; (ii) facile gas separation; (iii) enhanced surface tension; (iv) stable catalyst immobilization; (v) suppressed back-oxidation (reverse reaction) ; and (vi) facile supply of water. Reproduced with permission from ref. [264].

In the field of environmental remediation, Chaudhuri et al. [265] developed a photochemical rotor−stator spinning disk reactor (pRS-SDR) capable of handling and scaling solid-containing photochemical reactions on the flow condition. They investigated the efficiency of the reactor in handling slurries for the TiO2-mediated aerobic photodegradation of methylene blue (MB). By utilizing a high-speed rotating disk, the pRS-SDR achieved uniform mixing of the reaction components, enhanced mass transfer, and optimized irradiation throughout the reaction mixture. This innovative system is expected to facilitate the shift from batch to continuous-flow processes in the pharmaceutical and agrochemical sectors. Its ability to scale up complex multiphase synthetic reactions makes it a valuable tool in advancing industrial operations. The schematics of photocatalytic degradation of MB using TiO2 photocatalyst in the pRS-SDR are illustrated in Figure 36. The highest MB photodegradation rate of 90.0% was reported at a liquid flow rate of 19.5 mL/min [265].

Figure 36. A schematic of the photocatalytic degradation of MB over TiO2 photocatalyst using a photochemical rotor-stator spinning disk reactor (pRS-SDR). (a) The pRS-SDR facilitates the scaling up of complex heterogeneous reaction conditions. (b) TiO2 photocatalyst enabled the degradation of MB. Reproduced with permission from ref. [265].(attached in the file)

Despite all extensive efforts in the field of industrial photocatalysis, there is still a strong desire to develop appropriate semiconductor photocatalysts with high quantum efficiency and a broad spectrum response [266]. In this context, continuous studies have been carried out to maximize the utilization of solar energy and improve the charge carrier separation. A recent and highly promising approach in photocatalysis involves the integration of up-conversion (UC) photocatalyst particles, which convert near-infrared (NIR) photons into UV-visible light. This integration shows the potential ability to harness NIR light in a photocatalytic process. As a result, using up-conversion materials in designing efficient photocatalysts is a promising approach to improve the utilization of a wide range of solar energy by semiconductors [267]. Zhang et al. [266] synthesized Lu3NbO7:Yb, Ho/CQDs/AgInS2/In2S3 nanocomposite by using carbon quantum dots (CQDs) as an electron mediator and Lu3NbO7:Yb, Ho as an up-conversion material, and studied its photocatalytic activities in Cr(VI) reduction and H2O2 production, simultaneously. Under visible light irradiation, the as-prepared photocatalyst achieved a remarkable efficiency of 99.9% and 78.5% within 15 and 30 minutes for Cr(VI) reduction at 20 and 40 ppm, respectively. Additionally, under NIR light irradiation, they achieved a significant efficiency of 94.0% Cr(VI) reduction at 20 ppm within 39 minutes. Simultaneously, the photocatalyst effectively generated 902.9 μM H2O2 within a 5-hour under visible light illumination. They proposed the Z-scheme electron transfer mechanism of the process, as illustrated in Figure 37. As can be seen, CQDs enhanced light absorption in the Lu3NbO7:Yb, Ho up-conversion material, leading to transitions between energy levels and emission of green and red light. Subsequently, the emitted light activated heterojunctions, generating electron-hole pairs. Electrons in the CB of the AgInS2 were transferred to the CQDs and combined with holes in the VB of the In2S3, resulting in stronger fluorescence. The CB potential facilitated the reduction of Cr(VI) to Cr(III) and generated H2O2 through electron transfer with O2. The enhanced photocatalytic performance has been attributed to improved light absorption, efficient solar-to-energy conversion via Lu3NbO7:Yb, Ho/CQDs up-conversion property, and accelerated charge transfer in the Z-schematic pathway.

Figure 37. The Z-scheme electron transfer mechanism for H2O2 production and removal of Cr (VI) over Lu3NbO7: Yb, Ho/CQDs/AgInS2/In2S3 heterostructure. Reproduced with permission from ref.[266].(attached in the file)

Another effective strategy for improving charge separation involves the generation of an internal electric field in a semiconductor. The induced electric polarization facilitates the separation of photogenerated charge carriers, allowing them to migrate in opposite directions and effectively preventing charge recombination [268]. Yu et al. [269] designed the polarization and built-in electric field in the bulk structure of NiO by introducing highly dispersed erbium (Er) atoms into the unit cell of NiO. The density functional theory (DFT) calculations and experimental characterizations revealed that introducing Er dopant atoms induced localized lattice distortion. These properties facilitated changes in dipole moments and polarization and increased the surface potential difference, leading to efficient charge separation. Furthermore, introducing highly dispersed Er atoms in the n-type NiO exhibited a significant enhancement in both adsorption and activation of CO2, leading to a remarkable reduction in the energy barrier of CO2 photoreduction resulting in a CO yield of 368 μmol g−1 h−1 at an optimized (≈2%) Er-dopedNiO.

From an economical and industrial viewpoint, the combination of the up-conversion (UC) active materials with an appropriate semiconductor photocatalyst, along with the application of external electric or magnetic fields, can provide state-of-the-art future perspectives in designing highly efficient and practical semiconductor photocatalysts. These advances in photocatalyst technology can remarkably provide the projected growth of the global photocatalyst market. According to a document reported in ref. [270], the global photocatalyst market was valued at USD 2.51 billion in 2021. It is further projected to show a compound annual growth rate (CAGR) of 9% during the period 2023-2032 (Figure 38). These findings highlight the potential opportunities in the photocatalyst market and emphasize the importance of continued research progress in this field to meet the growing demand for clean energy production and environmental remediation.

Figure 38. Global photocatalyst market prediction during 2023-2032 [270].(attached in the file)

Reviewer 2 Report

The manuscript reviews recent advances in semiconductor materials for heterogeneous photocatalysis with a particular focus on a targeted design of heterojunctions of different types. The structure of this review is well-balanced providing both current views on fundamental aspects of photocatalysis and a useful summary on the application aspects of these materials. The authors address all issues point by point following a clearly emphasized logical thread relating the structure of the semiconductor photocatalysts  and their functionalities, The time span of the literature adequately matches all the topics discussed.

This review will certainly be of interest for a broad audience comprising experts in modern heterogeneous photocatalysts as well as the researches only starting to explore their opportunities in this fields.

There are few points for revision:

-         Grammar and style need to be carefully revised throughout the text to harmonize tenses and numbers (e.g. “energy consumption worldwide rose by approximately 50% between 2018 and 2050” instead of “will rise” or “is expected to rise” etc.)

-         Al least a brief discussion of AOPs based on generation of singlet oxygen would be highly desirable to complete the picture

-         2D semiconductors (GO, rGO and MeXenes) and their hybrid combinations of potent chromophores such as porphyrins and their SURMOFs with would deserve some more attention, especially in a context of simultaneous photocatalysis (section 7). For citing, there are some recently published examples of such ambivalent photocatalysts potentially capable of both phooxidation and photoreduction (see, for example, Nugmanova A. G. et al. (2022). Interfacial self-assembly of porphyrin-based SURMOF/graphene oxide hybrids with tunable pore size: An approach toward size-selective ambivalent heterogeneous photocatalysts. Applied Surface Science579, 152080; Nikoloudakis E. (2022). A covalently linked nickel (ii) porphyrin–ruthenium (ii) tris (bipyridyl) dyad for efficient photocatalytic water oxidation. Chemical Communications58(86), 12078-12081 etc).

Please, check grammar and style carefully.

Author Response

Reviewer 2:

Dear reviewer:

Thank you so much for taking the time to review the manuscript. We greatly appreciate your thoughtful, effective and kind remarks that helped us to improve the manuscript. We have tried to answer the subjects you offered about the manuscript.

Point 1:  Grammar and style need to be carefully revised throughout the text to harmonize tenses and numbers (e.g. “energy consumption worldwide rose by approximately 50% between 2018 and 2050” instead of “will rise” or “is expected to rise” etc.)

Response 1:

Thank you very much for the comment. Apologizing for the mistake, we have corrected the sentence and revised the entire manuscript grammatically. “energy consumption worldwide is expected to rise by approximately 50% between 2018 and 2050”

Point 2: At least a brief discussion of AOPs based on generation of singlet oxygen would be highly desirable to complete the picture.

Response 2:

Thank you very much for the comment. According to your valuable comment, we have added a brief discussion of AOPs based on generation of singlet oxygen in the introduction part of the manuscriot (ref [31]-[35]).

“In addition to hydroxyl radicals, other reactive species including, hydrogen peroxide (H2O2), anion radical (O2), and singlet oxygen (1O2), can be generated in AOPs [31]. Meanwhile, 1O2 is a highly selective oxidant that can quickly oxidize electron-rich moieties in organic pollutants owning to its electrophilic nature. This form of oxygen species possesses a prolonged lifetime (nearly 2–4 μs) and high concentration (e.g., 10−14 − 10−11 M) in water and demonstrates excellent resilience to inorganic anions and natural organic matter in wastewater [32]. These properties make this species suitable for selectively oxidizing high-priority contaminants commonly found in wastewater, such as endocrine-disrupting chemicals (EDCs), antibiotics, and pharmaceuticals [33]. Besides, 1O2 can be utilized for the inactivation of pathogenic microorganisms and the elimination of antibiotic-resistance genes in both natural and drinking water [34]. Several AOPs systems, such as catalytic ozonation, hydrogen peroxide (H2O2), persulfate (PS) activation processes, and photocatalysis, can be applied to generate 1O2 [32]. In some ozone-based systems, O3 can be activated and produce free peroxide species (O2) that can be directly converted to 1O2 through an electron transfer mechanism. H2O2, as a common oxidant, can generate a variety of ROS after catalytic activation, followed by 1O2 generation in the system. In persulfate (PS) based AOPs, in the presence of carbon-based materials, the carbonyl (C=O) functional groups on the catalyst reacts with PS and generate 1O2. Among the aforementioned approaches photocatalysis is the most effective method to produce 1O2 through either a directly energy transfer on the photocatalyst or indirectly from other ROS. Despite all the advantages of 1O2, this species has low redox potential (0.81 V/SHE). As a result, its reaction rate with most organic compounds is lower than that of observed with radicals [35].”

Point 3: 2D semiconductors (GO, rGO and MeXenes) and their hybrid combinations of potent chromophores such as porphyrins and their SURMOFs with would deserve some more attention, especially in a context of simultaneous photocatalysis (section 7). For citing, there are some recently published examples of such ambivalent photocatalysts potentially capable of both phooxidation and photoreduction (see, for example, Nugmanova A. G. et al. (2022). Interfacial self-assembly of porphyrin-based SURMOF/graphene oxide hybrids with tunable pore size: An approach toward size-selective ambivalent heterogeneous photocatalysts. Applied Surface Science, 579, 152080; Nikoloudakis E. (2022). A covalently linked nickel (ii) porphyrin–ruthenium (ii) tris (bipyridyl) dyad for efficient photocatalytic water oxidation. Chemical Communications, 58(86), 12078-12081 etc).

Response 3:

Thank you very much for the comment. According to your constructive comment, we have mentioned the remarkable potential of these 2D semiconductors and their hybrid combinations of potent chromophores in the fabrication of simultaneous photocatalysts and added these references in the section 7 of the manuscript.

“For example, 2D semiconductors (e.g., GO, rGO, and MXenes) and their hybrid combinations can be applied to produce hydrogen generation and pollutant degradation simultaneously. Nugmanova A. G. et al. [248] synthesized zinc porphyrin metal-organic frameworks non-covalently attached to graphene oxide (SURMOF/GO) in Pickering emulsions, and investigated its photocatalytic activity during photodegradation of rhodamine 6G (Rh6G) and 1,5-dihydroxynaphtalene (DHN). In another study, Nikoloudakis et al. [249] fabricated a covalently linked nickel(II) porphyrin–ruthenium(II) tris(bipyridyl) dyad for photocatalytic water oxidation reaction in dimethylformamide (DMF) using methyl viologen as a sacrificial electron acceptor. As depicted in Figure 32, the NiP-Ru dyad demonstrated notably higher O2 evolution in comparison with the non-covalent system consisting of NiP and Rubpy and the turnover number (TON) of 18 was achieved for the NiP-Ru dyad after one hour of continuous visible light irradiation. This group acknowledged that utilizing an organic solvent (DMF) in this photocatalytic system offers the advantage of simultaneously coupling water oxidation and CO2 reduction catalysts within the same cell for efficient CO2 conversion.”

Figure 32. O2 evolution through photocatalytic H2O oxidation after irradiation with a 450 W Xenon lamp (with a λ > 420 nm cutoff filter) in the presence of 50 mM of Methyl Viologen in DMF and 4% H2O. Reproduced with permission from ref.[249].(attached at the file)

Reviewer 3 Report

Comments:

This minireview focuses on the recent progress on semiconductor heterogeneous photocatalysts in clean energy production and environmental remediation. In addition, the AI-assisted designer design is discussed in detail. However, before it is considered to publish in this journal, some questions should be focused as follows:

1.     Because the title of the article is two aspects of clean energy production and environmental remediation, but the introduction section has less description of clean energy production, please add a summary map on energy.

2.     Authors need to properly evaluate the strengths and weaknesses of selected examples instead of just summarizing these works.

3.     If possible, ask the author at the end of 3.1.6 to summarize how the inferences of the above heterojunctions are proved in other people 's work.

4.     Whether the author can add a few more citations in Section 3.2.4 Reaction temperature to better illustrate this point of view.

5.     In this article, sections 7 should be the same as sections 4, 5, and 6, with charts so that readers can see the synergies more directly, rather than just the representations.

6.     There are some inappropriate English words or expressions in the manuscript. The authors should carefully polish the English of the whole manuscript.

7.     The frontier background of photocatalysis can be referred to the following literature: Journal of Hazardous Materials 2022, 423: 127172; Advanced Functional Materials, 2022, 32: 2111999;

The further improvment  in English writing is needed.

Author Response

Reviewer 3:

Dear reviewer: Thank you so much for taking the time to review the manuscript. We greatly appreciate your thoughtful, effective and kind remarks that helped us to improve the manuscript. We have tried to answer the subjects you offered about the manuscript.  

Point 1: Because the title of the article is two aspects of clean energy production and environmental remediation, but the introduction section has less description of clean energy production, please add a summary map on energy.

Response 1:  Thank you very much for the comment. According to your constructive comment, we first provided one new schematic consists of two parts (H2 production from water splitting and CO2 reduction) in the introduction section of the revised manuscript.  “ Inspiring natural photosynthesis, Figure 1 illustrates the application of solar energy by using semiconductor photocatalysts toward hydrogen generation from water splitting (Figure 1 (a)) and CO2 reduction to valuable chemicals (Figure 1 (b)).”   Figure 1. A schematic diagram of (a) hydrogen production from water splitting and; (b) reduction of CO2 to useful chemicals using solar light.(attached at the file) Then we added more description of clean energy production in the introduction part of the manuscript. Moreover, a timeline of hydrogen energy, highlighting its evolution from atom to fuel cell and its potential as a chemical fuel is provided in the manuscript. “As a non-carbon-based energy carrier, hydrogen energy has attracted considerable attention due to its numerous advantages and may become a chemical fuel in the future. It has been proven that hydrogen has the potential to be a preferred alternative to reduce human society's reliance on non-renewable resources and decrease environmental concerns. Water, which covers 71% of the earth's surface, has exellent potential as a source of hydrogen energy. Using water power, we can extract H2 by using an appropriate photocatalyst. This process not only provides a means of generating energy, but also helps to reduce greenhouse gas emissions. While the high flammability of hydrogen is considered a disadvantage, it is also beneficial because it eliminates the need for additional energy to combust the fuel. Additionally, once released, hydrogen diffuses quickly and is non-toxic. These properties make hydrogen a potentially safe and efficient energy source for the future [10].  “A timeline of hydrogen energy, highlighting its evolution from atom to fuel cell, and its potential application as a chemical fuel is schematically shown in Figure 2. Several research organizations have reported the practical feasibility of hydrogen-based fuel cells. Between 1950 and 2000, extensive research was carried out in developing of infrastructure and storage mechanisms from small to large-scale applications. Since 2000, the focus has shifted towards the commercializing of hydrogen energy and conducting system integration research. Based on this trend, it is predicted that by 2040, there will be significant growth in the hydrogen energy market and an increasing number of initiatives to replace conventional fuels with hydrogen energy[10,20]. “   Figure 2. Timeline of hydrogen energy. Reproduced with permission from ref.[10].(attached at the file)  

Point 2: Authors need to properly evaluate the strengths and weaknesses of selected examples instead of just summarizing these works.

Response 2:  Thank you very much for your comment. With respect to your valuable comment, due to space limitations and simplicity, it is not possible to present the details of all examples in the manuscript. However, the explanations and examples provided in the manuscript effectively illustrate the general concept of each section. For example, In this particular case, an example was provided to highlight the use of metal decoration as a powerful strategy for improving the performance of photocatalysts. “Gu et al. [117] reported that the introduction of AgNPs in 4,4,4,4-(porphyrin-5,10,15,20-tetrayl) tetrakis (benzoic acid) (TCPP) ligand into UiO-66-NH2Zr-TCPP (AZT5) (Zr-TCPP) results in a remarkable enhancement in the photodegradation efficiency of Cr(VI). They have found rapid charge separation, and strong visible light absorption increased the reaction rate constants of the AgNPs-modified semiconductors to about 3.6–5.4 times compared to that of Zr-TCPP (0.075 ∼ 0.114 min−1) as shown in Figure 10 (b). Therefore, the results obtained from these types of studies confirm the participation in co-catalyst decoration can boost the activity of a photocatalytic reaction.” Overall, the example demonstrates how the incorporation of metals as co-catalysts can enhance the photocatalytic activity and the reader can refer to the article for more detailed information. However, we summerized some examples of commonly used photoanode materials for PEC water oxidation with their advantages and disadvantages. Besides the advantages and disadvantages of Surface modification strategies to improve photocatalytic performance of g-C3N4, as a widly used semicinductor, are summerize in table 2. In general, due to inherent limitations in performance and defects in properties such as low solar light absorption, poor charge-carrier transportation, and severe photocorrosion under illumination, a photoanode made of a single semiconductor cannot meet the required performance of either photocurrent or conversion efficiency for energy production applications such as hydrogen production from the photoelectrochemical splitting of water (Table 1).  As a result, effective methods such as surface modification must be utilized to improve photoanode properties. What is fascinating is that one of the state-of-the-art in PEC water splitting is producing working photoanodes paired with cocatalysts to overcome the constraints of single material. Advantages and disadvantages of popular photoanode materials for PEC water oxidation are listed in Table 2.   Table 1. Advantages and disadvantages of common photoanode materials for PEC water oxidation [1].(attached at the file) Photoanodes Advantages Disadvantages Fe2O3 Earth-abundance; Nontoxicity; Good photochemical stability; Narrow bandgap (2.2 eV). Low absorption coefficient; Short excited-state lifetime (3–10 ps); Poor oxygen evolution kinetics; Short hole diffusion length (2–4 nm); Poor conductivity (≈10−2 cm2 V−1 s−1 at 20 °C). WO3 Stable in acid conditions (pH < 4); Moderate hole-diffusion length (≈150 nm); Highly tunable composition; Nonstoichiometric properties; Good electron transport properties. Wide bandgap of 2.7 eV; Unstable when pH > 4; Corrosion induced by the peroxo-species created during water oxidation. ZnO Good stability; Environmentally friendly; Inexpensive; High carrier mobility. Large bandgap of 3.2 eV; High recombination rate of electron–hole pairs. BiVO4 Suitable bandgap of 2.4 eV; Low onset potential for O2 evolution; Good stability. Low IPCE at lower potentials; Poor photocurrent stability; Low intrinsic carrier mobility, due to short carrier diffusion length of ≈70 nm; Poor carrier separation efficiency; Slow hole kinetics of oxygen evolution. TiO2 Highly stable over a wide range of pH values in aqueous environments upon illumination; Good catalytic properties; Appropriate valence band edges. Large bandgap of 3.2 eV; Low conductivity. CdS Moderate bandgap (2.4 eV). Unstable in water upon illumination; Low charge separation and transfer efficiency.   Many attempts have been made to increase the photocatalytic capability of pure g-C3N4, including surface modification of the photocatalyst structure, in order to overcome the individual shortcoming of pure g-C3N4. In general, surface modification strives to improve photocatalyst specific surface area, charge separation, and optical properties. There are currently five modification strategies under investigation, including the incorporation of heteroatoms (metals and nonmetals) into the g-C3N4framework, noble metal deposition, hybridization of g-C3N4 with carbon nanomaterials, and coupling g-C3N4 with a photocatalyst. Table 2 summarizes the theory, benefits, and drawbacks of each strategy [2].     Table 2. Surface modification strategy used to improve photocatalytic performance of g-C3N4 [2].(attached at the file) Modification method Principle Advantages Disadvantages Metal doping Doping of various metallic species such as the alkali metals, rare earth metals and noble metals into g-C3N4 Bandgap narrowing, surface area improvement, charge separation and fine-tuning the band structure Can often cause secondary pollution due to leaching of the metal ions Non-metal doping Doping g-C3N4 with non-metals No secondary pollution, improve visible light absorption and charge separation Non-metal species does not take part in charge transportation hence recombination centers are formed Noble metal deposition Deposition of noble metal nanoparticles such as Cu, Pt, Au and Pd on g-C3N4 Metal content positively influence the photocatalytic activity until the optimum loading is reached Beyond the optimum metal loading, the excess metal ions act as recombination centers for the electron/hole pairs Hybridizing g-C3N4 with carbon nanomaterials (CNM) Carbon nanomaterial such as carbon nanotubes (CNTs), carbon nanospheres (CNS), graphene oxide (GO) and reduced graphene oxide (RGO) High thermal and electrical conductivity, remarkable adsorption properties for organic and inorganic compounds Excess CNM (i.e. RGO) facilitate adsorption of large amounts of the dye molecules onto the catalyst surface thereby reducing light penetration to the photocatalyst Coupling g-C3N4 with semiconductor Coupling two or more semiconductors to form a semiconductor heterojunction Improved stability, visible light utilization, charge separation and transfer and more efficient formation of the oxidizing species Difficult to find a proper semiconductor photocatalyst with suitable band edge position References: [1] Xu, Xiao-Ting, Lun Pan, Xiangwen Zhang, Li Wang, and Ji-Jun Zou. “Rational Design and Construction of Cocatalysts for Semiconductor-Based Photo-Electrochemical Oxygen Evolution: A Comprehensive Review.” Advanced Science 6, no. 2 (2019): 1801505. https://doi.org/10.1002/advs.201801505. [2] Fakhrul Ridhwan Samsudin, Mohamad, Nurfatien Bacho, and Suriati Sufian. “Recent Development of Graphitic Carbon Nitride-Based Photocatalyst for Environmental Pollution Remediation.” In Nanocatalysts, edited by Indrajit Sinha and Madhulata Shukla. IntechOpen, 2019. https://doi.org/10.5772/intechopen.81639  

Point 3: If possible, ask the author at the end of 3.1.6 to summarize how the inferences of the above heterojunctions are proved in other people 's work.

Response 3: Thanks to very much for the comment. According to your constructive comment, We have improved this section by providing a more comprehensive explanation of the proof of the inferences of the mentioned heterojunctions. In addition, we have added alternative examples to enhance clarity and supplemented this section with additional figures. “Ahmad et al. fabricated [160] AgI/CdS binary composite via an in situ precipitation method and examined its performance for photodegradation of methyl orange (MO) and tetracycline hydrochloride (TCH) under visible light illumination. The as-prepared photocatalyst showed high efficiency of 94.5 and 91% for MO and TCH degradation, respectively. Based on their results obtained from the active species trapping experiment, a mechanism of type-II heterojunction formation is suggested. As shown in Figure 15 (a), the addition of isopropyl alcohol (IPA), as •OH scavenger, did not show a significant effect on the photocatalytic activity of MO degradation. Whereas, adding benzoquinone (BQ) to the reaction mixture dramatically reduced the MO photodegradation from 94.5 to 23%, indicating that •O2- as an active reactive species played an essential role in the reaction mixture. Besides, the presence of EDTA-2Na resulted in a moderate reduction of 60% in MO degradation, indicating that h+ synergistically participated in the photodegradation process but to a lesser degree compared to •O2-. The schematic of the charge carrier transfer mechanism of the photocatalyst shows in Figure 15 (b). As shown in the Figure, the VB of AgI is higher positive than that of CdS (2.33 vs. 1.65), and the CB of CdS is higher negative than that of AgI (-0.73 vs. -0.47). Consequently, the electrons migrate to CB of AgI and react with O2 to generate •O2- . Subsequently, the generated •O2- species decomposed the organic pollutants. Simultaneously, the holes migrate to the VB of CdS and directly react with pollutants. Due to the relatively lower redox potential of CdS (+1.65) compared to the •OH/OH− (+2.38 eV vs. NHE) and the H2O/•OH (+2.72 eV vs. NHE), the generated holes are incapable of oxidizing OH− and H2O to form •OH.”       Figure 15. (a) Change in the MO concentrations in the absence and presence of various scavengers over AgI/CdS photocatalyst; (b) Schematic illustration of charge carrier transfer mechanism for MO and TCH photodegradation over the AgI/CdS photocatalyst. Reproduced with permission from ref [160].(attached at the file) In the case of clean energy production, Wang et al. [170], synthesized Bi2WO6/TiO2 photocatalyst through the solvothermal method and exhibited a remarkable efficiency of 12.9 mmol⋅g−1h−1 in H2 production. Based on the radical trapping experiment in the presence of 5, 5-dimethyl-1-pyrroline-N-oxide (DMPO) as an •OH scavenger, four distinct peaks corresponding to •OH have been observed (Figure 17 (a)). As depicted in Figure (17(b)), they showed that if the charge transfer mechanism is type II, the photogenerated electrons on the CB of TiO2 transfer to the CB of Bi2WO6, and the photogenerated holes on the VB of Bi2WO6 transfer to VB of TiO2 due to the more positive potential of the VB and CB of Bi2WO6 compared to those of TiO2. However, The ECB of Bi2WO6 (0.36 eV) exhibited a higher positive potential than the standard redox potential of H+/H2 (0 eV vs. NHE), making it impossible for electrons to reduce H+ to H2. Similarly, the EVB of TiO2 (2.75 eV) is nearly equivalent to the redox potential of H2O/•OH (2.72 eV vs. NHE), resulting in inefficient oxidation of H2O by holes to generate •OH. These results confirmed that the fabricated photocatalyst is not formed a type-II heterojunction. As illustrated in Figure 17(c), after light illumination and excitation of TiO2 and Bi2WO6, due to the more negative Fermi level of TiO2 compared to that of Bi2WO6, the electrons in TiO2 transferred to Bi2WO6. Followed by, the internal electric field formed at the interface of the Bi2WO6/TiO2 in the direction of TiO2 to Bi2WO6, and the migration of electrons and holes from CB of TiO2 to Bi2WO6 and VB of Bi2WO6 to TiO2 occurs. Then, the transformation of electrons on the CB of Bi2WO6 to VB of TiO2 along the Z direction resulting in the accumulation of electrons and holes with high redox potential in CB of TiO2 and VB of Bi2WO6 to participate in the photocatalytic reaction.   Figure 17. (a) ESR spectra of •OH; schematic illustration of photocatalytic mechanism: (b) traditional type-II, and (c) Z-scheme heterostructure. Reproduced with permission from ref [170].(attached at the file) The charge-migration pathway of the CZS/BMO heterojunction was examined using the electron spin resonance (ESR) technique. As depicted in Figure 23 (a), distinctive signals of DMPO−•O2− were observed for the BMO, CZS, and 1.0CZS/BMO, indicating their sufficient photoredox capacity to generate •O2− under visible-light irradiation. Notably, the signal intensity of DMPO−•O2− for 1.0CZS/BMO was significantly enhanced compared to the BMO and CZS, indicating electron aggregation on the CB of CZS, resulting in abundant •O2− production. In Figure 23 (b), no noticeable peaks of DMPO−•OH were detected for CZS due to its VB potential (1.34 V), which was lower than those of •OH/OH− (1.99 V) and •OH/H2O (2.38 V). However, the 1.0CZS/BMO photocatalyst exhibited a remarkable intensity of the DMPO−•OH signal, surpassing that of BMO, indicating an enhanced oxidation capacity achieved through the S-scheme heterojunction construction. These observations confirmed that photocarriers were separated and transferred in the S-scheme mechanism. The photocatalytic of this process is schematically illustrated in Figure 23 (c and d). Considering the lower position of the Fermi level of the BMO compared with that of the CZS, the electrons transferred from the CZS to BMO, which resulted in the creation of a built-in electric field. The presence of strong internal electric field (IEF) can effectively ensure that the photocarriers follow the separation pathway of the S-scheme. The built-in electric field induced an upward and downward band bending in the energy levels. Under the influence of IEF and bands bending, the photo-generated electrons in the CB of the BMO automatically migrated to the VB of the CZS and combined with the holes, In contrast, the electrons of the CZS and holes of the BMO were well preserved and participated in the reaction. The constructed S-scheme in this study provided a suitable pathway for electron transfer, resulting in a superior photocatalytic activity that yields the highest degradation efficiency of 76.3 % within 40 min [179].”   Figure 23. ESR spectra of (a) •O2- and (b) •OH for BMO, CZS and 1.0CZS/BMO under visible-light illumination; (c) Schematic illustration of the Cd0.5Zn0.5S/Bi2MoO6 S-scheme heterojunction. Reproduced with permission from ref. [179].  

Point 4: Whether the author can add a few more citations in Section 3.2.4 Reaction temperature to better illustrate this point of view.

Response 4: Thank you very much for the comment. According to your effective comment, we have added two recently published examples of the effect of reaction temperature on the photocatalytic H2 production and CO2 reduction in Section 3.2.4. “For example, Wang et al. [191] reported that the rate of the photocatalytic hydrogen production over C3N4/NixP/Red phosphorus Z‑Scheme photocatalyst during water splitting reaction enhanced from 1.78 to 6.16 μmol h−1 by increasing the reaction temperature up to 70 °C under visible light irradiation (Figure 27 (a)). In another study, Lais et al. [192] investigated the effect of reaction temperature on the rate of methanol formation during CO2 reduction. They observed that the reaction rate constant of methanol production increased with increasing temperature. This result was attributed to the enhanced diffusion and collision frequencies of the reactants at higher temperatures.”   Figure 27. (a) H2 production efficiency over the C3N4/NixP/RP photocatalyst under visible light irradiation at different temperatures. Reproduced with permission from ref [191]; (b)Arrhenius plot of ln (k0) vs. (1/T) for the photocatalytic degradation of Enrofloxacin over CdS/CuAg bi-alloy composite. Reproduced with permission from ref [193].(attached at the file)  

Point 5:  In this article, sections 7 should be the same as sections 4, 5, and 6, with charts so that readers can see the synergies more directly, rather than just the representations.

Response 5: Thank you very much for the comment. According to your valuable comment, we have provided a table (Table 4) consisting of various recently published examples of simultaneous pollutant photodegradation and H2 evolution (ref. [253-262]) in Section 7.       Entry   Photocatalyst   Synthesis method Photocatalytic test  condition H2 activity (μmol·h-1·g-1) Pollutant: photodegradation efficiency (%) Ref. 1 0.5 wt% Pt/Zn-V-20 Calcination, hydrothermal 300 W Xe lamp AM 1.5 G filter 0.5 M Na2SO4, 15% CH3OH 5230.4 2,2′,4,4′- tetrahydroxybenzophenone (BP-2): (99.6), methylene blue (MB): (99.4), acetaminophen (AAP ): (92.0) [253] 2 g-C3N4/BiOI/CdS calcination, solvothermal, and solution chemical deposition 300-W Xe lamp with a λ > 420 nm cutoff filter 863.44 bisphenol A: (98.62) [254] 3 Mo1@CNNTs template free polymerization 300 W xenon lamp coupled with λ = 420 nm cutoff filter. 4861 tetracycline hydrochloride: (97.3) [255] 4 [g-C3N4/polymethylmethacrylate (PMMA)]//[TiO2/polyaniline (PANI)/PMMA]//[self-assembled 3, 4, 9, 10-perylene tetraformyl diimide (PDI)/PMMA] (TMOP) tri-axial parallel electrospinning simulated sunlight 536.7 Ciprofloxacin: (88.99), tetracycline hydrochloride: (91.15), chlortetracycline hydrochloride: (77.55), levofloxacin: (69.51), and colored dye methylene blue: (92.50) [256] 5 SCN/NiS-1 hydrothermal visible light (400 nm filter) irradiation 700.9 Rhodamine B (RhB): (98.5) [257] 6 [email protected] hydrothermalXe lamp (CEL-PF300-T9, CEAU) with an AM1.5G filter36000Helianthine: (94.2) [258] 7 NbO-BRGO hydrothermal 300 W Xe > 400 nm 1742 crystal violet (CV): (97.6) [259] 8 ZnIn2S4@SiO2@TiO2 sol–gel and solvothermal 300 W xenon lamp 618.3 methylene blue: (99.7) [260] 9 Ag@TiO2-P25-5%MoS2 Combination of photocatalysts solar simulator composed of two white light bulbs (60 watts) 1792 Ciprofloxacin: (75) [261] 10 MoS2/ZnO hydrothermal 250 W metal halide lamp 235 Ciprofloxacin: (89) [262]        

Point 6:  There are some inappropriate English words or expressions in the manuscript. The authors should carefully polish the English of the whole manuscript.

Response 6: Thank you very much for the comment. According your cunstructive comment, we revised the entire manuscript grammatically and made corrections.  

Point 7:  The frontier background of photocatalysis can be referred to the following literature: Journal of Hazardous Materials 2022, 423: 127172; Advanced Functional Materials, 2022, 32: 2111999;.

Response 7: Thank you very much for the comment. According to your constructive comment, we referred to the frontier background of photocatalysis, specifically the following references: Journal of Hazardous Materials 2022, 423: 127172; Advanced Functional Materials, 2022, 32: 2111999, at the end of the Real/commercial use of the photocatalyst section. “Despite all extensive efforts in the field of industrial photocatalysis, there is still a strong desire to develop appropriate semiconductor photocatalysts with high quantum efficiency and a broad spectrum response [266]. In this context, continuous studies have been carried out to maximize the utilization of solar energy and improve the charge carrier separation. A recent and highly promising approach in photocatalysis involves the integration of up-conversion (UC) photocatalyst particles, which convert near-infrared (NIR) photons into UV-visible light. This integration shows the potential ability to harness NIR light in a photocatalytic process. As a result, using up-conversion materials in designing efficient photocatalysts is a promising approach to improve the utilization of a wide range of solar energy by semiconductors [267]. Zhang et al. [266] synthesized Lu3NbO7:Yb, Ho/CQDs/AgInS2/In2S3 nanocomposite by using carbon quantum dots (CQDs) as an electron mediator and Lu3NbO7:Yb, Ho as an up-conversion material, and studied its photocatalytic activities in Cr(VI) reduction and H2O2 production, simultaneously. Under visible light irradiation, the as-prepared photocatalyst achieved a remarkable efficiency of 99.9% and 78.5% within 15 and 30 minutes for Cr(VI) reduction at 20 and 40 ppm, respectively. Additionally, under NIR light irradiation, they achieved a significant efficiency of 94.0% Cr(VI) reduction at 20 ppm within 39 minutes. Simultaneously, the photocatalyst effectively generated 902.9 μM H2O2 within a 5-hour under visible light illumination. They proposed the Z-scheme electron transfer mechanism of the process, as illustrated in Figure 37. As can be seen, CQDs enhanced light absorption in the Lu3NbO7:Yb, Ho up-conversion material, leading to transitions between energy levels and emission of green and red light. Subsequently, the emitted light activated heterojunctions, generating electron-hole pairs. Electrons in the CB of the AgInS2 were transferred to the CQDs and combined with holes in the VB of the In2S3, resulting in stronger fluorescence. The CB potential facilitated the reduction of Cr(VI) to Cr(III) and generated H2O2 through electron transfer with O2. The enhanced photocatalytic performance has been attributed to improved light absorption, efficient solar-to-energy conversion via Lu3NbO7:Yb, Ho/CQDs up-conversion property, and accelerated charge transfer in the Z-schematic pathway.     Figure 37. The Z-scheme electron transfer mechanism for H2O2 production and removal of Cr (VI) over Lu3NbO7: Yb, Ho/CQDs/AgInS2/In2S3 heterostructure. Reproduced with permission from ref.[266].  Another effective strategy for improving charge separation involves the generation of an internal electric field in a semiconductor. The induced electric polarization facilitates the separation of photogenerated charge carriers, allowing them to migrate in opposite directions and effectively preventing charge recombination [268]. Yu et al. [269] designed the polarization and built-in electric field in the bulk structure of NiO by introducing highly dispersed erbium (Er) atoms into the unit cell of NiO. The density functional theory (DFT) calculations and experimental characterizations revealed that introducing Er dopant atoms induced localized lattice distortion. These properties facilitated changes in dipole moments and polarization and increased the surface potential difference, leading to efficient charge separation. Furthermore, introducing highly dispersed Er atoms in the n-type NiO exhibited a significant enhancement in both adsorption and activation of CO2, leading to a remarkable reduction in the energy barrier of CO2 photoreduction resulting in a CO yield of 368 μmol g−1 h−1 at an optimized (≈2%) Er-dopedNiO.      
